# A multiomic atlas of the aging hippocampus reveals molecular changes in response to environmental enrichment

Aging involves the deterioration of organismal function, leading to the emergence of multiple pathologies. Environmental stimuli, including lifestyle, can influence the trajectory of this process and may be used as tools in the pursuit of healthy aging. To evaluate the role of epigenetic mechanisms in this context, we have generated bulk tissue and single cell multi-omic maps of the male mouse dorsal hippocampus in young and old animals exposed to environmental stimulation in the form of enriched environments. We present a molecular atlas of the aging process, highlighting two distinct axes, related to inflammation and to the dysregulation of mRNA metabolism, at the functional RNA and protein level. Additionally, we report the alteration of heterochromatin domains, including the loss of bivalent chromatin and the uncovering of a heterochromatin-switch phenomenon whereby constitutive heterochromatin loss is partially mitigated through gains in facultative heterochromatin. Notably, we observed the multi-omic reversal of a great number of aging-associated alterations in the context of environmental enrichment, which was particularly linked to glial and oligodendrocyte pathways. In conclusion, our work describes the epigenomic landscape of environmental stimulation in the context of aging and reveals how lifestyle intervention can lead to the multi-layered reversal of aging-associated decline.

The process of aging involves a gradual decline in physiological functions which occurs throughout lifespan and is associated with a host of diseases, including cancer and neurodegeneration. Indeed, the strong link between aging and neurodegenerative pathologies such as dementia is evidenced by the clear correspondence of their symptoms, for instance with regards to cognitive deterioration[1]. Furthermore, the trajectories of both aging and neurodegenerative disorders are clearly impacted by lifestyle factors such as diet and physical and cognitive activity[2,3] such that lifestyle interventions may help ameliorate or even prevent the appearance of aging-associated diseases involving cognitive decline[4,5]. Against this backdrop, epigenetic mechanisms, being well-known mediators between the environment and the genomic response[6,7], may be of help in the design of targeted clinical interventions for healthy aging—i.e. aging linked to well-being[8]. However,

undertaking this task necessitates a better comprehension of the role of epigenetics in both aging and cognition, and the particular interaction between these two phenomena.

Epigenetic mechanisms have important roles across a wide spectrum of brain-related processes, ranging from core biological events such as neurogenesis[9] or neurodegeneration[10] to cognitive processes such as learning, memory and behaviour[11,12], many of which have been associated with physiological responses to external stimuli[13]. On the other hand, aging has also been extensively characterized from the epigenetic perspective[14,15], and in consequence, more recent studies have started to link aging-associated epigenetic changes with neurodegenerative disease molecular mechanisms[10]. Nonetheless, there is a lack of systematic studies which suitably integrate the two processes (brain aging and its environmental

✉e-mail: agustin.fernandez@cinn.es; mffraga@cinn.es

stimulation) so as to produce an unbiased characterization of their interrelations at a genome-wide scale[16,17].

In this study, we have generated a molecular atlas of the murine dorsal hippocampus encompassing gene and protein expression, DNA methylation, chromatin accessibility, histone modifications, and single cell expression and accessibility. We have characterized young and aged mice which were subjected to lifestyle stimulation in the form of environmental enrichment (EE). We have focused on the dorsal hippocampus because it is an important target of both cognitive and physical stimulus where adult neurogenesis occurs[18,19] and it is known to suffer aging-associated decline linked to cognitive deterioration[20]. On the other hand, the EE paradigm is a well-established system of general lifestyle stimulation (both cognitive and physical) linked to hippocampal changes at the cellular and molecular level[21]. Thus, with this in-depth dataset of aging and EE at both the bulk-tissue and single-cell level we have aimed to explore the molecular alterations associated with aging and environmental stimulation, and also their putative interactions.

## Results

### Aging induces bidirectional transcriptomic and proteomic signatures in the dorsal hippocampus

Our experimental set-up consisted of young and old mice exposed to a 2-month environmental enrichment (EE) intervention with control animals being kept under standard housing conditions (Fig. 1a, Methods; Supplementary Dataset 1; YC young control, YE young enriched, OC old control, OE old enriched). Following the intervention, we performed structural analyses via magnetic resonance imaging (MRI) and measured hippocampal dentate subgranular zone (SGZ) size and dentate gyrus (DG) volume through histology. Both the SGZ area and the DG volume of the hippocampus increased significantly after enrichment (Fig. S1a-b) but did not change with age. A similar age-independent increase of hippocampal volume was observed in MRI data (Fig. S1c), with total brain volume (Fig. S1d) not changing with enrichment. These data agree with prior knowledge on the effect of EE on hippocampal volume[22,23], confirming an expected brain region-specific response to enrichment. To explore the physiological impact of aging and cognitive stimulation, we performed a battery of behavioural tests and immunohistochemical analyses of the hippocampus (Methods). First, aging caused a notable disruption of neural markers in the SGZ (Fig. S1e-i), particularly: reduced levels of SOX2 + /GFAP+ neural stem cells, mitotic pH3+ cells and immature neuron subpopulations of the neurogenic population (DCX + /CLR- and DCX + / CLR+ cells, Fig. S1e−i), confirming the well-documented decrease in neuronal progenitor proliferation states in the hippocampus during aging[24,25], though we did not detect relevant changes as a result of enrichment. Regarding behaviour, actimetry revealed a general decrease in horizontal and vertical physical activity in aged animals in the second day session (Fig. S2a). Elevated plus maze (EPM) tests revealed significantly shorter times spent in open arms for enriched animals, a finding replicated in two laboratory-independent set-ups of 5- and 10-min exploration times (Fig. S2b-c). With respect to cognitive features, we implemented a novel object recognition (NOR) test followed by a novel object location (NOL) test in the same setting (using the same first pair of objects). We detected no effects of aging or EE in short-term (90 min) or long-term (24 h) NOR (Fig. S2d), nor in the associated NOL (26 h, Fig. S2e). Also in a laboratory-independent setting, an additional NOL was carried out (without prior NOR) with similar results (Fig. S2f). Finally, a contextual fear conditioning (CFC) test was performed which showed a non-significant trend for an effect of both age and enrichment on reaction to shock (Fig. S2g, left), though no differences in the test (Fig. S2g, right). Taken together, these observations indicate that our study system manifested the typical features of aging and the expected response to the EE paradigm in a number of structural and morphological parameters, together

with an enrichment-induced behavioural outcome consisting of increased anxiety and consequent absence of differences in cognition, frequent in novelty enrichment protocols using male C57BL/6 J mice.

To delve into the alteration of biological programmes in response to aging in the dorsal hippocampus, we profiled gene expression levels via RNA-seq, measuring a total of 19,835 genes with detectable expression across all samples (Methods). Principal component analysis (PCA) revealed how aging led to substantial transcriptomic changes across all subjects (Fig. 1b). For subsequent analyses, we made use of all the study samples in order to increase the statistical power (Methods). We performed differential expression analyses to uncover aging-associated expression signatures (FDR < 0.05, Methods, Supplementary Dataset 2), detecting 1038 upregulated and 1035 down-regulated differentially expressed genes (DEGs; these signatures were also validated using only the control samples, OR = 69.7, Fisher's $p < 0.001$, Fig. S3a). Up-regulated differentially expressed genes (DEGs) manifested more intense alterations (Wilcoxon $p < 0.001$, Fig. S3b) and were also strongly enriched in canonical aging genes in the GenAge[26] and Digital Ageing Atlas[27] databases (ORs = 20.3 and 17.9, all $p < 0.001$, Fig. S3c, d), while also agreeing with murine brain markers of aging recently described by Ximerakis and colleagues[28] (Fig. 1c), including well-known cognitive players such as *Apod*[29], *Neat1*[30] and *Il33*[31]. We performed gene set enrichment analyses across various databases (FDR < 0.05, Methods, Supplementary Dataset 3), including Gene Ontology (GO)[32], WikiPathways[33], Reactome[34], ImmuneSigDB[35] and cell type signatures (C8) and chemical and genetic perturbation pathways (CGP) from MSigDB[36] uncover axes of aging-associated dysregulation. We first noticed how, across all databases, up-regulation alterations were much more enriched in biological pathways in general (Fig. S3e), with two distinct core functions being targeted: a general activated inflammatory response, including a clear microglial response, while mRNA- and splicing-related pathways, involving neural cells, appeared down-regulated (Fig. 1d, Fig. S3f−i). Gene expression patterns are cell type-dependent; thus, to explore whether these responses could be associated with overall alterations in cellular composition within our study system, we made use of publicly-available single cell expression datasets to perform cellular deconvolution on our data (Methods) and found that there were no extensive changes in cell types during aging in our model across two independent datasets dissecting murine hippocampal and cortex tissue[37,38] (Fig. S4a), thus suggesting that the decrease in neurogenesis during aging does not necessarily lead to a decrease in the ratio of neural populations within the hippocampus, as has been discussed previously[39], nor does the transcriptomic up-regulation of the inflammatory response imply a relevant increase in microglial populations.

In order to deepen our understanding of the molecular functional response during aging, we used SWATH mass spectrometry[40] to characterize the proteome in our study system (Methods), quantifying a total of 2250 expressed proteins across all samples. In general, protein expression qualitatively agreed with RNA expression, with the detectable proteins consisting of a subset of highly expressed genes (Wilcoxon $p < 0.001$; Fig. S4b) which by themselves showed a modest quantitative correlation between the two omics (Spearman's coef. 0.33, $p < 0.001$; Fig. S4c), as has been reported previously[41]. Indeed, the normalization of protein measurements by their half-lives as reported by Mathieson and colleagues[42] greatly improved the correlations (Spearman's coef. 0.53, $p < 0.001$; Fig. S4d) and, moreover, a differential splicing analysis (FDR < 0.05, dIF ≥ 0.1, Methods) revealed aging-associated alteration of 86 isoforms in up to 71 genes (Supplementary Dataset 4, Fig. S4e), this mainly involving changes in the usage of alternative transcription initiation and termination sites, suggesting how multiple layers of biological regulation exist that separate RNA and protein expression, and that these technologies target complementary, but not equivalent, molecular layers[43]. Nonetheless, the protein landscape was able to differentiate, even more clearly, the

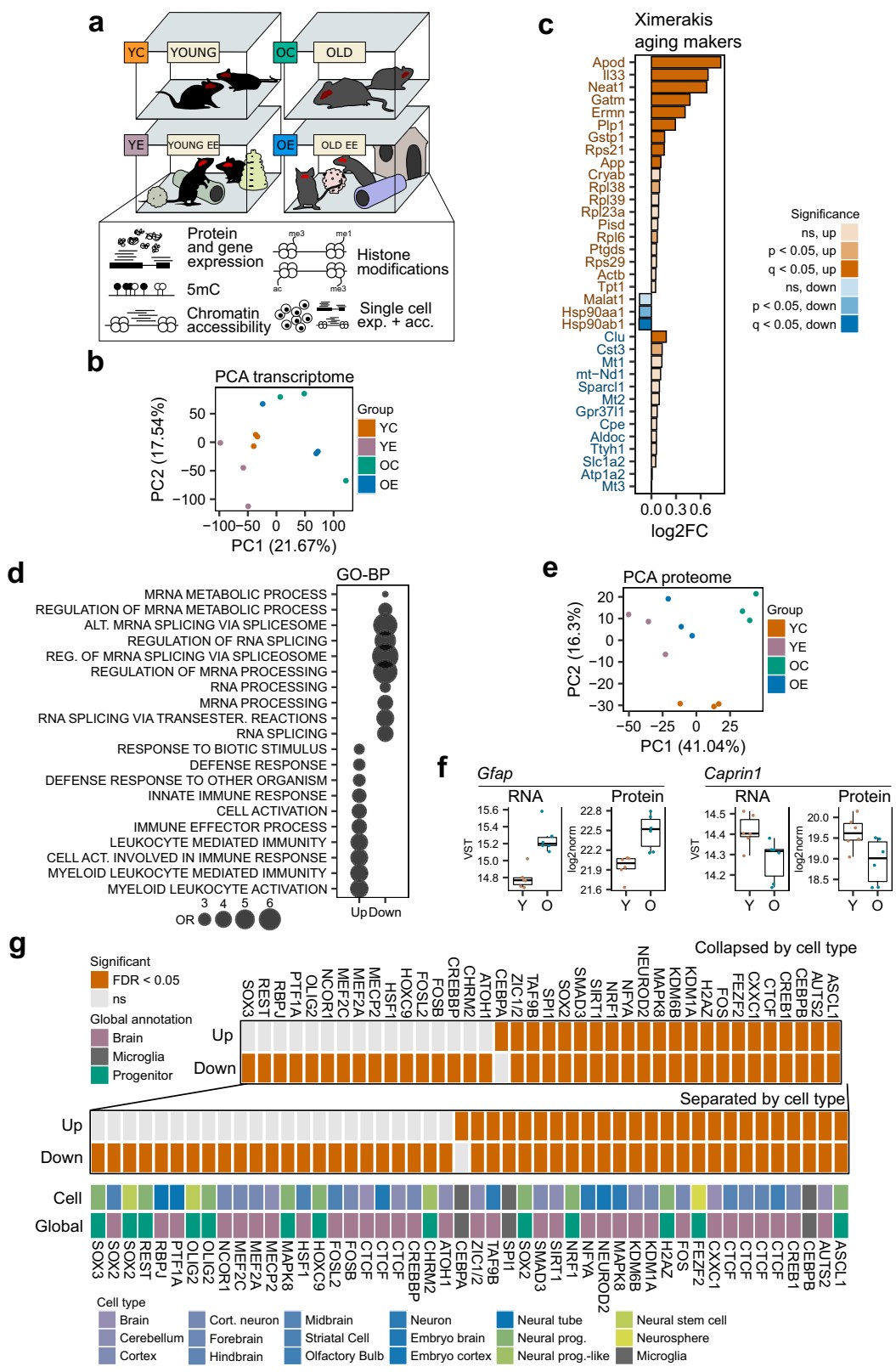

phenotypes through PCA (Fig. 1e), and abundant, bidirectional aging-associated alterations were also found (200 up- and 218 down-regulated proteins; FDR < 0.05, Methods, Supplementary Dataset 5; signatures again validated using only the control samples, OR = 13.7, Fisher's $p < 0.001$, Fig. S4f), with down-regulated differentially expressed proteins (DEPs) displaying stronger alterations in this case (Wilcoxon $p < 0.001$, Fig. S4g). In spite of the notable baseline

differences and the reduced space of common measurements between the omic layers (2227 genes), we were able to detect an over-enrichment in commonly upregulated DEG and DEP expression alterations, particularly for the most strongly altered genes (log2FC > 50th percentile; OR = 4.3, Fisher's $p = 0.001$, Fig. S4h). Genes with robust and coherent RNA and protein alterations (Supplementary Dataset 5) included established aging-upregulated brain markers such

**Fig. 1 | Transcriptomic and proteomic signatures of aging in the dorsal hippocampus. a** Schematic of the study design. **b** PCA plot of the transcriptomic profiles across samples and groups. **c** Bar plot describing the measured aging log2(fold change) across a panel of murine brain aging markers from Ximerakis et al. (2019). The colour of the gene labels indicates up-regulation (orange) or down-regulation (blue) in the original publication. Bar plots are coloured with regards to $p < 0.05$ or $p$ adj $< 0.05$ from two-sided Wald tests. **d** Bubble plot indicating the top 10 significant pathways (FDR < 0.05, one-sided Wallenius tests) found enriched for aging up- and down-regulated genes, in the Gene Ontology Biological Process database. The size of the bubbles indicates the odds ratio of enrichment. **e** PCA plot of the proteomic profiles across samples and groups. **f** Boxplots showing the expression measurements for the RNA-seq and SWATH-MS omic layers for the *Gfap* and *Caprin1* genes across young and old subjects. RNA expression is shown in variance stabilizing transformation (VST) values, and protein expression is shown in log2-normalized values. ($n = 6$ for all groups; $p$ adj $< 0.05$ across all data from two-sided Wald tests or moderated *t*-tests for RNA-seq and SWATH-MS data, respectively). All box plots shown indicate median value, interquartile range (IQR), up to 1.5 IQR (whiskers), and individual data points showing minimum and maximum value. **g** Heatmap showing significant (FDR < 0.05, LISA tests) regulators of aging-associated DEGs determined by LISA analysis using Cistrome brain-associated datasets.

as *Gfap*[44] and cognitive agents like *Caprin1*, which is necessary for long-term memory formation and healthy brain development[45,46], and which we found to be down-regulated with aging (Fig. 1f). Moreover, even though SWATH pathway enrichments provided far fewer results, the top detected pathways (unadjusted $p < 0.05$) also intersected with the top RNA-seq enriched pathways, particularly at the level of Gene Ontology, ImmuneSigDB and C8 cell type sets (ORs = 1.7-7.0, all Fisher's $p < 0.05$, Supplementary Dataset 6), thus indicating that aging-associated RNA expression signatures can also be captured at the protein level.

To gain insight into the possible transcriptional regulators of the alterations found, we performed an epigenetic Landscape In Silico deletion Analysis[47] (Lisa, Methods) using Cistrome DB[48] mouse datasets and detected many chromatin regulators associated with dysregulated aging DEGs (Supplementary Dataset 7), interestingly, with more functional associations being linked to gene down-regulation in this case (Fig. S4i). The Lisa analysis, focused on brain-related tissues (Fig. 1g), revealed the aging-associated bi-directional alteration of downstream targets of canonical neural factors involving differentiation and identity (e.g. ASCL1, FEZF2, NEUROD2, REST, OLIG2)[49–53] and, additionally, enrichments for a host of epigenetic modifiers covering diverse mechanisms, including histone modifiers (KDM1A, KMD6B, SIRT1)[54,55], methylation readers (CXXC1, MECP2)[56,57] and other players (H2AZ, CTCF)[58,59]. When looking at the full enrichments across any tissue, we observed an enrichment in macrophage-associated tracks for aging up-DEGs (OR = 2.0, Fisher's $p < 0.001$, Fig. S4j), in agreement with our previous observations of there being an important inflammatory response linked to aging upregulation involving macrophage-like cells. Moreover, 6 of these factors (KDM6B, NFYA, CEBPB, HSF1, CREB1, FOSL2) were detected as downregulated DEGs, which is considerably more than would be expected by chance (OR = 3.6, Fisher's $p = 0.01$), suggesting that at least some of the observed expression alterations may be driven by the aging-associated loss of chromatin regulators.

On the whole, these results reveal how aging induces transcriptomic and proteomic alterations in the dorsal hippocampus which are characterized by specific directional responses involving inflammatory activation and neural splicing-related repression, some of which may be linked to the aging-associated dysregulation of epigenome modifiers.

**Functional remodelling during aging can be partially linked to methylomic alterations and changes in chromatin accessibility**

In light of the aforementioned findings, we set out to map multiple epigenomic layers in our study system, in order to deepen our understanding of the mechanistic basis of our observations. We first profiled genome-wide methylation using Enzymatic Methyl sequencing[60] (EM-seq, Methods), characterizing a total of 18,201,112 CpG sites (≥10 coverage) across all samples. Global CpG methylation was measured at 76%, with no relevant differences across groups (Fig. 2a), nor did we observe particular changes in non-CpG methylation (Fig. S5a, b) or global cytosine methylation levels as measured by LC-MS/MS in an independent set of samples (Fig. S5c), indicating that

there is no pronounced loss of DNA methylation with aging in our study system, as has been observed across several tissues using sequencing technologies[61,62]. These similarities were also maintained across genomic elements such as CpG islands and gene locations (Fig. S5d). We next performed a differential methylation analysis (FDR < 0.05, Methods, Supplementary Dataset 8), uncovering 237 aging-associated differentially methylated regions (DMRs), 148 of which suffered loss of methylation and tended to be bigger and less dense in CpGs, and have stronger alterations, than their hypermethylated counterparts (Fig. S5e-g). Aging-DMRs were enriched at CpG island and promoter locations (Fig. 2b, c; ORs = 3.8–16.4, all Fisher's $p < 0.001$) and also at genes detected as expressed in RNA-seq by permutation testing (Fold Enrichment, FE = 2.4, empirical $p < 0.001$), hinting at their putative functionality. Interestingly, when we integrated these regions with our previously defined gene expression alterations, we found 17 genes (Supplementary Dataset 8) that displayed alterations in both omics, including important neural regulators such as *Cbln1*[63] and Parkinson's-associated *Pink1*[64], both of which showed hypomethylation associated with CpG island or enhancer elements coupled to an increase in gene expression (Fig. 2d), as well as other neurodevelopment targets like *Myt1l*[65] and methylation-expression alterations in inflammatory response genes such as *Irf8*[66], among others. Gene pathway enrichments in the DMRs were minor, with the particular exception of there being a strong enrichment in hyper- and hypomethylation of murine imprinted genes collated in the CGP MSigDB database[67] (FDR < 0.05, ORs = 23.2-41.6, Supplementary Dataset 9). Interestingly, the *Tle3* gene, which has been associated with imprinting phenomena[68] was among the genes with parallel aging DNA methylation and gene expression changes (Fig. S5h, i).

To investigate if the reported observations could be related to aging-associated alterations in chromatin accessibility, we mapped genome-wide accessibility using Assay for Transposase-Accessible Chromatin sequencing[69] (ATAC-seq, Methods). We detected accessible regions with substantial enrichment (FRiP scores 59% to 71%) and did not observe any noticeable trend of a general increase or decrease in accessibility between young and old samples (Fig. S5j). As expected, regions of accessible chromatin were enriched at promoter and CpG island locations (Fig. S5k, l; ORs = 9.8–43.2, all Fisher's $p < 0.001$) and displayed similar characteristics across groups (Fig. S5m). Additionally, accessibility showed a strong qualitative correlation (Wilcoxon $p < 0.001$, Fig. S5n) and moderate quantitative correlation with the gene expression data (Spearman's coef. 0.62 and 0.50, both $p < 0.001$, for correlation across gene bodies or promoters, respectively; Fig. S5o). We performed a differential accessibility analysis (FDR < 0.05, Methods, Supplementary Dataset 10) and discovered 46 differentially accessible regions (DARs), most of which (65%) presented an increase in accessibility with aging and were predominantly found at open sea and intergenic regions (ORs = 3.3, 2.4, Fisher's $p = 0.057$, 0.016, respectively), while loss of accessibility was linked to CpG islands (OR = 3.9, Fisher's $p < 0.01$) (Fig. 2e, f). Interestingly, 6 of the aging-DARs intersected with aging-DEGs, 5 of them displaying increased promoter accessibility coupled to gene upregulation, an

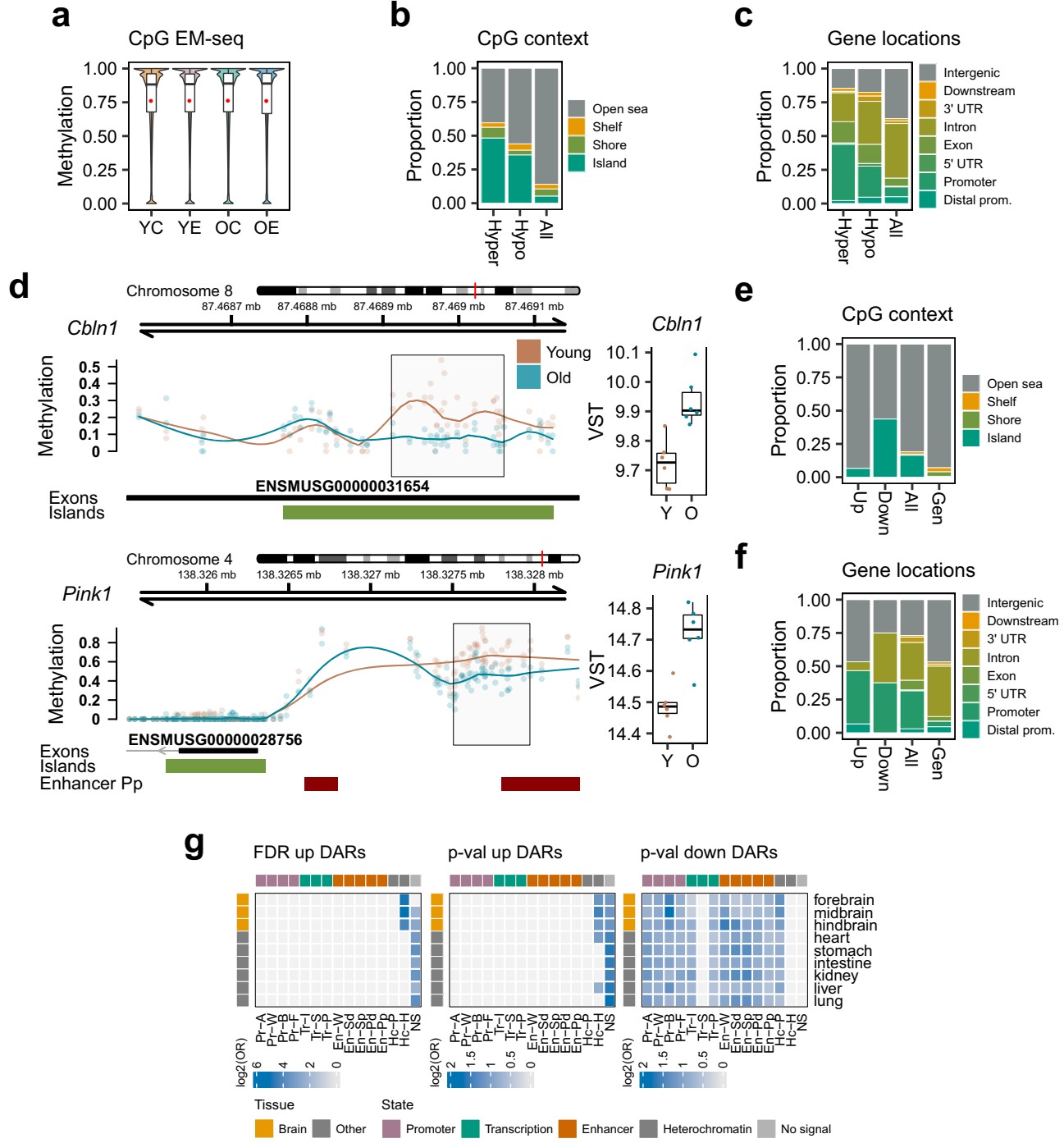

**Fig. 2 | Methylomic and chromatin accessibility signatures of aging in the dorsal hippocampus. a** Violin plots showing the distribution of EM-seq methylation measurements of CpG sites across groups. Plots are based on a 1 M sample of CpGs. **b, c** Bar plots describing the proportion of hyper- and hypomethylated aging-DMRs mapped to CpG island (**b**) or gene (**c**) locations, as compared to the distribution of all of the profiled CpG sites. **d** Genomic plots showing the DNA methylation profiling values in young and old subjects of two regions containing aging-DMRs (grey boxes) associated with the *Cbln1* and *Pink1* genes. Below, the tracks indicate the presence of gene exons, CpG islands and ENCODE murine forebrain P0 enhancer Pp elements. On the right of the plots, the boxplots indicate the RNA-seq expression measurements (VST units) for the *Cbln1* and *Pink1* genes across young and old subjects (*n* = 6 for all groups; *p* adj <0.05 for two-sided Wald tests across all data). All box plots shown indicate median value, interquartile range (IQR), up to 1.5 IQR (whiskers), and individual data points showing minimum and maximum value. **e, f** Bar plots describing the proportion of up- and down- aging-DARs mapped to CpG islands (**e**) or gene (**f**) locations, as compared to the distribution of the consensus peak set ("All") or the whole genome ("Gen", 200 bp bins). **g** Heatmaps showing the significant (FDR < 0.05, one-sided Fisher's exact tests) LOLA enrichments in log2(odds ratio) of chromatin states associated with aging-DARs, either up-DARs defined with FDR < 0.05 (two-sided Wald tests) or up- and down-DARs defined with unadjusted *p* < 0.05 (two-sided Wald tests). The code for the ENCODE chromatin states shown is: Promoter, Active (Pr-A), Weak (Pr-W), Bivalent (Pr-B) and Flanking (Pr-F); Transcription, Strong (Tr-S), Permissive (Tr-P) and Initiation (Tr-I); Enhancer, Strong TSS-distal (En-Sd), Strong TSS-proximal (En-Sp), Weak (En-W), Poised TSS-distal (En-Pd) and Poised TSS-proximal (En-Pp); Heterochromatin, Polycomb-associated (Hc-P) and H3K9me3-associated (Hc-H); No significant signal (NS).

intersection higher than expected by chance (FE = 3.1, empirical $p < 0.01$, Supplementary Dataset 10), including inflammation-related $C4b$[70] and neural-related $Lin28b$[71]. Moreover, the top DARs (unadjusted $p < 0.05$) were also enriched at significant methylation DMRs (FE = 6.3, empirical $p < 0.001$), (Supplementary Dataset 10), suggesting an interplay between the different omic layers. To study the possible impact of chromatin accessibility changes on the epigenomic structure, we performed Locus Overlap Analysis (LOLA, FDR < 0.05, Methods) on the aging-DARs (Supplementary Dataset 11) using public ENCODE3 datasets[72] and discovered that gains in accessibility preferentially occurred at heterochromatic states (Fig. 2g), while loss of accessibility was more subtly associated with active states (enhancer, promoter) or Polycomb locations (signatures recovered with unadjusted $p < 0.05$ DARs). On the other hand, and even though the two omic layers have some degree of anti-correlation (Fig. Fig. S5p), EM-seq DMRs were associated with different chromatin signatures, though we did validate that aging DNA hypermethylation was enriched at Polycomb loci when compared to hypomethylation (Fig. S5q, Supplementary Dataset 12), a well-known phenomenon occurring during the aging process[73].

Together, these observations highlight how functional aging-associated changes can be linked to epigenomic features such as DNA methylation alterations at regulatory loci, and that chromatin compaction alterations occurring during aging possess differential signatures whereby increases in accessibility appear linked to heterochromatic states while condensation occurs at more active and functionally-defined regions.

### The chromatin landscape of aging in the dorsal hippocampus reveals widespread heterochromatin reconfiguration

We next comprehensively profiled the chromatin landscape in our study system by characterizing the genome-wide levels of a well-known set of histone post-translational modifications[72] (Methods): H3K4me1, H3K4me3, H3K27ac, H3K36me3, H3K27me3 and H3K9me3. The distribution of histone signal was associated with the expected functional elements (e.g. enhancers, promoters, heterochromatin) for each histone mark (Fig. S6a, b) and we determined regions of considerable enrichment for all modifications, with various degrees of concentration depending on the sparsity of each mark (FRiP scores 21% to 78%, Fig. S6c, d). The epigenome-wide profiles, via PCA, anticipated the histone-specific effect of aging on the histone post-translational landscape (Fig. 3a), and, indeed, a differential enrichment analysis using all samples (FDR < 0.05, Methods, Supplementary Dataset 13) uncovered widespread histone-specific reconfiguration during aging: global heterochromatin alterations were found at thousands of facultative heterochromatin loci (H3K27me3) and hundreds of constitutive heterochromatin blocks (H3K9me3), while local, less numerous changes were observed for the rest of the marks (Fig. S6e). In this scenario, we validated in our study system the well-known loss of constitutive heterochromatin[14] while observing bidirectional alterations in facultative heterochromatin.

Focusing on these differentially enriched regions (DERs), we first studied the interaction between histone mark alterations and changes in chromatin accessibility via permutation sampling (Fig. 3b, empirical FDR < 0.05), confirming that changes in active marks (H3K4me1/3, H3K27ac, H3K36me3) generally correlated positively with changes in accessibility. This was different to the repressive modifications, where constitutive H3K9me3 loss was strongly associated with chromatin decompaction while facultative H3K27me3 gains were linked to chromatin condensation, though this latter mark also appeared to play more of a dual role with respect to chromatin accessibility (Fig. 3b). Importantly, these observations indicate that the loss of H3K9me3 domains can be linked to chromatin decompaction during aging, as has been previously suggested[74], while facultative H3K27me3 loss may not necessarily lead to the same molecular effect.

These findings led us to also explore the interactions between aging-associated alterations in each epigenomic modification, using our previous strategy (Fig. 3c). First, we confirmed that alterations in active modifications were largely correlated between themselves and anti-correlated with repressive marks, with some subtle exceptions such as the loss of H3K4me3 sometimes coinciding with loss of H3K27me3. This latter observation could imply the aging-associated erasure of bivalent chromatin domains, which, in the context of cancer and regarding H3K4me3, has been posited to sensitize genes to additional epigenomic dysregulation[75]. To confirm this observation, we explored the histone modification dynamics at bivalent loci (Methods) and observed that the loss of both marks was linked to bivalent chromatin domains (ORs = 3.5–38.9, all Fisher's $p < 0.001$, Fig. S6f), with H3K27me3, in particular, showing a very distinct pattern where the majority of DERs associated with bivalent regions lost the modification, while, on the other hand, most non-bivalent loci tended to gain facultative H3K27me3 with aging (Fig. 3d). Additionally, we observed a subtle trend whereby DERs of H3K4me3 loss at bivalent domains were associated with EM-seq DNA methylation gains, and viceversa, with H3K27me3 playing a more ambivalent role (FDR < 0.05, Fig. S6g).

On the other hand, the relationship between heterochromatin modifications revealed more complex, and striking, associations (Fig. 3c): in addition to the expected positive correlation between aging-associated increases and decreases of H3K27me3 and H3K9me3, we also discovered an enrichment of H3K27me3 gains at regions of H3K9me3 loss. These 154 sizable domains (mean length ~37,000 bp) of chromatin exchange (Fig. 3e, Supplementary Dataset 14) were often characterized by being grouped in hotspots of facultative H3K27me3 increases coupled to broad constitutive H3K9me3 loss (Fig. 3f). These regions were enriched at gene locations when compared with the background of H3K9me3 peaks (OR = 3.1, Fisher's $p < 0.001$), showing a distribution similar to the rest of the H3K27me3 peaks, and genes associated with the loci were enriched in pathways related to G-protein associated perception of sensory stimuli and chemokines, as well as Alzheimer's and cancer pathways (Supplementary Dataset 15). Our observations suggest that these domains may represent a chromatin switch activated upon aging-associated constitutive H3K9me3 loss to re-repress these altered regions using a parallel mechanism based on H3K27me3 deposition. Indeed, switching of these epigenetic marks has been described in the context of experimental models involving transposon repression[76] and also in models of heterochromatin loss by mutation[77,78]. Additionally, though subtly, 78% of the switching regions presented increased EM-seq DNA methylation levels with aging (Fig. S6h), which might be expected because of the well-known interaction between Polycomb components and DNA methylation deposition[79].

With regards to the functional impact of the observed chromatin modifications, we confirmed that DERs across all histone marks were significantly enriched in RNA-seq DEGs (FDR < 0.05, Fig. S7a), highlighting how aging gene expression alterations are accompanied by epigenome-wide reconfiguration. These associations followed the expected trends: changes in active histone modifications were positively correlated with gene expression alterations, while loss of heterochromatin marks was linked to gene up-regulation (Fig. S7a). To characterize the nature of the chromatin-impacted genes, we performed gene set enrichment analyses (FDR < 0.05) with the genes associated with the top DERs (unadjusted $p < 0.05$) for each histone mark (Supplementary Dataset 16), revealing various functional relationships. First, regions of H3K4me3, H3K27me3 and H3K9me3 loss with aging were more robustly associated with molecular pathways than domains experiencing gains in these marks, with opposite trends being observed for the other modifications, and H3K36me3 showing the lowest number of enrichments (Fig. S7b). Regarding the specific pathways involved, active marks generally showed parallelisms

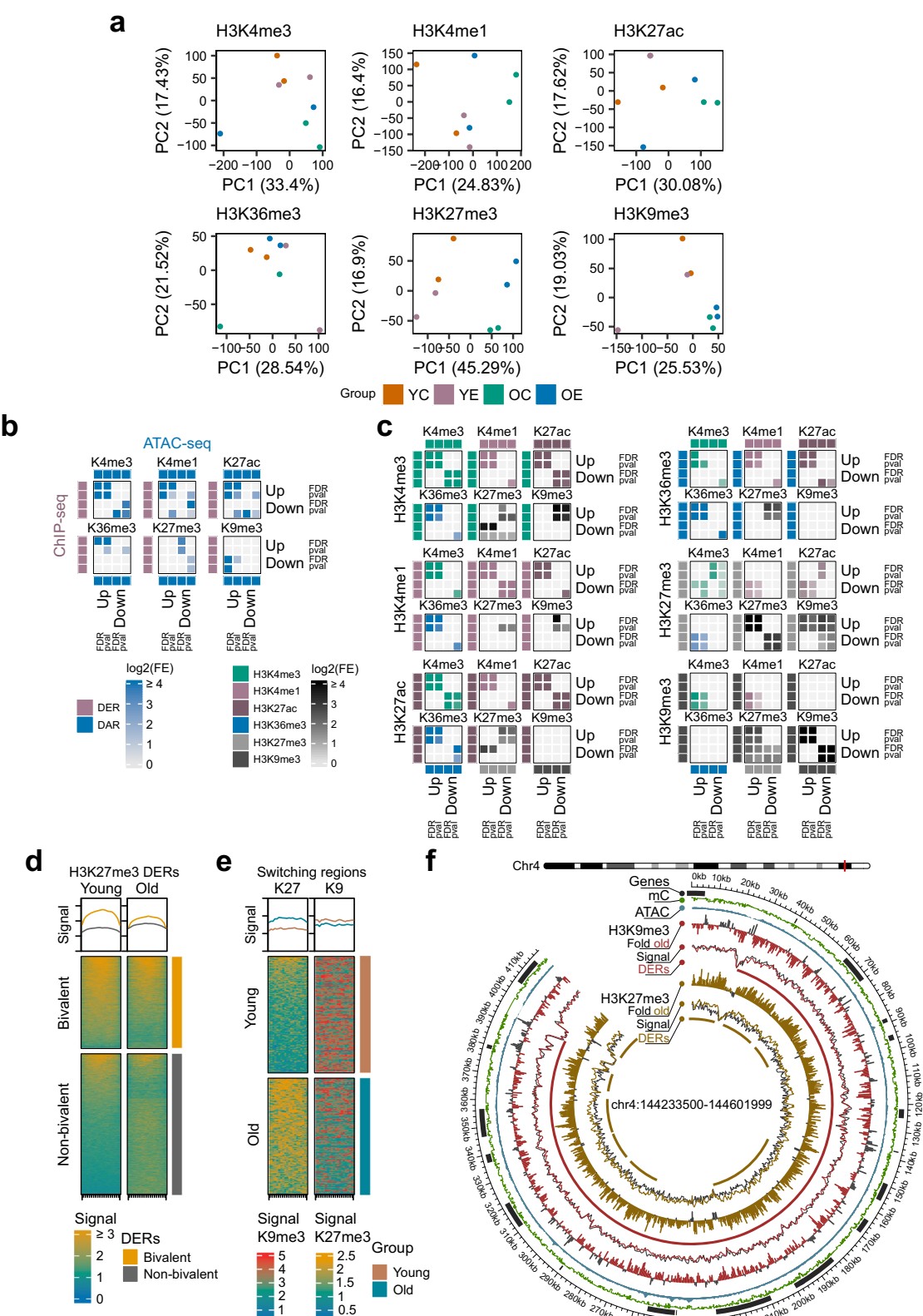

involving gains at developmental genes and TF regulatory sites (GO, CGP) and losses at neural pathways, particularly synaptic signalling and neural cell identity genes (GO, C8). On the other hand, repressive marks showed gains subtly associated with G protein and surface receptors (GO) and marked losses at neural differentiation, transcription regulation and chromatin organization (GO) (Fig. S7c, Supplementary Dataset 16). Of note is the fact that the H3K4me3, H3K27me3

and H3K9me3 modifications all showed strong enrichments in depleted DERs at Polycomb associated loci.

All of the aforementioned interactions highlight how the epigenome is a dynamic system in which aging-associated alterations of chromatin marks can drive parallel or opposite responses in other modifications. To expand on this concept, we integrated our epigenome-wide data to discover chromatin state segmentations

**Fig. 3 | Chromatin dysregulation of the dorsal hippocampus during aging.**
**a** PCA plots of the epigenomic profiles involving the levels of H3K4me3, H3K4me1, H3K27ac, H3K36me3, H3K27me3 and H3K9me3 across samples and groups. Histone signal is quantified across a consensus peak set for each mark. **b** Heatmaps showing the log2-fold enrichment of significant intersections (FDR < 0.05 within each set for one-sided permutation regioneR tests) between ChIP-seq aging DERs and ATAC-seq aging DARs, filtered either at FDR < 0.05 or $p$-value < 0.05 (from two-sided Wald tests). **c** Heatmaps showing the log2-fold enrichment of significant intersections (FDR < 0.05 within each set for one-sided permutation regioneR tests) between the ChIP-seq aging DERs. DERs are filtered either at FDR < 0.05 or $p$-value < 0.05 (from two-sided Wald tests). **d** Heatmaps indicating the levels of H3K27me3 signal at aging DERs (FDR < 0.05, two-sided Wald tests) in young (left) and old (right) samples. The regions are grouped into those with overlapping bivalent domains (top) or not (bottom). Histone signal is represented with BigWig RPGC-normalized values (10 bp bins). **e** Heatmaps indicating the levels of H3K27me3 (left) and H3K9me3 (right) signal at aging chromatin switching regions, defined as the intersection between aging H3K27me3 up-DERs (FDR < 0.05, two-sided Wald tests) and aging H3K9me3 down-DERs (FDR < 0.05, two-sided Wald tests). The regions are represented for young (top) and old (bottom) samples. Histone signal is represented with BigWig RPGC-normalized values (10 bp bins). **f** Circos plot describing the distribution of various epigenomic marks across an aging heterochromatin switching genomic region (chr4:144233500-144601999) where global loss of H3K9me3 is coupled with a hotspot of H3K27me3 gains during aging. The "fold" tracks represent the difference in log-CPM values (500 bp bins) between old and young samples, where aging decreases in H3K9me3 and increases in H3K27me3 are coloured, while the opposite trend is coloured in grey.

using ChromHMM[80] (Methods). We learned 15 chromatin states in our data, which we classified into 4 promoter states (Pr-A, Pr-W, Pr-F), 4 transcription states (Tr-S, Tr-S2, Tr-P, Tr-I), 3 enhancer states (En-Sd, En-Sp, En-Pd), 3 heterochromatin states (Hc-P, Hc-Pw, Hc-H) and 1 non-signal (NS) state (Fig. S7d, Supplementary Dataset 17). Our biological annotation was highly consistent with that defined for mouse forebrain in recent ENCODE3 datasets[72] (Fig. S7e), demonstrating how the functional identity of chromatin states is robust and can be recovered even when epigenome-wide profiling does not target the same chromatin modifications[81]. Using our segmentations, we observed how the previously discovered regions of aging-associated heterochromatin switching present an increase in Polycomb-associated chromatin states in old samples (Fig. S7f).

All in all, the results presented in this section provide an extensive characterization of the chromatin reconfiguration which occurs during aging in the murine dorsal hippocampus. This phenomenon specifically involves the re-arrangement of heterochromatin modifications, with bidirectional changes in facultative H3K27me3 and a general loss of constitutive H3K9me3. Importantly, we link the paired loss of H3K27me3 and H3K4me3 to the erosion of bivalent domains and, strikingly, we uncover a heterochromatin switch whereby regions of aging-associated constitutive heterochromatin loss present an enrichment in gains of facultative heterochromatin, suggesting a molecular mechanism of chromatin re-repression during the process of aging.

## Molecular rejuvenation of the dorsal hippocampus in enriched environments

Up to this point, we had painted a deep, multi-omic landscape of the process of aging in the murine dorsal hippocampus, identifying multiple dimensions across which aging can lead to molecular changes. Thus, we next set out to explore the putative effects, at the molecular level, of a lifestyle intervention based on the EE paradigm in our study system (Methods).

First, through a differential expression analysis (Methods), we identified 94 RNA-seq DEGs (FDR < 0.05) associated with EE across all samples, with 60 and 36 DEGs, respectively, being found for young and for old samples (Fig. 4a, Supplementary Dataset 18). The trends of the changes across DEGs were mostly consistent between young and old samples, though a small number of changes displayed opposing directions depending on age, suggesting the existence of age-specific effects (Fig. S8a). Furthermore, our EE transcriptomic signatures clearly matched those recently described by Wassouf and colleagues[82], providing validation of our experimental model (Fig. 4b). This validation is particularly relevant in the context of enrichment in mouse models and because our system used male animals. While males are very responsive to enrichment in laboratory settings, both the behavioural and the adult neurogenic niche outcomes can show an absence of differences or even increased anxiety (as in the present work), because of the increased eustress induced by some types of enrichment environments in some mouse strains[83]. We performed gene set

enrichment analyses (FDR < 0.05) on the EE-associated signatures (top genes with unadjusted $p < 0.05$, Supplementary Dataset 19). Across all subjects, and particularly for young individuals, EE-associated down-regulation appeared to play more functional roles (Fig. S8b): while EE up-regulation was linked to NGF, TFG and NTRK signalling pathways (Reactome, CGP), down-regulation was associated with neural functions and glial differentiation, including synaptic and myelin sheath pathways (GO, Fig. 4c), neural and oligodendrocyte cell types (C8, CGP) and also, very interestingly, to markers upregulated in the aging brain (CGP) (Supplementary Dataset 19). These results suggest that EE influences hippocampal function at both the neural and glial level, and that EE-induced changes may interact with aging alterations.

We thus turned to examining the intersections between aging- and EE-associated gene expression changes (FDR < 0.05) and, remarkably, observed that practically all of the common alterations displayed opposite directions of change in the two processes (Fig. 4d), with the intersections being extremely enriched (ORs = 11.7 and 14.7, Fisher's $p < 0.001$), a finding which held for the top signatures (unadjusted $p < 0.05$, Fig. S8c), suggesting a transcriptome-level phenomenon. Indeed, when we looked at the direction of change in response to EE of all the previously defined aging-DEGs, we observed the EE-associated reversion of the majority of the aging changes (Wilcoxon all $p < 0.001$, Fig. 4e), which occurred in both young and old samples (Fig. S8d). To validate this major finding, we profiled the transcriptome of an independent set of samples including young and old subjects exposed to EE. Again, and particularly regarding EE-down-regulation, we demonstrated the reversal of aging-associated alterations (Fig. S8e, f). Expanding on this functional characterization, a differential expression analysis of the proteomic data revealed a widespread reconfiguration of the proteome in response to EE (Fig. 4f, Supplementary Dataset 20), with the responses again being similar across both age groups (Fig. S8g). Significantly, when looking at the intersections between aging- and EE-DEPs (Fig. 4g), we again discovered the strong reversal of aging-associated alterations by EE (ORs = 5.3 and 12.5, Fisher's $p < 0.001$), with the great majority of aging-DEPs showing a reversed direction of change with those from EE (Wilcoxon all $p < 0.001$, Fig. 4h) for both young and old samples (Fig. S8h). This latter observation is of particular interest because, as previously mentioned, there is not a complete correlation between transcriptomic and proteomic levels, such that these two molecular layers appear to undergo clear, but partly independent, aging-reversal effects in response to environmental enrichment.

Considering the aforementioned results, we defined EE "reversed" changes as those molecular changes that went in the opposite direction of aging-associated alterations in both young and old individuals, while we considered "rejuvenation" changes as those which were reversed specifically in the old group and did not show alterations in young subjects. Thus, we curated a list of 163 genes displaying the EE-associated reversal of aging expression alterations (Fig. 4i, j, Methods, Supplementary Dataset 21), most of which suffered age-independent effects across all samples, with an independent selection ($N = 353$)

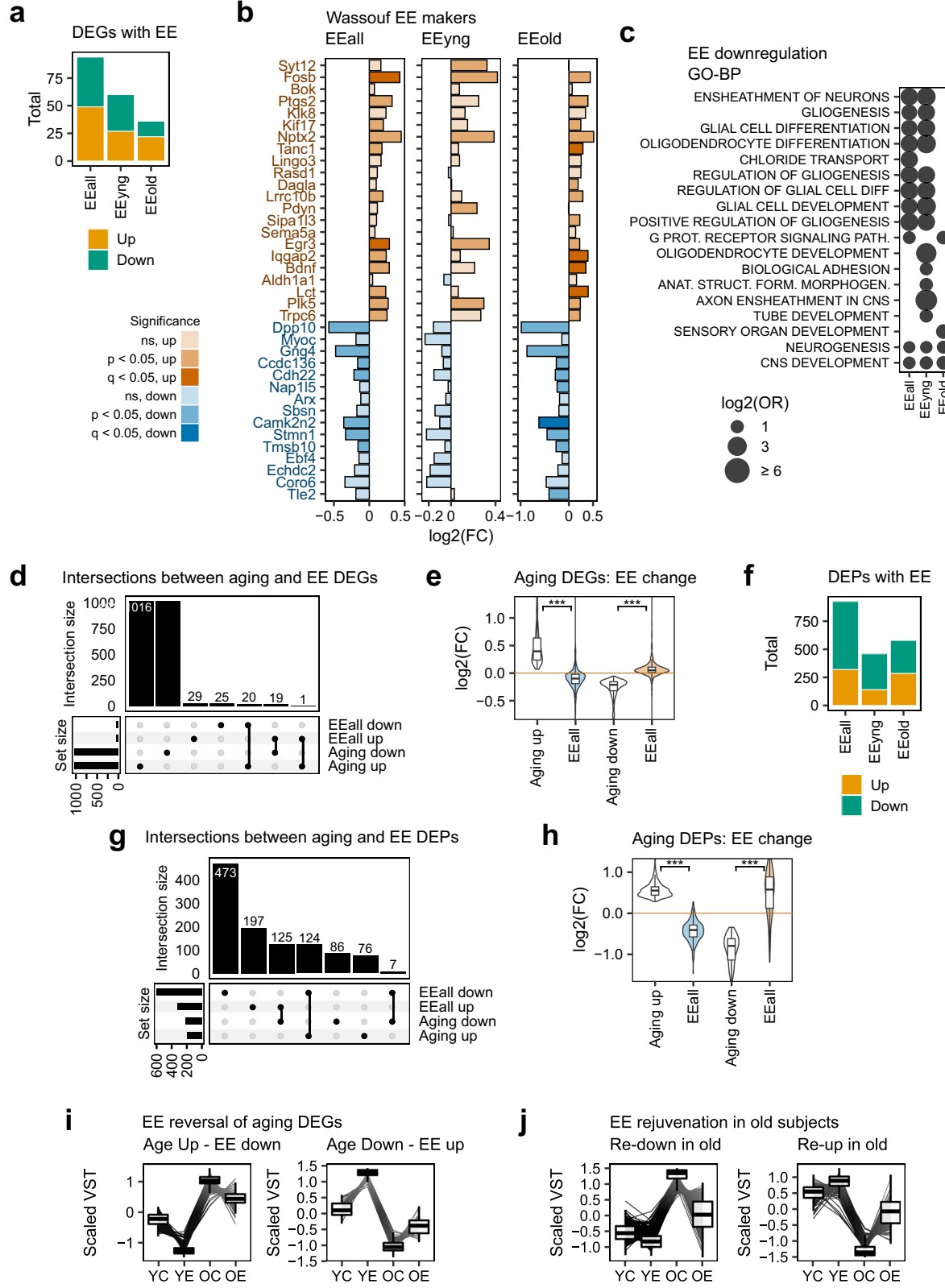

displaying rejuvenation specifically in old samples (Fig. 4i, j). We also defined similarly reversed and rejuvenated genes at the protein level, and, for a selection of 48 genes with subtler changes, we demonstrated RNA- and protein-level validation (Fig. S8i, Supplementary Dataset 21). At the level of gene expression, the effects of EE reversal were stronger in young subjects when compared with their aged counterparts (Wilcoxon both $p < 0.001$, Fig. S8j), suggesting that these individuals could

be more plastic or sensitive to EE stimulation, a finding in agreement with the well-known loss of plasticity occurring during brain aging[84]. Intriguingly, this was not the case for protein changes (Wilcoxon both $p > 0.05$, Fig. S8k), indicating that, in terms of protein regulation, old subjects could retain responses more comparable to young individuals. Gene set analyses (FDR < 0.05) for the transcriptomic reversal genes revealed noticeable enrichments in oligodendrocyte markers

**Fig. 4 | Functional rejuvenation of the dorsal hippocampus in enriched environments. a** Bar plots indicating the total number of DEGs (FDR < 0.05, two-sided Wald tests) with increased or decreased levels in response to EE in young, old or all samples. **b** Bar plots describing the measured EE log2(fold change) across a panel of EE-associated DEG markers from Wassouf et al. (2018). Each column indicates the change for the effect of EE on all, young or old samples (two-sided Wald tests). The colour of the gene labels indicates up-regulation (orange) or down-regulation (blue) in the original publication. **c** Bubble plot showing the top 10 significant pathways (FDR < 0.05, one-sided Wallenius tests) found enriched for the top EE down-regulated DEGs (unadjusted $p < 0.05$, two-sided Wald tests) in the Gene Ontology Biological Process database. The size of the bubbles indicates the log2(odds ratio) of enrichment. **d** UpSet plot describing the intersections between aging and EEall DEGs (FDR < 0.05, two-sided Wald tests). **e** Violin plots showing the log2(fold change) values for the aging up- and down-DEGs (FDR < 0.05, two-sided Wald tests) and also, for the same sets of genes, the fold change values with EE

(***$p < 2.2e\text{-}16$ for two-sided Wilcoxon rank sum tests). Box plots shown indicate median value, interquartile range (IQR), up to 1.5 IQR (whiskers), and have been cropped at their 2nd and 98th percentiles. **f** Bar plots indicating the total numbers of DEPs (FDR < 0.05, two-sided moderated $t$-tests) with increased or decreased levels in response to EE in young, old or all samples. **g** UpSet plot showing the intersections between aging and EEall DEPs (FDR < 0.05, two-sided moderated $t$-tests). **h** Violin plots describing the log2(fold change) values for the aging up- and down-DEPs (FDR < 0.05, two-sided moderated $t$-tests) and also, for the same sets of genes, the fold change values with EE (***$p < 2.2e\text{-}16$ for two-sided Wilcoxon rank sum tests). Box plots as described in (**e**). **i, j** Line plots showing the RNA-seq gene expression values (scaled VST units) of curated genes which show opposing aging and EE changes (**i**) or rejuvenation specifically in old subjects (**j**). Box plots shown indicate median value, interquartile range (IQR) and up to 1.5 IQR (whiskers), and individual data lines showing minimum and maximum value.

and neural and myelin pathways (C8, CGP, GO), with age-up-EE-down genes showing the main functional associations and no enrichments for inflammatory pathways (ImmuneSigDB) (Fig. S8l, Supplementary Dataset 22), while the specific selection of old-rejuvenation genes showed scant enrichments in general. Interestingly, reversal genes were also enriched in bivalent/Polycomb chromatin-associated brain pathways (Fig. S8l, bottom). Significantly, these observations suggest that, even though there is a major inflammatory component in aging up-regulation alterations (see previous Fig. 1d), the EE-associated reversal of aging signatures appears to mostly target brain-specific pathways involving both neurons and glial cells, along with bivalent chromatin locations, which are known to be involved in aging-associated epigenetic dysregulation[73].

Next, we sought to explore whether the observed phenotype-level (RNA and protein) reversal of aging alterations could be traced back to epigenomic dynamics. Focusing on the chromatin landscape, we performed differential analyses and observed minor changes in response to EE in both young and old samples (Supplementary Dataset 23). As with the transcriptomic observations, it is not surprising that a modest, non-pharmacological intervention such as environmental enrichment did not lead to the drastic chromatin remodelling observed during aging. We selected the top (unadjusted $p < 0.05$) EE-associated chromatin DERs and performed intersection testing to investigate if aging- and EE-associated chromatin changes could target common genomic loci. Notably, we did indeed detect enrichments in overlaps between aging- and EE-changes: for all the active histone modifications, aging-DERs were enriched (Fisher's FDR < 0.05) in overlaps with EE-DERs with the opposite direction of change (Fig. 5a, b). This unbiased approach suggests that environmental enrichment also generates a chromatin response which tends to counteract aging-associated dysregulation. Because intersecting aging- and EE-opposite regions often showed stronger reversal effects within old samples, we curated a list of histone modification peaks that showed rejuvenation in old samples (FDR < 0.05, Methods, Supplementary Dataset 24). The majority of rejuvenated DERs corresponded to H3K27me3 changes (Fig. 5c), in line with this modification being the one that was most altered during the aging process (see previous Fig. S6e), and most of them involved changes where EE raised the modification back to young levels. We intersected these reju-DERs with the previously defined RNA-seq reju-DEGs, finding 28 epigenome-rejuvenation regions overlapping 21 rejuvenated genes (Supplementary Dataset 24). Significantly, 27 of these regions (96%) showed changes consistent with the observed gene expression trends whereby the rejuvenation of active modifications positively correlated with expression, while the rejuvenation of H3K27me3 was accompanied by a gene expression reversal in the opposite direction, and, moreover, permutation testing (empirical FDR < 0.05) revealed significant enrichment of the histone reju-DERs in reju-DEGs, though the number of intersections was low (Fig. 5d). These genes included several candidates of interest, such as Protocadherin 15

(*Pcdh15*), associated with oligodendrocyte progenitors and the in vitro suppression of their proliferation[85], which we found suffered an age-dependent increase in gene expression coupled to a reduction of H3K27me3 levels which was reversed with EE in old individuals (Fig. 5e), along with the Receptor Tyrosine Kinase Like Orphan Receptor 1 gene (*Ror1*), which is a synapse-formation- and neural progenitor regulator[86,87] found decreased in cellular models of Alzheimer's[88], for which we observed the EE-associated recuperation of its expression and opposing H3K27me3 levels (Fig. 5f). Other relevant examples involve *Lrrc10b*, which has paired expression, H3K27ac and H3K4me3 rejuvenation and is known to be dysregulated in mouse models of addictive behaviour[89], neural-related *Pbx3*[90], dentrite trafficking-associated *Sorcs1*[91] and the glutamate-like receptor *Grid2*[92], among others (Supplementary Dataset 24).

Collectively, these observations indicate that a lifestyle intervention based on the environmental enrichment paradigm leads to functional transcriptomic and proteomic changes in the hippocampus which largely tend to oppose or revert those alterations accumulated during aging, and that this rejuvenating effect can also be revealed at the level of epigenomic dynamics, providing mechanistic explanations of how an environmental stimulus can lead to molecular changes in a higher-order biological system.

## The single-cell dynamics of aging and environmental enrichment in the dorsal hippocampus

Because the central nervous system is an extremely complex structure, we turned our attention to exploring the regulatory dynamics of our experimental system at the single-cell level by characterizing transcriptomic and genome-wide accessibility levels (Methods). We profiled a total of 16,136 high-quality cells with paired RNA-seq and ATAC-seq measurements and took advantage of the multimodal data to perform joint clustering using Seurat's weighted-nearest neighbour (WNN) strategy[93], resulting in a total of 26 clusters (Fig. S9a). Then, we made use of two recent single-cell atlases to annotate our cell clusters (Methods): a map of the aging mouse brain by Ximerakis and colleagues[28] (37,069 cells) and a characterization of the mouse iso-cortex and hippocampal formation by Yao and colleagues[38] (73,347 cells). These two datasets provided us with complementary information and permitted the robust annotation of our cell clusters at both the functional and brain-region levels, with a high level of agreement between the two (adjusted rand index, ARI = 0.97 for the annotation in major cell types between the two atlases, Fig. S9b, c, Methods). Moreover, our clusters showed an average purity in cell types of 95%, indicating that our single-cell data were able to efficiently discriminate between different brain cell types. After curation, we distinguished 15 major cell type clusters in our data (Fig. 6a; NEU: neuron; OLG: oligodendrocyte; OPC: oligodendrocyte progenitor; ASC: astrocyte; MIG: microglia; END: endothelial cell; PER: pericyte; VLM: vascular and leptomeningeal cell; see Supplementary Dataset 25 for extended

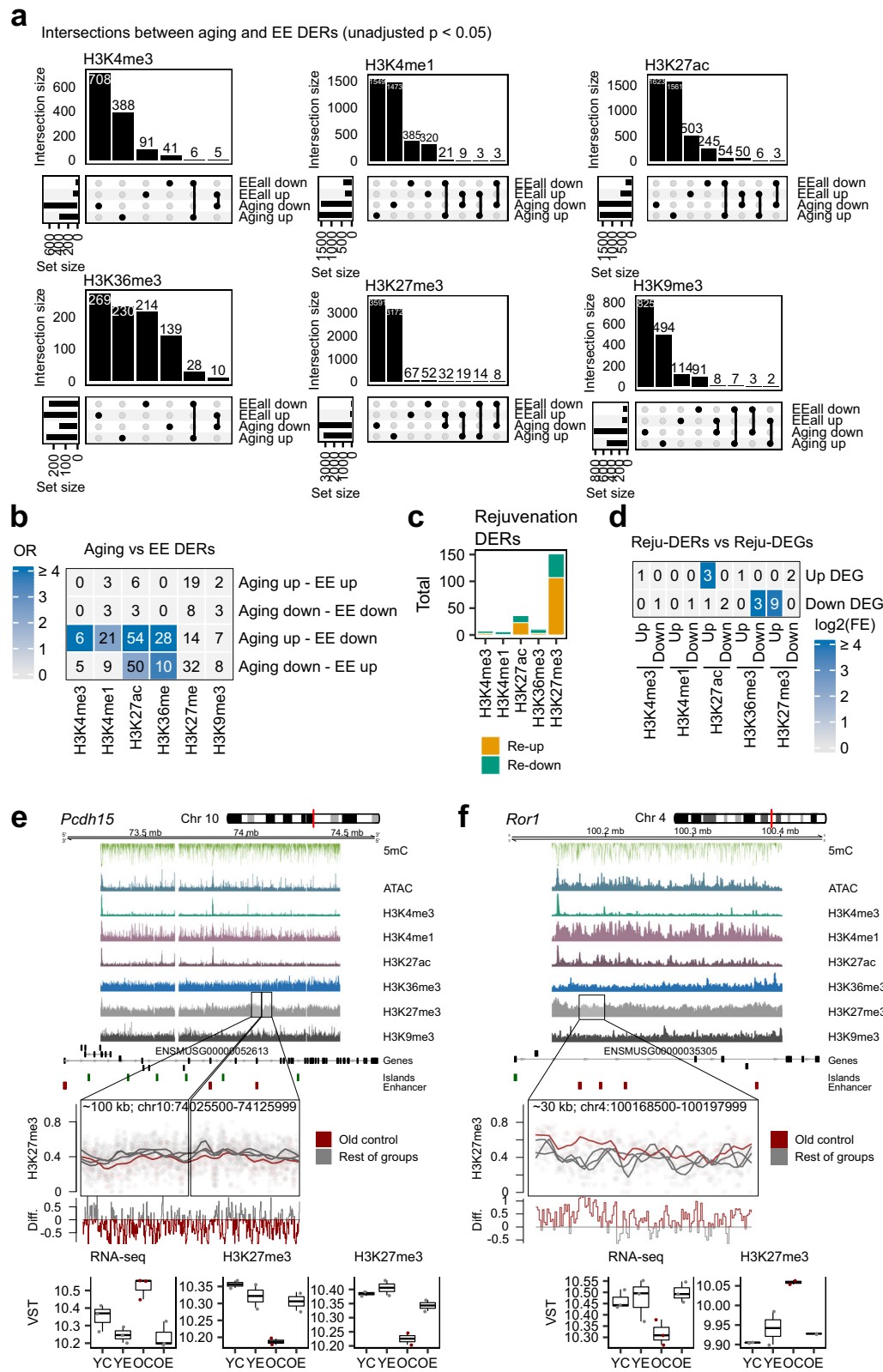

annotations), which expressed well-known markers associated with their annotated identities (Fig. 6b). The majority of cells (~40%) consisted of dentate gyrus neurons, followed by oligodendrocytes (~18%) and CA1/3 area neurons (~15% and 5%, respectively), with the lower, although noticeable, presence of microglia and astrocytes (~4–5%) (Fig. 6c), while the numbers of detected genes were similar across all types (Fig. 6d). When examining the observed group-wise proportions

in cell types, we first confirmed that aging did not lead to large-scale changes in cell composition in the dorsal hippocampus, validating our previous predictions using bulk RNA-seq cell type deconvolution analyses (see previous Fig. S4a) and suggesting that functional alterations taking place during the aging process may be more related to cell-intrinsic changes rather than population-level alterations. An exploratory analysis of smaller alterations revealed the age-associated

**Fig. 5 | Epigenomic rejuvenation of the dorsal hippocampus in enriched environments. a** UpSet plots showing the intersections between aging and EEall ChIP-seq DERs (unadjusted $p < 0.05$, two sided Wald tests) for each histone modification. **b** Heatmap summarizing the significant (FDR < 0.05, one-sided Fisher's exact tests) intersections between aging and EEall ChIP-seq DERs (unadjusted $p < 0.05$, two sided Wald tests) across all histone modifications. Significant intersections are coloured on the basis of their odds ratio. **c** Bar plots describing the numbers of curated DERs which display EE-associated rejuvenation of aging alterations in old samples. **d** Heatmap showing the significant intersections (FDR < 0.05 for one-sided permutation regioneR tests) between curated rejuvenation DERs and curated rejuvenation DEGs. Significant intersections are coloured by log2(fold enrichment). **e, f** Genomic plots indicating, in the upper panels, the distribution of epigenomic marks (5mC, ATAC-seq, histone post-translational modifications) across the bodies of the *Pcdh15* (**e**) and *Ror1* (**f**) genes. The measurements shown are either DNA methylation values for EM-seq profiled CpGs or log-CPM values (200 bp bins and normalized to [0–1] scale) for ATAC-seq and ChIP-seq, while the lower tracks represent genes, CpG Islands and ENCODE murine forebrain P0 enhancer elements. Below, highlighted from the upper plots are regions curated as rejuvenated in old samples, and the log-CPM values are shown across the groups accompanied by the difference in log-CPM between old samples and the rest of samples. Finally, the bottom boxplots show the RNA-seq and ChIP-seq values for the gene and the whole region in VST units ($n = 3$ for all RNA-seq groups and $n = 2$ for all ChIP-seq groups, the reju-DERs and reju-DEGs were defined as stated in Methods). All box plots shown indicate median value, interquartile range (IQR), up to 1.5 IQR (whiskers), and individual data points showing minimum and maximum value.

increase of CA3 neurons, partially reversed by EE, and a reduction in minor populations (Cajal-Retzius cells) (Fig. 6e, f). On the other hand, environmental enrichment led to an increase in vascular cells and also in oligodendrocyte progenitors, coupled with a decrease in mature oligodendrocytes (Fig. 6e, f). This last finding is of particular interest because we had previously pinpointed glial and myelin pathways as being targets for both EE transcriptomic remodelling (see previous Fig. 4c) and the EE-associated rejuvenation of aging alterations (see previous Fig. S8j). As such, these observations indicate that, though aging and EE do not particularly impact cell population dynamics, some changes can be linked to previously detected functional gene expression alterations, and they further confirm that some part of the EE-caused rejuvenation of aging phenotypes may be mediated by oligodendrocyte-dependent pathways.

Next, we performed differential expression analyses within major cell types (Methods) to try and understand the possible functional impact of aging and EE across different populations. Using all groups, we detected numerous changes (adj. $p < 0.05$, logFC > 0.25, Methods, Supplementary Dataset 26) in response to aging and EE, with the former having a stronger effect, as expected (Fig. S9d). Both processes affected the major NEU types (CA1, CA3, DG), and aging also presented numerous MIG and ASC alterations, as well as OLG-related changes, while EE appeared to particularly target a subtype of inhibitory neurons (GABA MGE) (Fig. S9d). Nonetheless, aging and EE alterations were spread across cell types, and we did not observe marked differences in cell type prioritization through these processes using Augur[94] (Fig. S9e). An important proportion of the expression changes were cell type specific, suggesting population-dependent effects of both aging and EE, as has been previously observed for aging[28], with common changes being mostly observed across similar cell types (e.g. CA1, CA3, DG) (Fig. S9f). To investigate if we could extract, as previously observed in the bulk omics, evidence of the EE-associated reversal of aging alterations from the sparse single cell data, we performed permutation testing using the top DEGs ($p < 0.05$) to agnostically check the intersections between aging and EE alterations. Significantly, we observed that the majority of enrichments (empirical FDR < 0.05), across all cell types, corresponded to opposing alterations in the two processes (Fig. 6g), thus indicating that EE-associated aging-reversal signatures are induced in multiple hippocampal populations, with, interestingly, OPC-associated changes being the most enriched.

To extend our exploration to the chromatin level, we carried out differential accessibility analyses of the scATAC data and detected moderate changes (adj. $p < 0.05$, logFC > 0.25, Methods, Supplementary Dataset 27) in response to both aging and EE. Once more, aging had a stronger effect (Fig. S10a), which was present in excitatory populations (CA1, CA3, DG), astrocytes, microglia and oligodendrocytes, while EE mostly targeted glutamatergic neurons (CA1, CA3, DG). The alterations were mostly independent across cell types, though common regions of aging-induced increased accessibility were observed with aging (Fig. S10b). We performed enrichment testing using the top DERs ($p < 0.05$) to explore the putative associations between aging- and EE-induced accessibility changes and, again, discovered that the bulk of intersections (empirical FDR < 0.05) represented opposing changes between the two phenomena (Fig. 6h).

On the whole, these results indicate that signatures of the EE-reversal of aging alterations in both transcriptomic and chromatin accessibility layers can be recovered at the single-cell level.

## Discussion

The aging process involves the accrual of internal and external damage across lifespan. Environmental stimuli, such as lifestyle, can impact the trajectory of this life-long decline, at least partly, through epigenetic modifications, though the precise mechanisms at play remain to be clarified. To tackle these issues, here we set up a murine model of hippocampal aging which was stimulated by a medium-term lifestyle intervention based on environmental enrichment. The EE paradigm used involves an unspecific stimulation which entails alterations in physical, cognitive, and social activity[21,95]. As such, the molecular changes observed in this study comprise a mixture of stimuli which cannot be narrowed down to specific behavioural pathways, and better-controlled interventions should be used to dissect the different molecular signatures attributable, for example, to exercise or cognition[96]. Nonetheless, here we have used EE as a laboratory proxy for general lifestyle or environmental stimulation, which is indeed a very complex phenomenon in humans but for which EE is frequently used as a model[97]. We extensively characterized the molecular dynamics of this model by profiling multiple layers of epigenomic regulation, thus generating a molecular map resource to aid understanding of these processes.

Focusing on aging, we first observed that functional gene expression alterations follow two major axes of regulation whereby up-regulation, more intense, involved an inflammatory response, while down-regulation was associated with less numerous, mRNA metabolism-related pathways (Fig. 1d), partially in agreement with current gene expression analyses across multiple tissues[98,99] and being suggestive of the fact that depletion of the splicing machinery may be explanatory in terms of the recurrent splicing-related alterations described during the aging process across multiple systems[100]. On the other hand, as has been recently described in other tissues such as kidney[43], we observed quite independent age-signatures in the proteome, indicating the involvement of other regulatory layers in linking these two omics–for instance, simply normalizing by protein half-lives increased the observed correlation between RNA and protein levels by 60% (Fig. S4d). We detected numerous epigenetic modifiers as being putatively involved in the regulation of the aforementioned functional alterations (Fig. 1g) and thus profiled DNAm and accessibility levels to find minor changes, including no evidence of age-associated genome-wide demethylation, as is increasingly being reported[73] (Fig. 2a), although we were able to link both DNAm and accessibility changes to the differential expression of certain gene candidates, such as *Cbln1*[63], *Pink1*[64], *C4b*[70] and *Lin28b*[71]. Through locus overlap enrichment, we

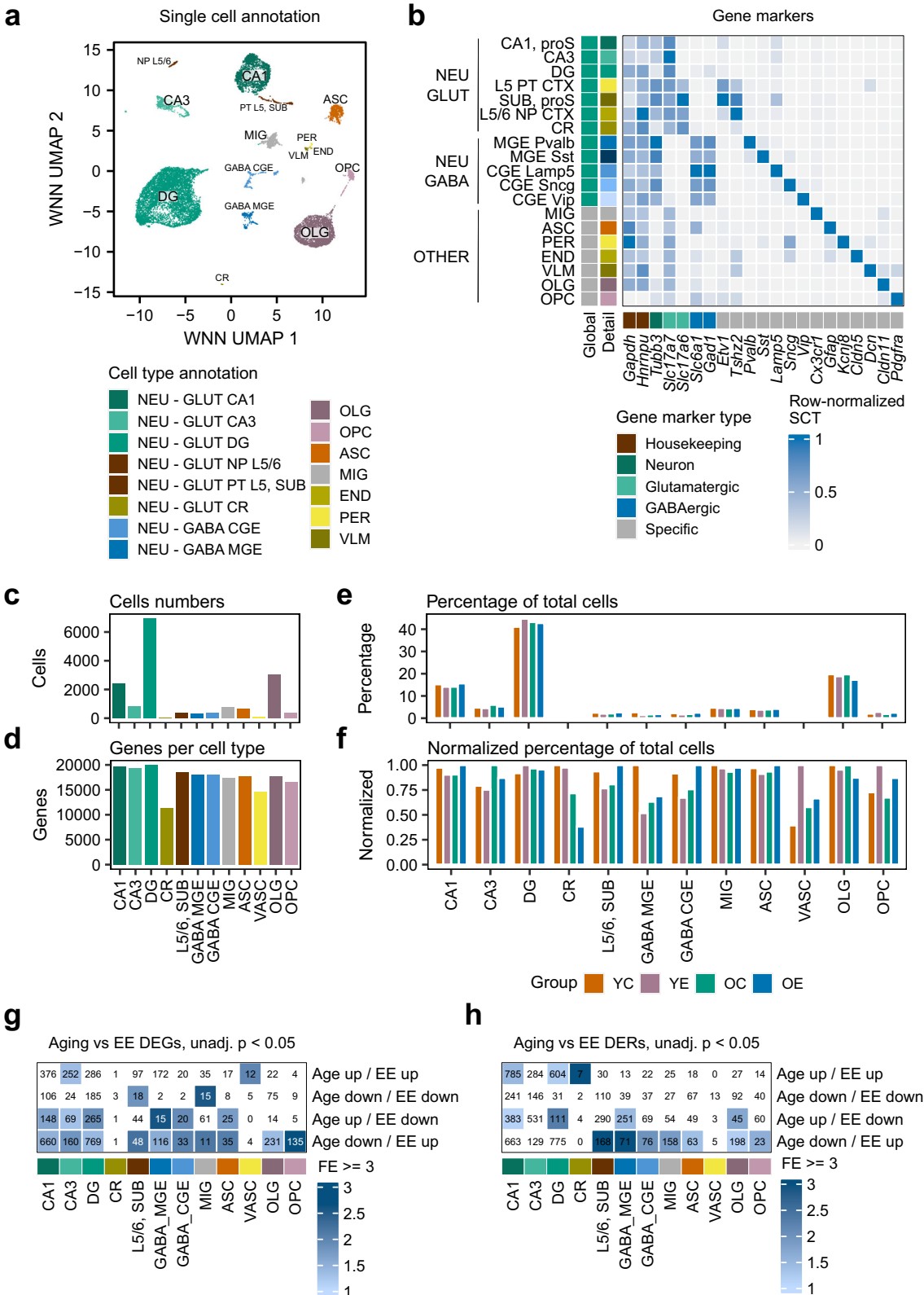

observed that aging accessibility gains were preferentially located at heterochromatin regions (Fig. 2g).

Hence, we characterized the aging-associated histone modification changes of six well-known histone marks (H3K4me1, H3K4me3, H3K27ac, H3K36me3, H3K27me3 and H3K9me3) and uncovered widespread facultative (H3K27me3) and constitutive (H3K9me3) heterochromatin alterations (Fig. 3). Specifically, (a) we observed a

generalized loss of constitutive heterochromatin domains, linked to chromatin decompaction (Fig. 3b), as has been recently described in the aging murine brain[74] and which has been associated with neurodegeneration in experimental models[101,102]; (b) we detected bidirectional changes in H3K27me3 (Fig. 3d), which presented a differential behaviour whereby loss of this mark, and also of H3K4me3, was particularly linked to bivalent chromatin, suggesting the aging-associated

**Fig. 6 | The single-cell dynamics of aging and environmental enrichment in the dorsal hippocampus. a** Dimensional reduction plot showing the distribution of cells, labelled by major cell type annotations for the multimodal WNN UMAP reduction (NEU: neuron; OLG: oligodendrocyte; OPC: oligodendrocyte progenitor; ASC: astrocyte; MIG: microglia; END: endothelial cell; PER: pericyte; VLM: vascular and leptomeningeal cell). **b** Heatmap describing the mean gene expression values ([0–1] row-normalized SCTransform units) within each cell type annotation across a representative panel of marker genes. **c, d** Bar plots showing the total numbers of profiled cells (**c**) and detected genes (**d**), segregated by cell type ("L5/6, SUB" contains NP L5/6, PT L5 and SUB cells). **e, f** Bar plots indicating the percentages (**e**) or normalized percentages (by maximum value for each cell type) (**f**) of cell type counts across each cell type and experimental group. **g** Heatmap describing the significant intersections (FDR < 0.05, one-sided permutation tests) between top (unadjusted $p < 0.05$, two-sided likelihood ratio tests) scRNA-seq DEGs for the aging and EE comparisons (using all samples) across each cell type. Significant intersections are colour-coded according to fold enrichment. **h** Heatmap indicating the significant intersections (FDR < 0.05, one-sided permutation tests) between top (unadjusted $p < 0.05$, two-sided likelihood ratio tests) scATAC-seq DERs for the aging and EE comparisons (using all samples) across each cell type. Significant intersections are colour-coded according to by fold enrichment.

erosion of bivalent domains, an event hardly ever studied in cancer or aging but which could help explain other epigenetic alterations observed in these processes[75] and which may be different between stem and differentiated cell types[103]; and (c) strikingly, we detected a subset of constitutive H3K9me3 loss regions which displayed robust enrichment in facultative H3K27me3 gains (Fig. 3e, f). This phenomenon of "chromatin switching", which has some parallelisms with certain experimental models[76–78], has very recently been described in the aging liver[104]. Whereas Yang N. and colleagues reported a generalized H3K27me3-associated heterochromatinization with aging[104], here we describe more balanced, bidirectional H3K27me3 changes. Nonetheless, our results are in line with their observation that localized loss of H3K27me3 is associated with developmental gene promoters while gains of H3K27me3 are linked to H3K9me3 loss at lamin-associated domains[104]. We hypothesize that in the brain this latter phenomenon could be caused by the cellular re-repression, using facultative heterochromatin, of the constitutive heterochromatin loss during aging.

Finally, we turned to examining how a lifestyle intervention of cognitive stimulation, which has been widely reported to have health benefits[21], could interact with the observed aging molecular dynamics. Significantly, we observed robust evidence of the EE-associated reversal of aging alterations at multiple levels: first, both the transcriptome and the proteome displayed generalized, and often independent, reversal or rejuvenation effects (Fig. 4e, h). Interestingly, we had initially observed that inflammation was a major player in functional aging alterations, with multiple generic and microglia-associated pathways being detected in the gene set enrichment analyses (see Fig. 1d, Fig. S3e, Supplementary Dataset 3). However, the EE-rejuvenation changes were more closely linked to neuronal and glial pathways (Fig. S8l), with no C8 cell type enrichments related to microglia being observed (Supplementary Dataset 22), contrary to the case of aging (Fig. S3h), suggesting that cognitive simulation counteracts specific axes of aging dysregulation which do not involve the age-associated brain inflammatory state[105]. Moreover, our results pointed towards the importance of oligodendrocyte functions in mediating these rejuvenation effects, thus supporting theses discussed elsewhere regarding the putative role of oligodendrocyte pathways in aging and rejuvenation[106–109]. Myelin function is known to be influenced by both neuronal and physical activity[110,111], hence it makes sense that the complex stimulation brought on by EE could have an impact on these pathways. Furthermore, there is evidence that glial cell generation is maintained throughout a great fraction of the murine lifespan (up to 2 years of age)[112], so it is possible that, especially for middle- or old-aged subjects, the glial component is more targetable by environmental stimulation than the neural component of the brain. Notably, reversed genes were also linked to Polycomb/bivalent chromatin pathways, which are widely reported to be epigenetically dysregulated during aging[73] (Fig. S8l). A great number of gene and protein expression EE-reversal changes were found in both young and old individuals. This a priori counterintuitive finding could be explained in the light of recent evidence showing that molecular aging starts very early in life[113,114], so that young subjects could already have accumulated molecular damage amenable to EE reversal. The strength of the gene expression reversal was stronger in young individuals, as expected due to their increased plasticity[84], but this was not the case for the reversal of protein levels, with older subjects displaying similar levels of change.

Regarding the epigenome, we also recovered evidence, though subtler, of the opposing nature of EE and aging alterations through agnostic approaches (Fig. 5a–d), generating a selection of EE-rejuvenation genes with paired epigenomic rejuvenation events, such as *Pcdh15*, a suppressor of OPC proliferation[85], which was elevated in aging with paired H3K27me3 gains and rejuvenated by EE. It is also of interest to mention that protocadherins have been suggested to be targets of stochastic DNAm variation[115], which could reflect an increased sensitivity towards environmental influence. Thus, our results indicate that EE induces a partial, multi-omic reversal of the aging phenotype in the murine dorsal hippocampus. It must be noted that the nature of this reversal appears to be mostly independent across the different omic layers. Indeed, it is known that different molecular layers often display low levels of correlation[43,116], although we did observe significant intersections across several inter-omic changes (see for instance Fig. S4h, Fig. 3b, c or Fig. 5d). Nonetheless, these observations point towards the existence of an omic-specific response of rejuvenation in response to environmental stimulation. To expand on these findings, we additionally profiled the single-cell dynamics of our experimental system (Fig. 6). In general, aging did not cause profound changes in cell type composition in the hippocampus, as has been discussed elsewhere[39], but we did observe a slight increase in the proportion of OPCs, coupled to loss of OLGs, in response to EE (Fig. 6e, f), again suggesting that OLG/OPC pathways are targeted by this stimulation. Finally, by looking at the associations between aging and EE alterations in gene expression and chromatin accessibility, we observed further evidence of the EE-associated reversal of aging changes at the single-cell level (Fig. 6g, h).

To sum up, we have profiled the epigenomic landscape of aging in the dorsal hippocampus and shown that cognitive stimulation by way of environmental enrichment leads to multi-omic rejuvenation in this system.

## Methods
### Animal handling and interventions
Male C57BL/6JRj mice of 9 weeks and 17 months old were acquired from Janvier Labs. Male mice were used because of their longer lifespan which better suited the aging model. Animals from each age group were randomly assigned to control or enriched groups (YC young control, YE young enriched, OC old control, OE old enriched), always housing 5 or 6 animals per cage, and they were acclimated for 2–4 weeks prior to the start of the experiments. The animals were exposed to control or enriched environments for 2 months. Mice were housed under standard laboratory conditions including ad libitum access to food and water, light/dark cycles (12 h/12 h) and stable temperature (20-22 °C), in accordance with the European Union Directive 2010/63/EU. All experiments were performed following the European Community Guidelines (Directive 2010/05/2016) and the Spanish Guidelines (Real Decreto 53/2013) for animal research, and were approved by the ethical committees of the Cajal Institute (Committee

of Ethics and Animal Experimentation), the Spanish Research Council (Subcommittee of Ethics), the Animal Protection Area of the Ministry of Environment of the Community of Madrid (code: PROEX 222/16) and the Research Ethics Committee of the University of Oviedo (Subcommittee on animals and genetically modified organisms, code: PROAE 08/2021). For tissue collection, animals were deeply anaesthetized with pentobarbital and perfused with 0.9% saline. One hemisphere was removed from the skull and the hippocampal formation was isolated in an iced dish and immediately fresh-frozen. The other hemisphere was fixed into 4% paraformaldehyde in 0.1 M phosphate buffer overnight at room temperature. See Supplementary Dataset 1 for information on all of the omics experiments performed.

### Environmental enrichment

The enriched groups were homecaged in groups of 5 or 6 in large PVC cages where multiple objects and toys of different textures, sizes and shapes were placed randomly. The cages also contained ramps, tunnels and different bedding materials. The objects and structures were changed every 2 days or on weekends to form new environments. The bedding was changed every one or two weeks. The control groups were kept in the same room in groups of 5 or 6 in standard conditions with no access to toys.

### Immunohistochemistry

Animals were deeply anaesthetized with pentobarbital and perfused with 0.9% saline and fixed in 4% PFA. Brains were collected and post-fixed over-night. Coronal sections (50μm) were obtained on a Leica VT1000S vibratome. One random series was chosen for each immunohistochemistry. Another randomly chosen series was used for Nissl staining to measure the area of the subgranular zone (SGZ) via the Cavalieri method.

For immunohistochemistry, primary antibodies (DCX, goat anti-doublecortin, 1:500, Santa Cruz #sc-8066; CLR, rabbit anti-calretinin, 1:3000, Swant #7697; SOX2, goat anti-sex determining region Y-box 2, 1:200, R&D Systems #AF2018; GFAP, Rabbit anti-glial fibrillary acidic protein, 1:2000, Abcam #ab7260; pH3, Rabbit anti-phospho-histone H3, 1:500, Millipore #06-570) were incubated in PB 0.1 M, 1% Triton-X100, 1% Bovine Serum Albumin (PBT-BSA) for 1 h at room temperature and 72 h at 4 °C. Secondary antibodies (Donkey anti-goat alexa fluor 594 #A-11058; Donkey anti-rabbit alexa fluor 594 #A-21207; Donkey anti-rabbit alexa fluor 488 #A-21206; Donkey anti-rat alexa fluor 594 #A-21209; all used 1:1000 and obtained from Invitrogen) were incubated in PBT-BSA for 1 h at room temperature and 24 h at 4 °C. Cell nuclei were counter-stained with 4′,6-diamino-2- phenylindole (DAPI, 1:1000, Sigma-Aldrich #D9542).

The physical-dissector method adapted to confocal microscopy (Leica TCS SP5) was used to estimate the total number of neural precursors, i.e. SOX2 + /GFAP+ cells, that displayed radial glia-like morphology (cell body located in subgranular zone and apical dendrite extended to the molecular layer). 7 confocal stacks per animal positioned randomly along the rostro-caudal axis of the dentate gyrus were taken ( ~ 13 photos per stack, 1024×1024, 40x oil immersion objective, step-size of 2.01, zoom 2.4). Cell density per area of the SGZ was calculated in each stack and the mean density was obtained for each animal. The total number of neural precursors was extrapolated using the total area of the subgranular zone measured for each animal in the Nissl staining. A similar methodology was used to calculate the total number of progenitor cells (DCX + /CLR-, DCX + /CLR+ and DCX-/CLR+ cells) in dentate gyrus (6 stacks along rostro-caudal axis, 11 confocal photos per stack, 512×512, 63x oil immersion objective, step-size of 1,76, zoom 2,46).

The fractionator method was used for the estimation of the total number of pH3+ cells that were located in the SGZ. Briefly, the number of pH3+ cells positioned in the SGZ was counted in 1 of each 8 hippocampal sections and multiplied by the fraction.

## Behavioural testing

### Activity measurements

To study basal locomotor activity, a VersaMax Legacy Open Field activity box (Omnitech Electronics, In.) was used. Activity levels were measured on two consecutive days in the same cage to ascertain the activity displayed in a novel environment (first day) and in a known environment (2nd day). The animals freely explored the cage during 5 min on both days. Behavioural measures were automatically scored by the VersaMax system.

### Elevated plus maze (EPM) tests

The animals were placed at the centre of the maze, facing an open arm, and were allowed to explore the maze freely for 5 min. The duration and the number of entries into each arm were recorded on video and manually scored. The apparatus was cleaned between trials with a dilution of water with 0.03% acetic acid. In an independent-laboratory setting, a 10 min exploration test was also performed in similar conditions, with the duration and the number of entries being recorded with a Basler Ace acA1300-60gm camera and semi-automatically computed with the Ethovision XT 16 analysis software. The device was cleaned between trials with 70% ethanol.

### Novel object recognition (NOR) test coupled with novel object location (NOL) test

A rectangular cage (1815 cm²) was used to carry out the NOR test in 3 phases: training (TR), short-term (ST) and long-term (LT). The duration of each phase was 5 min, and mice were habituated (5 min free exploration) to the cage 24 h prior to the first phase. During the TR phase, mice were reintroduced to the cage and exposed to two identical objects (A0 and A1) positioned in two opposing cage quadrants. For the ST phase, 90 min later, object A1 was replaced by a new object (B) and mice were given freedom of movement to explore (A and B; ST test phase). For the LT phase, 24 h later, a third object (C) replaced object B and, again, mice were left to explore (A and C; LT test phase). The apparatus was cleaned between trials with 70% ethanol. The whole test was recorded using a Basler Ace acA1300-60gm zenithal camera and semi-automatically computed with the Ethovision XT 16 analysis software. ST and LT were inferred by measuring the time spent exploring each object and computing a discrimination index (DI): $(N − A) / (N + A)$, where N stands for B at ST and C at LT.

The NOL test for short-term memory was integrated in the same experimental setting of the aforementioned NOR test, made use of the same habituation phase, and was carried out 90 min after the LT phase. Mice were again exposed to two already-known identical objects (A0 and A1), but this time object A1 was displaced 30 cm towards the centre of the homolateral quadrant. Mice were left to explore for 5 min. The arena and objects were cleaned between trials with a dilution of water with 70% ethanol. All phases were video-recorded with a Basler Ace acA1300-60gm and the time exploring each object was semi-automatically computed with the Ethovision XT 16 analysis software. Pattern separation was inferred by measuring the time spent exploring each object and computing a discrimination index (DI): $(M − F) / (M + F)$, where M is time spent with moved object and F is time spent with fixed object.

### Novel object location (NOL) test, independent setting

The test comprised two phases: Training (TR) and Test (TS). During the TR phase, mice were left for 4 min to explore a circular arena that contained two identical columns located symmetrically at its centre. For the TS phase, 40 min later, animals were returned to the arena where one of the columns was displaced diagonally by a distance equivalent to two diameters of the column. As mice normally notice displacements of 3 diameters or higher, in this test the column was moved a shorter distance to evaluate any cognitive improvement derived from treatment, as has been used previously[117]. Mice were

again left to explore for 4 min. The arena and objects were cleaned between trials with a dilution of water with 0.03% acetic acid. All phases were video-recorded and the time exploring each column was manually scored. Pattern separation was inferred by measuring the time spent exploring each object and computing a discrimination index (DI): (M − F) / (M + F), where M is time spent with moved object and F is time spent with fixed object.

### Contextual fear conditioning (CFC) test

A fear conditioning chamber (Ugo Basile Fear Conditioning 2.1, 46003 Mouse Cage) was used to test contextual aversive memory and context discrimination abilities. The test consisted in two phases: acquisition (ACQ) and short term memory (STM). The duration of each phase was 5 min. In the ACQ phase, animals were left to freely explore the conditioning chamber (17 × 17 × 25 cm) whose walls had a pattern of black and white squares (context A). At minutes 3, 3:30 and 4, a floor shock (0.5 mA, 2 s duration) was administered through the floor grid. The animal was left another minute in the cage before the test ended. The STM phase was conducted 24 h after the ACQ phase. Animals were put back in the conditioning chamber, which had the same walls (context A) and left to explore for 5 min. No shock was applied. The conditioning cage was cleaned between trials with a dilution of water with 0.03% acetic acid. Freezing time was automatically scored with the software ANY-MAZE (v6.0).

### Magnetic resonance imaging

Magnetic resonance imaging (MRI) studies were conducted using a 9.4 T horizontal bore magnet (Bruker BioSpin) with 12 cm wide actively shielded gradient coils (440 mT/m). Radiofrequency transmission was achieved with a birdcage volume resonator and signal was detected using a two-element arrayed surface coil (RAPID Biomedical) positioned over the brain of the animal, which was fixed with adhesive tape. MRI procedures were carried out under sevoflurane anaesthesia (4.5% induction and 2.5% maintenance in a gas mixture of 70% $NO_2$ and 30% $O_2$). During MRI studies, each animal was fixed in a Plexiglas holder using a tooth bar, ear bars and adhesive tape to minimize spontaneous movement during imaging acquisition. Respiratory frequency was monitored throughout the experiment.

MRI sequences: the protocol of the study consisted of covering the whole brain with T2-weighted image (T2-wi) in axial and coronal orientations. Also, T1-weighted images (T2-wi) were acquired in order to evaluate any possible artifacts in the images. A Rapid Acquisition with Refocused Echoes (RARE) sequence with the following parameters was used to obtain a T2-weighted anatomical image: 23 axial orientation slices, echo time (TE) = 11 ms, repetition time (TR) = 2.5 s, rare factor (RF) = 8, slice thickness 0.5 mm, no slice separation, field of view (FOV) = 20 × 20 mm$^2$, matrix size 256 × 256 (isotropic in plane resolution of 0.078 mm/pixel). The RARE-T2 in coronal orientation has the same parameters with 16 slices. Images were processed using the software FIJI: ImageJ (v1.50i)[118], and the mouse hippocampus atlas from Badhwar A and colleagues (https://scalablebrainatlas.incf.org)[119].

## RNA-seq analyses

### RNA extraction, RNA library preparation and sequencing

For RNA sequencing, total RNA from fresh-frozen dorsal hippocampi (animals were deeply anaesthetized with pentobarbital and perfused with 0.9% saline) was extracted and DNAse-treated with a silica-membrane column protocol (RNeasy, Qiagen #74104). Concentration of the RNA was determined with the Qubit RNA HS Assay kit (Thermo Fisher Scientific, #Q32852), and integrity on a 2100 Bioanalyzer system (Agilent). Library preparation was carried out with 500 ng of input RNA using the Illumina TruSeq Stranded mRNA protocol (Illumina, #20020594), including polyA selection, following the manufacturer's instructions. Finally, the mRNA-seq libraries were sequenced,

generating 51 bp paired-end reads on an Illumina NovaSeq 6000 system. 3 biological replicates were used per condition.

For the validation cohort, libraries were prepared with 250 ng of input RNA using the NEBNext Ultra II Directional RNA Library Prep kit (NEB, #E7760) including polyA selection with the NEBNext Poly(A) mRNA Magnetic Isolation Module (NEB, #E7490). Finally, the mRNA-seq libraries were sequenced, generating 75 bp paired-end reads, on an Illumina HiSeq 3000/4000 system.

### RNA-seq data preprocessing

FASTQ files were preprocessed and quality controlled with fastp (v0.20.1)[120] using the following options: -r -M 10 -l 20 -p -x --adapter_fasta. Transcript-level quantification was obtained using Salmon (v1.5.0)[121] with the options: --libType A --validateMappings --seqBias --gcBias. For the pseudo-alignment, a decoy-aware index gentrome was built from the mm10 genome and transcriptome. Further preprocessing and statistical analyses were performed within R: transcript-level estimated counts from Salmon were imported and normalized or aggregated to the gene level through the R/Bioconductor package tximport (v1.14.2)[122]. Different analyses used either raw counts, TPM values or VST-normalized values (see below). Ensembl IDs were annotated to gene symbols or other IDs using the R/Bioconductor package biomaRt (v2.42.0)[123].

### Differential gene expression analyses

Transcript-level RNA-seq quantification files from Salmon were imported into R and aggregated to the gene-level through the R/Bioconductor package tximport (v1.14.2). Next, low-expression genes were filtered out with the filterByExpr() function of the R/Bioconductor package edgeR (v3.28.1)[124]. Finally, the R/Bioconductor package DESeq2 (v1.26.0)[125] was used on the filtered gene-level estimated counts to define differentially expressed genes (DEGs) using generalized linear models, based on various comparisons: effect of EE in young, old, or all samples (YC vs YE; OC vs OE; YC + OC vs YE + OE), effect of aging in control or all samples (YC vs OC; YC + YE vs OC + OE). For the comparisons involving all samples, the effect of age or EE was controlled for in the EE and aging comparisons, respectively. An FDR significance level of 0.05 was used to call DEGs. Other analyses or visualization of results were carried out using VST-normalized values, obtained using the vst function in DESeq2.

### Differential isoform and splicing analyses

Isoform-level analyses were handled within the R/Bioconductor package IsoformSwitchAnalyzeR (v1.8.0)[126]. First, transcript-level data were imported and scaled (internal tximport). Then, the pre-Filter function was used to filter out single-isoform genes and isoforms with low usage and low expression (<1 TPM, <1% usage). Next, differential isoform usage analyses were carried out using the DEX-Seq method[127], based on various comparisons: effect of EE in young, old, or all samples (YC vs YE; OC vs OE; YC + OC vs YE + OE), effect of aging in control or all samples (YC vs OC; YC + YE vs OC + OE). For the comparisons involving all samples, the effect of age or EE was controlled for in the EE and aging comparisons, respectively. An FDR significance level of 0.05 and a minimum change in usage of 10% (dIF) were required to call differentially used isoforms. Alternative splicing analyses were performed by integrating the differential isoform usage results with functional and sequence information, through the analyzeAlternativeSplicing method in IsoformSwitchAnalyzeR[128]. In this way, different types of alternative splicing events were annotated and quantified across the differentially used isoforms (DUIs): intron retention (IR), alternative acceptor or donor sites (A3, A5), alternative transcription start or termination sites (ATSS, ATTS), single or multiple exon skipping (ES, MES) and mutually exclusive exons (MEE).

## Cell type deconvolution from bulk RNA-seq

Cell type proportions were predicted from the bulk RNA-seq data by weighted non-negative least squares regression (W-NNLS) implemented in the R package MuSiC (v0.2.0)[129]. In this method, an external single-cell RNA-seq reference dataset with annotated cell types is used to deconvolve cell composition in the input bulk RNA-seq data. To take into consideration possible between-study variability, two independent single-cell RNA-seq datasets, covering different areas of the brain, were used as references for the cell type deconvolution: (1) The Zeisel dataset includes scRNA-seq data for mouse cortex and hippocampus across ~3000 cells[37] (accessed through the R/Bioconductor package scRNAseq (v2.0.2)); (2) The Yao dataset contains scRNA-seq data for mouse isocortex and hippocampal formation across ~75,000 cells[38] (accessed through the Allen Brain Atlas database[130]). Genes with no expression across all cells and those cells labelled as outliers in the datasets were removed.

## Quantification of global 5mC by LC-MS/MS

Total DNA from fresh-frozen dorsal hippocampi (animals were deeply anaesthetized with pentobarbital and perfused with 0.9% saline) was extracted and RNAse-treated following a standard phenol-chloroform protocol. Concentration of the DNA was determined with the Quant-iT Picogreen dsDNA assay kit (Thermo Fisher Scientific, #P7589) at 480/520 nm on a Fluostar Optima plate reader (BMG Labtech), and integrity was checked on an E-Gel EX Agarose gel 2% (Thermo Fisher Scientific, #G401002). To analyse global levels of 5-methylcytosine (5mC) by mass-spectrometry, 200 ng of genomic DNA per sample were digested to a final concentration of 1 ng/µL with DNA Degradase Plus (Zymo Research, #E2020) and subjected to mass spectrometry (liquid chromatography electrospray ionization tandem mass spectrometry). All samples were analysed using an Agilent 1200 liquid chromatograph (Agilent Technologies) coupled to an API 4000 Liquid Chromatography Tandem−Mass Spectrometry (LC-MS/MS) system (AB Sciex). A Zorbax Eclipse XDB C18 column (2.1 mm * 150 mm, 5 µm; Agilent Technologies) was used, with a two-phase flow of formic acid 0.1% in $H_2O$ (A) and formic acid 0.1% in MeOH (B). Three biological replicates were pooled for each condition and 3 technical replicates were measured for each sample.

## EM-seq analyses

### DNA extraction, EM-seq library preparation and sequencing

For enzymatic methyl sequencing (EM-seq), total DNA from fresh-frozen dorsal hippocampi (animals were deeply anaesthetized with pentobarbital and perfused with 0.9% saline) was extracted and RNAse-treated following a standard phenol-chloroform protocol. Concentration of the DNA was determined with the Qubit dsDNA HS Assay kit (Thermo Fisher Scientific, #Q32854) and integrity on a 2100 Bioanalyzer system (Agilent). Library preparation was carried out with 100 ng of input DNA using the NEBNext Enzymatic Methyl-seq Kit (NEB, #E7120) following the manufacturer's instructions. Finally, the EM-seq libraries were sequenced, generating 151 bp paired-end reads, on an Illumina NovaSeq 6000 system. 3 biological replicates were pooled for each condition.

### EM-seq data preprocessing

Quality control of FASTQ files was performed with FastQC (v0.11.9), and an average 99% conversion rate was determined using BCREval (v0)[131]. Reads were trimmed with Trim Galore! (v0.6.7) using the options --gzip --paired and --2colour 20. Preprocessed reads were aligned to the bisulfite-converted- and indexed mm10 genome via Bismark (v0.23.1)[132] running Bowtie2[133] under default parameters. Aligned reads were deduplicated using Bismark and methylation calling was performed with Bismark's methylation extractor, using the options --paired-end --comprehensive --no_overlap --gzip --bedGraph --ignore_r2 2 --cytosine_report to obtain methylation counts for

cytosines belonging to CpG sites. The methylation callings were filtered to remove those mapping to alternative contigs and scaffolds as well as to blacklisted regions defined by ENCODE (mm10, v2)[134]. Next, methylation counts from both strands belonging to the same CpG site were pooled so as to have one methylation measurement per CpG site. Subsequently, the CpG sites were filtered to remove low coverage (<10 counts) and high coverage (>99.9th percentile counts) sites. Finally, for each position, methylation values were computed as the percentage of methylated cytosines with respect to total cytosines on a scale of 0–1.

### Differential methylation analyses

Differentially-methylated regions (DMRs) were computed using metilene (v0.2.8)[135] with options -c 2 -d 0.05 -f 1. The called DMRs were required to contain a minimum number of 10 CpGs with a mean methylation difference of at least 5% in 5mC, with an FDR significance level of 0.05.

## ATAC-seq analyses

### Nuclei extraction, ATAC-seq library preparation and sequencing

For the assays of transposase-accessible chromatin using sequencing (ATAC-seq), cell nuclei from fresh-frozen dorsal hippocampi (animals were deeply anaesthetized with pentobarbital and perfused with 0.9% saline) were extracted by first homogenizing the tissue with a Dounce homogenizer and filtering through a 40 µm Nylon cell strainer (Corning). Then, the nuclei were isolated using an iodixanol gradient and counted with trypan blue on a TC20 automated cell counter (Biorad). The transposition assay was performed on 50,000 pelleted (500 g) nuclei per condition, using the prokaryotic Tn5 transposase system (Nextera DNA Library Prep Kit, Illumina, #FC-121-1030) for 30 min at 37 °C. Next, transposed DNA was purified on DiaPure columns (Diagenode, #C03040001).

Library preparation was performed using the Nextera DNA Library Prep Kit (Illumina, #FC-121-1030) protocol. Amplification was evaluated using qPCR with NEBNext High-Fidelity PCR MasterMix (NEB, #M0541) on a LightCycler 96 System (Roche). Library purification and selection was conducted using Agencourt AMPure XP (Beckman Coulter, #A63881). The quality of the resulting libraries was checked with Qubit dsDNA HS Assay kit (Thermo Fisher Scientific, #Q32854) and their size assessed with High Sensitivity NGS Fragment Analysis Kit on a Fragment Analyzer (Agilent). Samples were pooled prior to sequencing. Finally, ATAC-seq libraries were sequenced, generating 50 bp paired-end reads, on an Illumina NovaSeq 6000 system. Two biological samples were pooled for each biological replicate, and 2 biological replicates were sequenced for each condition.

### ATAC-seq data preprocessing

FASTQ files were preprocessed and quality controlled with fastp (v0.20.1)[120] using the following options: -l 20 -p --adapter_fasta. In brief, Illumina Nextera adaptor sequences were trimmed using a custom FASTA file and reads containing >5 N nucleotides, reads containing >40% proportion of <15 Phred nucleotides and reads of <20 bp in length were filtered out. Next, reads were aligned to the indexed mm10 genome via Bowtie 2 (v2.4.2)[133] with the options -X 2000 --very-sensitive-local. Raw mappings to the mitochondrial genome ranged from 1.5% to 8.7%. Alignments were deduplicated using Picard (v2.23.9) and filtered using SAMtools (v1.7)[136] to retain only properly paired reads with MAPQ > 10 mapping to autosomal and sex chromosomes. Subsequently, ENCODE blacklisted regions (mm10, v2)[134] were filtered out from the alignments using BEDTools (v2.29.2)[137]. Finally, the alignment coordinates were shifted 9 bp (+4 and −5 bp for positive and negative strands) to adjust for the transposition cut location[69] using deepTools alignmentSieve (v3.5.0)[138]. bigWig visualization files were generated at different resolutions with deepTools bamCoverage (v3.5.0) using the RPGC normalization[138].

## Chromatin accessibility analyses

Regions of accessible chromatin (peaks) were called for each biological replicate using the epic2 reimplementation (v0.0.48)[139] of SICER[140] and including the following parameters: --false-discovery-rate-cutoff 0.05 --bin-size 200 --gaps-allowed 3. The peaks of biological replicates were combined by taking the union of overlapping peaks.

Differentially accessible regions (DARs) were defined using the R/Bioconductor package DESeq2 (v1.26.0)[125]: first, a consensus peak set was defined by reducing the aforementioned replicate-combined peaks from each group; next, accessibility at consensus peaks for each biological replicate was quantified using featureCounts as implemented in the R/Bioconductor package Rsubread (v2.0.1)[141]; finally, the quantifications were imported to DESeq2 to perform differential analyses. DARs were called using generalized linear models, based on various comparisons: effect of EE in young, old, or all samples (YC vs YE; OC vs OE; YC + OC vs YE + OE), effect of aging in control or all samples (YC vs OC; YC + YE vs OC + OE). For the comparisons involving all samples, the effect of age or EE was controlled for in the EE and aging comparisons, respectively. An FDR significance level of 0.05 was used to call DARs. Other analyses or visualizations of results were carried out using VST-normalized values, obtained using the vst function in DESeq2. Quantification of accessibility at other genomic regions was also performed using the aforementioned method.

## ChIP-seq analyses

### Chromatin extraction, ChIP-seq library preparation and sequencing

For chromatin immunoprecipitation and sequencing (ChIP-seq), total chromatin from fresh-frozen dorsal hippocampi (animals were deeply anaesthetized with pentobarbital and perfused with 0.9% saline) was extracted, preprocessed and immunoprecipitated using the iDeal ChIP-seq kit for histones (Diagenode, #C01010059). Fragmentation was performed with a Bioruptor Pico sonication device (Diagenode). The quality of the sheared chromatin was checked with the High Sensitivity NGS Fragment Analysis Kit on a Fragment Analyzer (Agilent). DNA was quantified after reverse cross-linking via the Qubit dsDNA HS Assay kit (Thermo Fisher Scientific, #Q32854). ChIP efficiency was measured via qPCR using KAPA SYBR FAST (Sigma-Aldrich, #KK4601). The following antibodies were used for immunoprecipitation: H3K4me3 (Diagenode, #C15410003, Lot. A1051D), H3K4me1 (Diagenode, #C15410194, Lot. A1862D), H3K27ac (Diagenode, #C15410196, Lot. A1723-0041D), H3K9me3 (Diagenode, #C15410193, Lot. A0219P), H3K27me3 (Diagenode, #C15410195, Lot. A0821D) and H3K36me3 (Diagenode, #C15410192, Lot. A1845P). 1% of total chromatin was set aside to be used as Input control in the ChIP-seq experiments. Library preparation was performed with the MicroPlex Library Preparation Kit v3 (Diagenode, #C05010002) with 24 UDI for MicroPlex v3 - Set I and Set II (Diagenode, #C05010008 and #C05010009). The amplification was assessed with qPCR and capillary electrophoresis using the aforementioned equipment. Library purification (double size selection) was carried out using Agencourt AMPure XP (Beckman Coulter, #A63881). The quality of the resulting libraries was checked by Qubit dsDNA HS Assay kit (Thermo Fisher Scientific, #Q32854) and their size assessed with High Sensitivity NGS Fragment Analysis Kit on a Fragment Analyzer (Agilent). Samples were pooled prior to sequencing. Finally, ChIP-seq libraries were sequenced, generating 50 bp paired-end reads, on an Illumina NovaSeq 6000 system. 3 biological samples were pooled for each biological replicate, and 2 biological replicates were sequenced for each condition. If the enrichment efficiencies were subsequently found to be disparate between the samples, biological replicates were replaced with pseudo-technical replicates (see Supplementary Dataset 1).

### ChIP-seq data preprocessing

FASTQ files were preprocessed and quality controlled with fastp (v0.20.1)[120] using the following options: -l 20 -p --adapter_fasta. In brief,

Illumina TruSeq adaptor sequences were trimmed using a custom FASTA file and reads containing >5 N nucleotides, reads containing >40% proportion of <15 Phred nucleotides and reads of <20 bp in length were filtered out. Next, reads were aligned to the indexed mm10 genome via Bowtie 2 (v2.4.2)[133] with the options -X 2000 --very-sensitive-local. Alignments were deduplicated using Picard (v2.23.9) and filtered using SAMtools (v1.7)[136] to retain only properly paired reads with MAPQ > 10 mapping to autosomal and sex chromosomes. Subsequently, ENCODE blacklisted regions (mm10, v2)[134] were filtered out from the alignments using BEDTools (v2.29.2)[137]. In addition, the Input control samples were used to define experiment-specific greylists with the R/Bioconductor packages GreyListChIP (v1.18.0) and BSgenome.Mmusculus.UCSC.mm10 (v1.4.0) by detecting regions of high and probably spurious signal in the Input samples. These regions were also removed from the alignments using BEDTools (v2.29.2)[137]. bigWig visualization files were generated at different resolutions with deepTools bamCoverage (v3.5.0) using the RPGC normalization[138]. The enrichment of histone signal at chromatin states and CpG islands was computed using featureCounts as implemented in the R/Bioconductor package Rsubread (v2.0.1)[141]. Then, the percentage of reads mapped to each region was computed and compared to the percentage in the input samples.

### Histone enrichment analyses

Regions enriched in histone modifications (peaks) were called for each biological replicate using the epic2 reimplementation (v0.0.48)[139] of SICER[140] and optimized parameters for each histone mark: --bin-size 100 --gaps-allowed 2 for sharp marks (H3K4me3); --bin-size 200 --gaps-allowed 3 for narrow marks (H3K4me1, H3K27ac); --bin-size 500 --gaps-allowed 4 for broad marks (H3K36me3, H3K27me3, H3K9me3). All peaks were called with --false-discovery-rate-cutoff 0.05. The peaks of biological replicates were combined by taking the union of overlapping peaks.

Differentially enriched regions (DERs) were defined using the R/Bioconductor package DESeq2 (v1.26.0)[125]: first, a consensus peak set was defined by reducing the aforementioned replicate-combined peaks from each group; next, accessibility at consensus peaks for each biological replicate was quantified using featureCounts as implemented in the R/Bioconductor package Rsubread (v2.0.1)[141]; finally, the quantifications were imported to DESeq2 to perform differential analyses. DERs were called using generalized linear models, based on various comparisons: effect of EE in young, old, or all samples (YC vs YE; OC vs OE; YC + OC vs YE + OE), effect of aging in control or all samples (YC vs OC; YC + YE vs OC + OE). For the comparisons involving all samples, the effect of age or EE was controlled for in the EE and aging comparisons, respectively. An FDR significance level of 0.05 was used to call DERs. Other analyses or visualizations of results were carried out using VST-normalized values, obtained using the vst function in DESeq2. Quantification of accessibility at other genomic regions was performed using the aforementioned method.

### Chromatin state analyses

Chromatin states were learned by integrating the ChIP-seq data for histone modifications using ChromHMM (v1.23)[142]. A single 15-state model was learned to produce specific annotations for each sample following the "concatenated" strategy, and control Input samples were included to adjust the binarization threshold locally[80]. To learn the model, first, filtered BAM files were binarized with the BinarizeBam function including the -paired parameter and default settings. Then, the 15-chromatin state model was learned using the LearnModel function, including the -printstatebyline and -printposterior parameters and the mm10 assembly. Next, enrichments of the states in genomic regions, across all groups, were computed with the OverlapEnrichment and NeighborhoodEnrichment functions with the -posterior parameter, using mm10 genomic locations, as well as

ENCODE3 chromatin state tracks for mouse postnatal P0 forebrain[72]. Finally, the states were annotated to biological functions (Pr-A, Pr-W, Pr-F, Tr-S, Tr-S2, Tr-P, Tr-I, En-Sd, En-Sp, En-Pd, Hc-P, Hc-Pw, Hc-H, NS) by using prior biological knowledge and the aforementioned ENCODE3 chromatin state annotations for mouse postnatal forebrain[72,81].

## Mass spectrometry analyses

### Protein extraction, digestion and LC-MS/MS runs
For sequential window acquisition of all theoretical mass spectra (SWATH-MS) proteomic analyses, total protein from fresh-frozen dorsal hippocampi (animals were deeply anaesthetized with pentobarbital and perfused with 0.9% saline) was extracted using a standard RIPA buffer and was precipitated overnight in 80% acetone / 10% TCA. Protein extracts were resuspended in 0.2% RapiGest SF Surfactant (Waters, #186002123) and quantified via the Qubit Protein Assay kit (Thermo Fisher Scientific, #Q33211). Next, 25 μg of protein were incubated in 4.5 mM DTT for 30 min at 60 °C and in 10 mM iodoacetamide for 30 min at RT. Subsequently, a trypsin digestion was performed overnight at 37 °C in a 1:40 (enzyme:protein) proportion, which was stopped with formic acid. After incorporating SWATH alignment peptides at 40 fmol/μL (Sciex), LC-MS/MS runs were performed with injections of 1 μg of protein on a hybrid mass spectrometer TripleTOF 5600+ System (Sciex) coupled to a NanoLC 425 System (Sciex), using the Analyst TF (v1.7) software for equipment control, data acquisition and processing. Each biological sample was run 3 times generating 3 technical replicates.

### LC-MS/MS runs, library preparation and SWATH-MS quantification
Peptides were first loaded into a trap column (Acclaim PepMap 100 C18, 5 μm, 100 Å, 100 μm id × 20 mm, Thermo Fisher Scientific) isocratically in 0.1% formic acid/2% acetonitrile (v/v) at a flow rate of 3 μL/min for 10 min. Next, elution was performed in a reverse-phase column (Acclaim PepMap 100 C18, 3 μm, 100 Å, 75 μm id × 250 mm, Thermo Fisher Scientific) coupled to a PicoTip emitter (New Objective, #FS360-20-10-N-20-C12) using a lineal gradient of 2-35% (v/v) of the B solvent in 120 min at 300 nL/min. As A and B solvents, 0.1% formic acid (v/v) and acetonitrile with 0.1% formic acid (v/v) were used, respectively. Voltage was set to 2600 V and temperatures maintained at 100 °C. Gas 1 was selected at 15 psi, gas 2 at 0, curtain gas at 25 psi.

For library preparation, data acquisition was performed via DDA (data-dependent acquisition) using a TOF MS scan between 400–1250 m/z, accumulation time of 250 ms, followed by 50 MS/MS (230–1500 m/z), accumulation time of 65 ms and a total cycle time of 3.54 s. Ten runs were used for spectral library preparation, in which samples were mixed in pairs and injected using the aforementioned DDA method. The ProteinPilot software (v5.0.1, Sciex) was used for peptide identification in a joint search across the 10 runs. For the Paragon method, the following parameters were used: trypsin enzyme, iodoacetamide as alkylating agent, and the mouse UniProt proteome as reference (17/12/2021) with Sciex's contaminant database. Proteins were selected under an FDR significance level of 0.01.

For the final SWATH runs, data acquisition used a TOF MS scan of between 400–1250 m/z, accumulation time of 50 ms, followed by a DIA (data-independent acquisition) method with 60 m/z windows of variable size (230–1500 m/z) with 60 ms acquisition time and a cycle time of 3.68 s. The gradient used was the same as for the DDA method. Between samples (1 μg of digested protein), a standard control (Pepcalmix, Sciex, #5045759) was used to calibrate the equipment and control sensitivity and chromatographic conditions. Finally, the data were preprocessed using PeakView (v2.2, Sciex) with the SWATH 2.0 microapp, generating total area-normalized measurements for each protein.

### SWATH proteomics data postprocessing
UniProtKB IDs from the SwissProt and trEMBL databases were mapped to Ensembl IDs and gene symbols via the R/Bioconductor package biomaRt (v2.42.0)[123]. UniProtKB IDs mapping to >1 different Ensembl ID or not mapping to any Ensembl ID were considered unmapped and filtered out. In addition, intensity values for proteins with UniProtKB IDs mapping to the same gene were averaged. Normalized-area data were log2-transformed prior to statistical analyses.

### Differential protein expression analyses
The R/Bioconductor package limma (v3.42.2)[143] was used on the log2-transformed area-normalized protein measurements to define differentially expressed proteins (DEPs) using linear models, based on various comparisons: effect of EE in young, old, or all samples (YC vs YE; OC vs OE; YC + OC vs YE + OE), effect of aging in control or all samples (YC vs OC; YC + YE vs OC + OE). For the comparisons involving all samples, the effect of age or EE was controlled for in the EE and aging comparisons, respectively. An FDR significance level of 0.05 was used to call DEPs. Technical replicates were averaged into biological replicates within limma. Other analyses or visualizations of results were also carried out using log2-transformed area-normalized values.

### Other analyses
**Epigenetic landscape in silico deletion analysis.** To find transcriptional regulators of gene sets of interest, epigenetic landscape in silico detection analysis (Lisa) was performed[47], using the Python Lisa2 implementation (https://github.com/liulab-dfci/lisa2). Cistrome data representing ChIP-seq and DNase-seq tracks associated with transcriptional regulators were retrieved for the mm10 genome (http://cistrome.org/~alynch/data/lisa_data/mm10_1000_2.0.h5)[48] and installed into Lisa2. Next, gene sets of interest were assayed via the FromGenes and predict functions, using default parameters and the filtered list of RNA-seq genes as custom background. Finally, the summary p-values obtained were adjusted for multiple testing and significant results were filtered for FDR < 0.05 while only retaining tracks corresponding to brain-associated tissues.

**Canonical aging gene sets.** Canonical aging gene sets were retrieved from different databases: (1) the Digital Ageing Atlas database (https://ageing-map.org/)[27] was used to retrieve the mouse genes with aging-associated changes in >1 tissues. (2) The GenAge database (https://genomics.senescence.info/genes/microarray.php)[26] was used to retrieve the set of genes commonly altered during ageing across mammalian species. (3) Genes associated with murine brain aging were retrieved from Ximerakis et al.[28]. and correspond to the brain-level cell-type aggregated results. The top aging-associated genes with at least a 20%-fold change difference were selected.

**Pathway enrichment analyses.** Gene sets of interest such as detected DEGs, or genes associated with regions of interest, were evaluated for their enrichment in the pathway collections of the MSigDB v7.4 (Molecular signatures database)[36]. The collections were accessed through the R/CRAN package msigdbr (v7.4.1), which provides the Mus musculus orthologous sets defined using the HUGO Comparison of Orthology Predictions tool (HCOP) by choosing, for each human gene, the mouse ortholog supported by the largest number of databases. Over-enrichment tests were performed with the R/Bioconductor package goseq (v1.38.0)[144], which was used to correct for biases due to expression levels (for expression data) or gene length (for genes associated with genomic regions). Appropriate filtered lists of genes were used as backgrounds for each case, and the over-enrichment p-values obtained were adjusted for multiple testing, with significant pathways being selected at an FDR significance level of 0.05.

**Annotation of genomic regions.** Genomic loci were annotated to different genomic features using various tools. For CpG island annotation, the unmasked CpG island track for the mm10 genome was retrieved using the UCSC Table Browser tool[145], and CpG shores and shelves were defined as flanking regions next to CpG islands comprising the 0–2000 bp and 2000–4000 bp intervals, respectively. Then, regions of interest were annotated to CpG islands using the R/Bioconductor package GenomicRanges (v1.39.3)[146]. To annotate regions to genes, transcripts and gene parts, the Gencode transcriptome annotation GTF was used with the R/Bioconductor package ChIPseeker (v1.22.1)[147]. Promoters were considered to be those regions 1000 bp upstream and 100 bp downstream from transcript-level transcription start sites, with regions 1000–3000 bp upstream being considered distal promoters.

To generate a genomic background annotation to be used in certain comparisons involving regions of interest (e.g. ATAC peaks), the mm10 chromosome coordinates were retrieved. These were then filtered for masked regions (AGAPS and AMB) retrieved from the R/Bioconductor package BSgenome.Mmusculus.UCSC.mm10.masked (v1.3.99) and ENCODE blacklisted regions (mm10, v2)[134]. Then, the final genome was binned into 200 bp bins and annotated to genomic loci as described above.

**Genomic region intersection permutation tests.** Genomic region intersection tests were carried out using permutation testing via the regioneR package (v1.18.1)[148]. To test for over-enrichment in the overlap between sets of genomic regions (for example, between sets of histone DERs), their intersections were compared to those with randomly sampled genomic regions of an equivalent size distribution (randomizeRegions function). The background genome used for the sampling was defined as described in the previous section. For the case of testing over-enrichment of genomic regions in pre-defined regions of interest (for example, in gene locations), their intersections were compared to those with randomly sampled regions from the appropriate universe of regions (for example, all the gene coordinates; resampleRegions function). Regardless of the approach taken, for each test, null distributions were built by running 1000 permutations of the samplings to obtain an empirical $p$-value, and one-sided tests were performed. When performing multiple tests, the $p$-values were adjusted for multiple testing within each set of comparisons and declared significant at FDR < 0.05. The fold enrichment of the intersections was computed as observed_interesection / mean(permuted_intersection).

**Gene intersection permutation tests.** In order to test intersections between gene sets retrieved from different backgrounds (e.g. RNA DEGs versus protein DEGs, or single-cell DEGs across different cell types), permutation tests were used. Over-enrichment in the overlap between sets was computed by randomly sampling sets of equivalent size from the corresponding universes and measuring the intersection. Null distributions were built by running 1000 permutations of the samplings to obtain an empirical $p$-value. When performing multiple tests, the $p$-values were adjusted for multiple testing within each set of comparisons and declared significant at FDR < 0.05. A measure of fold enrichment was defined as the ratio of the observed intersection to the mean intersection of the null distribution.

**Locus overlap analyses.** Locus overlap analysis (LOLA) was performed with the R/Bioconductor package LOLA (v1.16.0)[149] using custom databases. Sets of regions were tested for over-enrichment in specific genomic regions using one-sided Fisher's exact tests (FDR < 0.05) with appropriate filtered backgrounds being used in each case.

The chromatin state tracks used in the LOLA analyses for mouse postnatal P0 tissues from ENCODE3[72] were obtained from http://renlab.sdsc.edu/renlab_website/download/encode3-mouse-histone-atac.

**Protein half-life normalization.** Protein half-lives for mouse embryonic neurons, as estimated by Mathieson and colleagues[42], were used to normalize protein measurements. The half-lives of any reported quality for the two replicates provided were averaged, and missing or infinite values were filtered out. Finally, log2-transformed area-normalized protein measurements were divided by their respective protein half-life to produced life-normalized values.

**Definition of bivalent and chromatin-switching chromatin domains.** Bivalent chromatin domains were defined as regions of intersecting H3K4me3 and H3K27me3 consensus peaks. Chromatin-switching domains were defined as regions of intersecting H3K27me3 aging up-DERs and H3K9me3 aging down-DERs.

**Curation of rejuvenating genes and regions.** To curate RNA-seq reversal genes (see Supplementary Dataset 21), aging DEGs (FDR < 0.05) were selected with log2(FC) > 0.25 which displayed opposite changes associated with EE with log2(FC) > 0.25. To select genes specifically rejuvenated in old samples, differential analyses were performed with DESeq2 to compare OC vs YC + YE + OE. Then, significant genes (FDR < 0.05) with log2(FC) < 0.25 and which, additionally, did not display evidence of change with EE in young samples ($p \geq 0.05$) were selected.

To curate proteomic reversal and old-rejuvenation genes (Supplementary Dataset 21), the same procedure was followed, except that limma was used for the differential protein analyses. Finally, rejuvenation genes with consistent measurements of RNA and protein alterations were defined by choosing aging DEGs and DEPs (unadjusted $p < 0.05$) which changed in the same direction and showed a parallel but opposite direction with EE.

To curate ChIP-seq regions rejuvenated in old samples (see Supplementary Dataset 24), differential analyses were performed with DESeq2 to compare OC vs YC + YE + OE. Then, significant regions (FDR < 0.05) with at least a log2(FC) > 0.25 change in OC samples compared to OE samples and which, additionally, did not display evidence of change with EE in young samples ($p \geq 0.05$) were selected.

With regards to the direction of change of rejuvenating genes and regions, "down" changes refer to events in which the rejuvenation brings back down the expression or epigenomic levels, while "up" changes refer to the opposite trend.

## Single cell sequencing analyses
### Single cell profiling of gene expression and chromatin accessibility
Fresh-frozen dorsal hippocampi (animals were deeply anaesthetized with pentobarbital and perfused with 0.9% saline) were homogenized in 4 ml of lysis solution (Tris-Hcl 20 mM pH: 7.5, Tween20 0.1%, Sucrose 0.25 M, KCl 25 mM, MgCl2 5 mM) using a douncer (Fisher Scientific) 10-15 times on ice. Resulting homogenates were centrifuged at 4 °C 500×g for 5 min and the debris-containing supernatants removed. Nuclei pellets were further cleaned using a mixture of 4 ml lysis solution and 2 ml Optiprep (ProteoGenix, #1114542) and centrifuging at 4 °C 1500×g for 10 min. Samples were resuspended in 300 μL of lysis solution and the concentration of nuclei quantified using a Neubauer counting chamber (Karl Hecht). Per condition, the equivalent nuclei quantities of three samples (1.5 M nuclei each) were pooled and centrifuged at 4 °C 500×g for 5 min. The rest were kept at -80 °C in a mixture of 70% lysis solution, 30% DMSO and 2% BSA. Nuclei were permeabilized in 100 μL of 0.1X lysis buffer (10x Genomics, #CG000375) for 2 min on ice. Permeabilization was stopped by adding 1 ml 1X Wash buffer (10x Genomics, #CG000375), mixing and centrifuging at 4 °C 500×g for 5 min. Permeabilized nuclei were then resuspended in 100 μL 1X Diluted Nuclei Buffer (10x Genomics, #CG000375), filtered using a pluriStrainer Mini 20 μm (Pluriselect,

#43-10020-40) and finally quantified using a Countess 3 FL Automated Cell Counter (Thermo Fisher Scientific).

Single-nuclei RNA- and ATAC-seq libraries were generated using the Chromium Next GEM Single Cell Multiome ATAC + Gene Expression kit according to the manufacturer's instructions (10x Genomics, #CG000338). The transposed nuclei suspension was loaded onto Next GEM Chip J targeting 16,000 nuclei and then run on a Chromium Controller instrument to generate GEM emulsion (10x Genomics). Libraries were quantified using Agilent Bioanalyzer High Sensitivity DNA kit (Agilent). Finally, the libraries were sequenced on an Illumina NovaSeq 6000 system.

## Single cell data preprocessing

The scRNA and scATAC FASTQ reads generated were jointly analysed using the Chromium Multiome Cell Ranger ARC pipeline (v2.0.1, 10X Genomics). Each sample was run independently to perform alignment (mm10 genome), quantification and joint cell calling using the paired RNA and ATAC information. The output filtered HDF5 matrices were subsequently processed within R using Seurat, SingleCellExperiment and ChromatinAssay objects from the R packages Seurat (v4.3.0)[93], SingleCellExperiment (v1.20.0)[150] and Signac (v1.8.0)[151].

Preprocessing was performed individually for each sample. First, cell doublets were inferred from the scRNA data using the R/Bioconductor packages scds (v1.14.0, hybrid method)[152] and scDblFinder (v1.12.0)[153], the latter also being used to infer cell doublets from the scATAC data by activating the aggregateFeatures parameter. Those cell barcodes marked as doublets across the two omic layers were filtered out. In addition, standard filtering steps were used within Seurat to remove low quality cells across both omics: for scRNA, those cells with library counts >99th percentile, mitochondrial percentage >99th percentile, ribosomal percentage <1st percentile, containing <1000 total counts and <200 total features were filtered out, while genes mapping to >1 Ensembl symbol or expressed in <3 cells per group were removed. For scATAC, those cells with library counts >99th percentile, feature counts >99th percentile, TSS enrichment score <2, nucleosome signal >4, blacklist percentage > 5%, containing <1000 total counts and <500 total features were filtered out, while peaks mapping to blacklisted regions defined by ENCODE (mm10, v2)[134] were removed.

Next, the scRNA or scATAC individual samples were integrated into a common dataset for each omic layer. For the scRNA data, samples were first normalized using the SCT method[154] implemented in Seurat and then integrated within Seurat by finding integration anchors and using the IntegrateData method. To integrate the scATAC data, peaks were first re-quantified across a common set of peaks across all samples. Samples were then normalized using the LSI (TF-IDF followed by SVD) method[155] implemented in Signac and then integrated within Seurat by finding integration anchors and then integrating the embeddings using the "lsi" reductions of the pre-merged samples.

## Joint multimodal single cell data clustering

First, omic-independent clustering was conducted within Seurat: for scRNA, PCA was run across the first 50 components and UMAP was performed on the PCA reduction for the first 30 dimensions (selected by the Elbow method), to find nearest neighbours ($k = 20$) and then identify 29 clusters via shared nearest neighbours (resolution = 0.8, algorithm = 1). For scATAC, UMAP was performed on the "integrated lsi" reduction for the first 30 dimensions excluding the first dimension (highly correlated with sequencing depth), to find nearest neighbours ($k = 20$) and then identify 25 clusters via shared nearest neighbours (resolution = 0.8, algorithm = 3). Using this initial clustering, problematic clusters were identified as those with > 50% cells belonging to the same sample or those depleted with <5% cells of any sample. The cells belonging to these clusters were filtered out and clustering was again performed using the aforementioned parameters.

Next, scRNA and scATAC were integrated into the same dataset by retaining barcodes present across both layers and joint multimodal clustering was performed using the information from both omics, via the weighted-nearest neighbour (WNN) method in Seurat, using the FindMultiModalNeighbors function which combines the PCA scRNA reduction (first 50 dimensions) and the scATAC "integrated lsi" reduction (first 30 dimensions excluding the first). UMAP was subsequently run, and 26 clusters were found via shared nearest neighbour analysis (resolution = 0.8, algorithm = 3).

The agreement between clustering or cell type annotations was measured by computing adjusted rand indices (ARI)[156].

## Cell type annotation of single cell data

To annotate the WNN clusters to cell types, two recent single cell expression atlases were used: a murine brain aging profiling by Ximerakis and colleagues[28] spanning 37,069 cells (accessed through the Single Cell Portal from Broad Institute, https://portals.broadinstitute.org/single_cell) and a single-cell characterization of the mouse isocortex and hippocampal formation by Yao and colleagues[38] across 73,347 cells (accessed through the Allen Brain Atlas database[130]). The use of two separate atlases allowed for the robust and high-confidence calling of cell types and also provided reciprocal validation of the annotated identities.

To achieve this, the atlases were independently used as references to annotate the study dataset by projecting their data structure to the query dataset using transfer anchors and UMAP projection via the MapQuery method within Seurat. Two parallel annotations were thus generated which showed extremely high concordance across major cell types (ARI = 0.97, see Fig. S9b, c). The WNN clusters were then independently annotated to cell types using the two references by assigning a cell type when detected in >75% of the cluster's cells. Mixed-cell clusters were subclustered using Seurat's FindSubCluster method and re-annotated. The two final annotations were then combined to produce a definitive annotation which leveraged information from both atlases (see Supplementary Dataset 25): for instance, oligodendrocyte progenitors were identified with the Ximerakis dataset, while hippocampal region subtypes, not distinguished in this atlas, were resolved with the Yao dataset.

## Differential expression analyses within cell types

Differential gene expression analyses were performed to find aging- and EE-associated changes within major cell types: NEU (CA1; CA3; DG; CR; L5/6, Sub; GABA MGE; GABA CGE), MIG, ASC, OLG, OPC and VASC (containing END, PER and VLM). A negative binomial generalized linear model was used within Seurat's FindMarkers function to test the effect of EE in young, old, or all samples (YC vs YE; OC vs OE; YC + OC vs YE + OE) and the effect of aging in all samples (YC vs OC; YC + YE vs OC + OE). For the comparisons involving all samples, the effect of age or EE was controlled for in the EE and aging comparisons, respectively. Differentially expressed genes (DEGs) were defined as those with logFC >0.25 and Bonferroni-adjusted $p$-value < 0.05 and which were expressed at least in 10% of cells in one of the compared groups.

## Cell type prioritization

The R package Augur (v1.0.3)[94] was used with default parameters to measure and compare the sensitivity of the different major cell types in response to EE and aging by computing "area under the receiver operating characteristic curve" (AUC) values.

## Differential accessibility analyses within cell types

Differential region accessibility analyses were performed to find aging- and EE-associated changes within major cell types: NEU (CA1; CA3; DG; CR; L5/6, Sub; GABA MGE; GABA CGE), MIG, ASC, OLG, OPC and VASC (containing END, PER and VLM). A negative binomial generalized linear model was used within Seurat's FindMarkers function to test the effect

of EE in young, old, or all samples (YC vs YE; OC vs OE; YC + OC vs YE + OE) and the effect of aging in all samples (YC vs OC; YC + YE vs OC + OE). For the comparisons involving all samples, the effect of age or EE was controlled for in the EE and aging comparisons, respectively and additionally, the total number of reads in peaks was included as a covariate. Differentially accessible regions (DARs) were defined as those with logFC >0.25 and Bonferroni-adjusted $p$-value < 0.05 and which were expressed in at least 5% of cells in one of the compared groups.

## Visualization of results

Visualization of results was carried out using IGV (v2.9.4)[157], deepTools (v3.5.0)[138] and the R packages ggplot2 (v3.3.5)[158], ComplexHeatmap (v2.8.0)[159], EnrichedHeatmap (v1.22.0)[160], circlize (v0.4.10)[161], GViz (v1.30.3)[162], ComplexUpset (v1.3.3), eulerr (v6.1.0) and ggalluvial (v0.12.3).

### Reporting summary

Further information on research design is available in the Nature Portfolio Reporting Summary linked to this article.

## Data availability

The data underpinning this article are available in the article and in its Supplementary Material. Additionally, the raw sequencing data have been deposited in the European Nucleotide Archive (ENA) under the following accession numbers: PRJEB58981 (RNA-seq), PRJEB59326 (EM-seq), PRJEB59328 (ATAC-seq), PRJEB59330 (ChIP-seq) and PRJEB59404 (single cell RNA-seq and ATAC-seq). The raw proteomics data have been deposited in the Proteomics Identification Database (PRIDE) under the accession number PXD045567. Finally, preprocessed and extended data sets, including chromatin state annotations and pre-processed single cell data (Seurat objects), are available in a Zenodo repository at https://doi.org/10.5281/zenodo.8372431.

## Code availability

All code underlying this study is publicly available at Zenodo: https://doi.org/10.5281/zenodo.8372431.

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

## Acknowledgements

We are grateful to Ronnie Lendrum for her manuscript editing and to the members of the Cancer Epigenetics Laboratory for their positive contributions.This work was supported by the Spanish Association Against Cancer (PROYE18061FERN to MFF), the Asturias Government (PCTI) co-funding 2018-2023/FEDER (IDI/2018/146 and IDI/2021/000077 to MFF), the Health Institute Carlos III (Plan Nacional de I + D + I) co-funding FEDER (PI18/01527 and PI21/01067 to MFF and AFF), CIBERER Acciones Cooperativas y Complementarias Intramurales (ACCI20-35 to MFF), the Fundación General CSIC (0348_CIE_6_E to MFF), the ISCIII (COV00624 to JRT and MFF), ISPA and the Asociación Galbán (2021-052-INTRAMUR GALBAN-GOURR to RGU), ISPA-Jannsen (2021-048-INTRAMURAL NOV-TEVAR to JRT), ELITE SPORTS 2015 (FUO-19-073 to BFG), CSIC (202020E092 to MFF), and the European Commission NextGenerationEU, through CSIC's Global Health Platform (PTI Salud Global) and

the Spanish Ministry of Science and Innovation through the Recovery, Transformation and Resilience Plan (SGL2021-03-39 and SGL2021-03-040). The laboratory of JVSM is supported by the Spanish Ministry of Science and Innovation (PID-2019-111240RA-I00), and the CSIC Interdisciplinary Thematic Platform (PTI +) NEURO-AGING+ (PTI-NEURO-AGING +). AR is supported by CSIC (SOLAUT_00038505 SGL2103040). JRT is supported by a Ramon y Cajal contract from the Spanish Ministry of Science and Innovation (RYC2021-031799-I). RFP (BP17-114) and PSO (BP17-165) are supported by the Severo Ochoa program. RGU is supported by the Centro de Investigación Biomédica en Red de Enfermedades Raras (CIBERER). JGV is supported by the Spanish Ministry of Universities (FPU20/04659). AGR is supported by the Spanish Ministry of Science and Innovation (PRE2020-093389). We also acknowledge support from the Institute of Oncology of Asturias (IUOPA, supported by Obra Social Cajastur Liberbank, Spain), the Health Research Institute of Asturias (ISPA-FINBA) and Consorcio Centro de Investigación Biomédica en Red (CIBERER-ISCIII).

## Author contributions

R.F.P., A.F.F. and M.F.F. conceived, coordinated, and supervised the study. R.F.P. designed all aspects of research, collected the omic data, performed computational analyses and wrote the manuscript. A.F.F., M.F.F. and J.L.T. participated in drafting the manuscript. P.T., M.M., E.C., P.M. and J.L.T. performed enrichment experiments, histology, immunohistochemistry, and behavioural tests. A.P., R.G.U., J.G.V. and M.R. performed enrichment experiments and behavioural tests. A.G.R. and J.V.S.M. performed the single cell experiments. J.R.T., P.S.O., J.J.A., L.S.L., A.R., V.L., C.M., I.O. analyzed and interpreted the data. A.P. and R.I. performed the MRI experiments. J.C.S., C.T.Z., E.I.G. and B.F.G. assisted in the enrichment experiments and behavioural tests. All authors revised, read, and approved the final manuscript.

## Competing interests

The authors declare no competing interests.

## Additional information

Raúl F. Pérez [1,2,3,4], Patricia Tezanos [5,6], Alfonso Peñarroya[1,2,3], Alejandro González-Ramón[7], Rocío G. Urdinguio [1,2,3,4], Javier Gancedo-Verdejo [1,2,3,4], Juan Ramón Tejedor[1,2,3,4], Pablo Santamarina-Ojeda[1,2,3,4], Juan José Alba-Linares[1,2,3,4], Lidia Sainz-Ledo[1,2,3], Annalisa Roberti[1,2,3], Virginia López [1,2,3,4], Cristina Mangas[1,2,3], María Moro[5], Elisa Cintado Reyes [5,6], Pablo Muela Martínez [5,6], Mar Rodríguez-Santamaría [2,3,8], Ignacio Ortea[2,9], Ramón Iglesias-Rey[10], Juan Castilla-Silgado[2,11], Cristina Tomás-Zapico[2,11], Eduardo Iglesias-Gutiérrez [2,11], Benjamín Fernández-García[2,11], Jose Vicente Sanchez-Mut [7], José Luis Trejo [5], Agustín F. Fernández [1,2,3,4] ✉ & Mario F. Fraga [1,2,3,4,12] ✉

[1]Cancer Epigenetics and Nanomedicine Laboratory, Centro de Investigación en Nanomateriales y Nanotecnología-Consejo Superior de Investigaciones Científicas (CINN-CSIC), Universidad de Oviedo, 33011 Oviedo, Spain. [2]Instituto de Investigación Sanitaria del Principado de Asturias (ISPA-FINBA), Universidad de Oviedo, 33011 Oviedo, Spain. [3]Instituto Universitario de Oncología del Principado de Asturias (IUOPA), Universidad de Oviedo, 33003 Oviedo, Spain. [4]Centro de Investigación Biomédica en Red de Enfermedades Raras (CIBERER), Instituto de Salud Carlos III (ISCIII), 28029 Madrid, Spain. [5]Departamento de Neurociencia Translacional, Instituto Cajal-Consejo Superior de Investigaciones Científicas (IC-CSIC), 28002 Madrid, Spain. [6]Programa de Doctorado en Neurociencia, Universidad Autónoma de Madrid-Instituto Cajal, 28002 Madrid, Spain. [7]Laboratory of Functional Epi-Genomics of Aging and Alzheimer's disease, Instituto de Neurociencias, Universidad Miguel Hernández-Consejo Superior de Investigaciones Científicas (UMH-CSIC), 03550 Alicante, Spain. [8]Bioterio y unidad de imagen preclínica, Universidad de Oviedo, 33006 Oviedo, Spain. [9]Proteomics Unit, Centro de Investigación en Nanomateriales y Nanotecnología-Consejo Superior de Investigaciones Científicas (CINN-CSIC), Instituto de Investigación Sanitaria del Principado de Asturias (ISPA-FINBA), 33011 Oviedo, Spain. [10]Neuroimaging and Biotechnology Laboratory (NOBEL), Clinical Neurosciences Research Laboratory (LINC), Health Research Institute of Santiago de Compostela (IDIS), 15706 Santiago de Compostela, Spain. [11]Departamento de Biología Funcional, Área de Fisiología, Universidad de Oviedo, 33006 Oviedo, Spain. [12]Departamento de Biología de Organismos y Sistemas, Área de Fisiología Vegetal, Universidad de Oviedo, 33006 Oviedo, Spain. ✉e-mail: agustin.fernandez@cinn.es; mffraga@cinn.es

