## [Peer Review File · Nature Communications]

A multiomic atlas of the aging hippocampus reveals molecular changes in response to environmental enrichmentREVIEWER COMMENTS

In this study, Perez et al present a multi-omic atlas of the aging mouse dorsal hippocampus and highlight 3 unique signatures; inflammation, dysregulation of mRNA metabolism and a heterochromatin switch. They used young (9 weeks) and middle-aged (17 months) male mice and performed bulk RNA-seq, proteomics, EM-seq, ATAC-seq, histone modification (H3K4me1, H3K4me3, H3K27ac, H3K36me3, H3K27me3 and H3K9me3) ChIP-seq and single-cell multiome studies to provide an overview of age-related molecular changes. Finally, the authors observed multi-omic rejuvenation of a subset of age-associated alterations when old male mice were exposed to enriched environments (EE). While this study presents an immense body of work, the lack of a central message and novelty slightly dampen enthusiasm. There are additional major concerns I note below:

Major:

1. The following conclusion needs to be clarified – “Taken together, these observations indicate that our study system manifested the typical features of aging and the expected response to the EE paradigm”. From the learning/memory tests done, NOR/NOL did not show any improvement, neither did CFC in the STM phase. EPM is a test of anxiety and EE is known to decrease emotional/anxiety related behavior (reviewed in PMID: 27012955). However, the authors clearly show the opposite. Although the authors try to explain this later, I am unsure whether their EE mice are behaving as expected. Did the authors do a Morris Water Maze in the mice subjected to EE? Relatedly, their 17 month-old mice also had no defects in the NOR/NOL or CFC suggesting their learning and memory are also not impaired. This is a significant concern in this study.
2. The authors attempt to connect the different “omic” layers in their study, but unfortunately a clear combined message is not evident, largely due to the lack of a strong overlap across datasets. Additionally, much of their interpretations are centered on age but not on EE (despite this being shown in many figures early on). It is recommended to show the EE-relevant data only when discussing it in the text (around Fig. 4).
3. In general, this manuscript is overloaded with figure panels. Though I understand the importance of showing everything for the sake of transparency, selected analyses can be presented for clarity.
4. “Using our segmentations, we confirmed how the previously discovered regions of aging-associated heterochromatin switching transition to Polycomb-associated states in old samples, while losing heterochromatic or non-signal domains (Extended Data Fig. S7f).” – this is not evident in the alluvial plots.
5. The enrichment of glial and myelin-related pathways in EE is interesting. Could the authors speculate how EE could be leading to their remodeling?
6. In the Discussion section, the relevant figures for each finding should be indicated.
7. All code used in this manuscript should be made available on github or similar repository.

Minor:

1. In the Extended Abstract – “while uncovering a hitherto undescribed aging-associated heterochromatin-switch phenomenon whereby constitutive heterochromatin loss in the brain is partially mitigated through gains in facultative heterochromatin”. I think this phenomenon has been recently noted in the aging liver and should be cited (PMID: 37116496).
2. Dataset accession numbers in PRIDE and Zenodo are not specified.

3. The comparisons made for statistical tests (for example in Fig. S1, S2) are difficult to understand and lacks consistency. Please clearly indicate all 4 comparisons with single lines over the groups and if not significant mark with a n.s. for all figures. Representative images for immunofluorescence studies in Fig. S1 must be also shown.

4. Can the authors explain why they did not use/include the Ximerakis scRNA-seq dataset for deconvolution?

5. In general, the figure legends need to be expanded so that readers can easily interpret complex analyses. One example is LOLA analysis in multiple figures, where the legend does not expand on the column labels.

6. In the Discussion section, the authors state – “Interestingly, while we had observed that inflammation was a major player in functional aging alterations, the EE-rejuvenation changes were more closely linked to neuronal and glial pathways, suggesting that cognitive stimulation counteracts specific axes of aging dysregulation which do not involve the age-associated brain inflammatory state”. The glial pathways could be related to inflammation, for example, microglia activation.

Reviewer #2 (Remarks to the Author):

In this study, Perez et al present a multi-omic atlas of the aging mouse dorsal hippocampus and highlight 3 unique signatures; inflammation, dysregulation of mRNA metabolism and a heterochromatin switch. They used young (9 weeks) and middle-aged (17 months) male mice and performed bulk RNA-seq, proteomics, EM-seq, ATAC-seq, histone modification (H3K4me1, H3K4me3, H3K27ac, H3K36me3, H3K27me3 and H3K9me3) ChIP-seq and single-cell multiome studies to provide an overview of age-related molecular changes. Finally, the authors observed multi-omic rejuvenation of a subset of age-associated alterations when old male mice were exposed to enriched environments (EE). While this study presents an immense body of work, the lack of a central message and novelty slightly dampen enthusiasm. There are additional major concerns I note below:

Major:

1. The following conclusion needs to be clarified – “Taken together, these observations indicate that our study system manifested the typical features of aging and the expected response to the EE paradigm”. From the learning/memory tests done, NOR/NOL did not show any improvement, neither did CFC in the STM phase. EPM is a test of anxiety and EE is known to decrease emotional/anxiety related behavior (reviewed in PMID: 27012955). However, the authors clearly show the opposite. Although the authors try to explain this later, I am unsure whether their EE mice are behaving as expected. Did the authors do a Morris Water Maze in the mice subjected to EE? Relatedly, their 17 month-old mice also had no defects in the NOR/NOL or CFC suggesting their learning and memory are also not impaired. This is a significant concern in this study.

2. The authors attempt to connect the different “omic” layers in their study, but unfortunately a clear combined message is not evident, largely due to the lack of a strong overlap across datasets. Additionally, much of their interpretations are centered on age but not on EE (despite this being shown in many figures early on). It is recommended to show the EE-relevant data only when discussing it in the text (around Fig. 4).

3. In general, this manuscript is overloaded with figure panels. Though I understand the importance of showing everything for the sake of transparency, selected analyses can be presented for clarity.

4. “Using our segmentations, we confirmed how the previously discovered regions of aging-associated heterochromatin switching transition to Polycomb-associated states in old samples, while losing heterochromatic or non-signal domains (Extended Data Fig. S7f).” – this is not evident in the alluvial plots.

5. The enrichment of glial and myelin-related pathways in EE is interesting. Could the authors speculate how EE could be leading to their remodeling?
6. In the Discussion section, the relevant figures for each finding should be indicated.
7. All code used in this manuscript should be made available on github or similar repository.

Minor:

1. In the Extended Abstract – “while uncovering a hitherto undescribed aging-associated heterochromatin-switch phenomenon whereby constitutive heterochromatin loss in the brain is partially mitigated through gains in facultative heterochromatin”. I think this phenomenon has been recently noted in the aging liver and should be cited (PMID: 37116496).
2. Dataset accession numbers in PRIDE and Zenodo are not specified.
3. The comparisons made for statistical tests (for example in Fig. S1, S2) are difficult to understand and lacks consistency. Please clearly indicate all 4 comparisons with single lines over the groups and if not significant mark with a n.s. for all figures. Representative images for immunofluorescence studies in Fig. S1 must be also shown.
4. Can the authors explain why they did not use/include the Ximerakis scRNA-seq dataset for deconvolution?
5. In general, the figure legends need to be expanded so that readers can easily interpret complex analyses. One example is LOLA analysis in multiple figures, where the legend does not expand on the column labels.
6. In the Discussion section, the authors state – “Interestingly, while we had observed that inflammation was a major player in functional aging alterations, the EE-rejuvenation changes were more closely linked to neuronal and glial pathways, suggesting that cognitive stimulation counteracts specific axes of aging dysregulation which do not involve the age-associated brain inflammatory state”. The glial pathways could be related to inflammation, for example, microglia activation.

Reviewer #3 (Remarks to the Author):

The manuscript titled “Molecular Rejuvenation of the Hippocampus in Enriched Environments” by Pérez et al. investigates the molecular rejuvenation of the hippocampus in mice exposed to enriched environments, focusing on epigenetic mechanisms underlying aging and their modulation by environmental factors. The authors find that environment enrichment can induce a multi-omic rejuvenation of aging-associated alterations, evidenced by the reversal or mitigation of many aging-related changes in gene and protein expression, as well as epigenomic features, in both young and old mice. The research is methodologically robust, integrating multi-omic data from bulk tissue and single-cell analyses to offer a comprehensive view of aging-related alterations and their reversal through environmental enrichment (EE). The study's strengths are its broad and in-depth analysis of various molecular layers, including gene and protein expression, DNA methylation, chromatin accessibility, and histone modifications, both in bulk tissue and at the single-cell level.

However, there are a few concerns I hope the authors address in a revision:

1. Based on the methodology, it is difficult to distinguish the effects of an enriched environment from simple increased levels of physical and/or mental activity. Some consideration or controls for the effects of physical activity would enhance this study.
2. Most of the "rejuvenation effects" observed with EE in old animals, have similar or even stronger effects in young animals. Although these are age-affected genes, it would not make sense to say that EE "rejuvenates" young animals or EE have stronger rejuvenation effects in young animals than in old animals. Instead of emphasizing rejuvenation, the data suggest that EE has stronger beneficial effects in young animals than in old animals.

Due to lack of mechanistic investigation, this manuscript is better suited for publishing as a resource paper.

RESPONSES TO THE REVIEWERS

General comments:

In the following response letter we provide point-by-point responses to the reviewers' comments. Our responses are indicated in blue text, while we use **red text** to refer to specific changes in the manuscript. In the manuscript document itself, text changes are also indicated with **red text**. Broadly, the major changes made include:

- Extensive modifications to the text, including key sections, such as a complete change of the study Title and Abstract or extensive modifications to Introduction, Discussion.
- Extensive modifications to the existing figures, including the simplification or removal of 11 panels.
- New data analyses (bulk cell type deconvolution, comparisons of EE in young and old).
- New experiments (Barnes maze tests).
- Incorporation of all code used across all analyses (>100 scripts), accompanied by documentation, to a custom Zenodo repository.
- Minor corrections of typos.

REVIEWER COMMENTS

Reviewer #2 (Remarks to the Author):

In this study, Perez et al present a multi-omic atlas of the aging mouse dorsal hippocampus and highlight 3 unique signatures; inflammation, dysregulation of mRNA metabolism and a heterochromatin switch. They used young (9 weeks) and middle-aged (17 months) male mice and performed bulk RNA-seq, proteomics, EM-seq, ATAC-seq, histone modification (H3K4me1, H3K4me3, H3K27ac, H3K36me3, H3K27me3 and H3K9me3) ChIP-seq and single-cell multiome studies to provide an overview of age-related molecular changes. Finally, the authors observed multi-omic rejuvenation of a subset of age-associated alterations when old male mice were exposed to enriched environments (EE). While this study presents an immense body of work, the lack of a central message and novelty slightly dampen enthusiasm. There are additional major concerns I note below:

We are thankful to the reviewer for the effort put into reviewing our manuscript. We are also grateful that the reviewer acknowledges the size of the work presented.

Before proceeding to the point-by-point response, we would like to first give a general response which addressed concerns raised by all the reviewers, because we have made changes to the manuscript which considerably change the perspective and conceptual approach of the paper.

We believe that we have not correctly presented and written the manuscript to reflect the following ideas:

- a. The first goal of our study was **to study the epigenomics of brain aging, by generating an epigenomic landscape resource** at a scale which was unprecedented in the field.
- b. The second goal of the study was **to explore if any lifestyle/environmental stimulation (in general) could particularly, or specifically, target aging-affected pathways**, or not.

Thus, we selected EE as a “generic” lifestyle/environmental stimulation because: 1) we know it leads to identifiable molecular changes; 2) the brain is a tissue in which epigenetic marks are known to play relevant roles; 3) it is possibly a more “diffuse” stimulus which is in line with the concept of lifestyle as a complex accrual of a diverse set of stimuli (Queen et al. 2020, DOI: 10.3389/fnins.2020.00605). Thus, EE, or studying the precise nature of EE, was not the main focus of this study nor was this study specifically designed for this. In fact, our long-term future plan is to carry out studies looking to try diverse specific stimuli (e.g. specific types of physical activity or mental activity) and then to compare their molecular signatures.

It is evident that we have not clearly conveyed these ideas, so we have rewritten key parts of the text and hope that now the scope of the study and the questions asked (and hopefully answered) are more apparent: 1) that we present a multiomic resource and 2) that the focus is to present the epigenomic landscape of aging and the impact of “environmental stimulation”, and not so much to specifically study the EE paradigm.

In particular, we have changed the Title of the manuscript, which now reads as: *“A multiomic atlas of the aging hippocampus reveals molecular rejuvenation in response to environmental stimulation”* and better encapsulates the goal and design of the study.

- We have rewritten the Abstract (see revised manuscript) and part of the Introduction to accommodate the aforementioned ideas (see revised manuscript), including this last paragraph in the Introduction, in which we also mention physical stimulation and how the EE will produce a general stimulation:

[...]

In this study, we have generated a molecular atlas of the murine dorsal hippocampus encompassing gene and protein expression, DNA methylation, chromatin accessibility, histone modifications, and single cell expression and accessibility. We have characterized young and aged mice which were subjected to lifestyle stimulation in the form of environmental enrichment (EE). We have focused on the dorsal hippocampus because it is an important target of both cognitive and physical stimulus where adult neurogenesis occurs^{18,19} and it is known to suffer aging-associated decline linked to cognitive deterioration²⁰. On the other hand, the EE paradigm is a well-established system of general lifestyle stimulation (both cognitive and physical) linked to hippocampal changes at the cellular and molecular level²¹. Thus, with this in-depth map of aging and EE at both the bulk-tissue and single-cell level we have aimed to explore the molecular alterations associated with aging and with environmental stimulation, and also their putative interactions.

- We have also modified parts of the Discussion: we have added a paragraph discussing the limitations of the model used, in terms of it being an unspecific mixture of physical and cognitive (and also social) stimulation:

[...]

To tackle these issues, here we set up a murine model of hippocampal aging which was stimulated by a medium-term lifestyle intervention based on environmental enrichment. The EE paradigm used involves an unspecific stimulation which entails alterations in physical, cognitive, and social activity^{95,21}. As such, the molecular changes observed in this study comprise a mixture of stimuli which cannot be narrowed down to specific behavioural pathways, and better-controlled interventions should be used to dissect the different molecular signatures attributable, for example, to exercise or cognition⁹⁶. Nonetheless, here we have used EE as a laboratory proxy for general lifestyle or environmental stimulation, which is indeed a very complex phenomenon in humans but for which EE is frequently used as a model⁹⁷. We extensively characterized the molecular dynamics of this model by profiling multiple layers of epigenomic regulation, thus generating a molecular map resource to aid understanding of these processes.

[...]

With regards to the concerns in relation to the lack of a central message or novelty, we would like to restate that a key aspect of the work is the generation of the multiomic resource, which, including the single cell multiomic data, represents a dataset of unprecedented depth both in the field of aging epigenetics or neurobiology epigenetics. The generation of such a dataset does necessitate a comprehensive analysis which we believe we have conducted, though further mechanistic exploration is out of the scope of this already sizable study. We would say that the central message of the manuscript is that we observe that environmental stimulation in mice specifically targets aging-related pathways, across most of the omic layers (including single cell). Novel results, aside from the data generation, to us include the aforementioned observation, the chromatin alterations found in aging (e.g. a bidirectional interactions between chromatin marks, erosion of bivalent domains, chromatin switching), the specific targeting of glial/oligodendrocyte pathways in EE and rejuvenation, and the general lists of candidate age-reversed genes with evidence from multiple molecular layers.

Major:

1. The following conclusion needs to be clarified – “Taken together, these observations indicate that our study system manifested the typical features of aging and the expected response to the EE paradigm”. From the learning/memory tests done, NOR/NOL did not show any improvement, neither did CFC in the STM phase. EPM is a test of anxiety and EE is known to decrease emotional/anxiety related behavior

(reviewed in PMID: 27012955). However, the authors clearly show the opposite. Although the authors try to explain this later, I am unsure whether their EE mice are behaving as expected. Did the authors do a Morris Water Maze in the mice subjected to EE? Relatedly, their 17 month-old mice also had no defects in the NOR/NOL or CFC suggesting their learning and memory are also not impaired. This is a significant concern in this study.

The comment raised by the reviewer is important. We agree that the sentence “[...] typical features of aging and the expected response to the EE paradigm” is not adequately positioned into the paragraph and in its present form reflects the set of results with both MRI hippocampal volume, immunohistochemistry and behavior. We also agree on the need to better contextualize our results and justify why we refer to them as “expected” given the heterogeneity of effects in mouse behavior reported in the literature after EE protocols. For that, we have included a brief argumentation here and added the reformulated paragraph at the end:

As for anxiety, it is well known that not all EE protocols induce a beneficial effect on anxiety for all mouse strains and their effects depend heavily on sex. Specifically, it may well induce increased aggressiveness and anxiety (see for example Nevison et al. *Animal Welfare* 1999, DOI: 10.1017/S0962728600021989), Marashi et al. *Horm Behav* 2003, DOI: 10.1016/s0018-506x(03)00002-3). Moreover, as the work cited by the reviewer in their comment (DOI: 10.2174/1570159X14666160325115909) adequately addresses, the enrichment-induced anxiolytic outcome is only “expected” in challenging situations consisting of exposure to intense stressors, not under more ethological stressors (Sampedro-Piquero & Begega, *Current Neuropharmacol* 2017, DOI: 10.2174/1570159X14666160325115909). Not less relevant is that in C57BL/6J (the strain used in the present work) the novelty versus complexity paradigm while using cage enrichment is not adequately resolved (Bohn et al. *Front Vet Sci* 2023, DOI: 10.3389/fvets.2023.1207332), and we have chosen the “novelty” model because is the best suited to generate neural plasticity according with our previous results. In this line, the concept of neural plasticity induced by enrichment is the milestone: the relevant aspect is that the cellular activation and plasticity (measured by a battery of molecular markers) induced by EE in the brain (and specially in hippocampus) is independent of the anxiety outcome (see for example Lin et al. *Behav Brain Res* 2011, DOI: 10.1016/j.bbr.2010.08.019). All together means that novelty enrichment in a cage is very efficient to induce neural activity and plasticity (as we have demonstrated here, see Figure 4B), although the anxiety trait may well be increased (especially when using male mice) because of increased aggressiveness, a trait completely compatible with increased c-fos or Bdnf hippocampal expression.

In addition, we would like to bring attention to the fact that, because we were well-aware of the heterogeneity of effects of EE in the literature, and to demonstrate the robustness of our model/results, we went as far as to perform independent enrichment experiments in a different laboratory. As shown in Figure S2b-c, which showcases EPM tests done in independent laboratories/enrichments, our observation is very reproducible (especially considering the well-known variability of behavioral tests in mice with low N):

Furthermore, regarding the molecular effects, we also performed RNA-seq in this independent experiment, and also observed similar reversal signatures between EE and aging, as is shown in Figure S8d,f:

As for cognition, many authors agree that the enrichment-induced improvements in cognition are a consequence of the reduction of anxiety when it happens (see for example Harris et al. *Animal Behav* 2009, DOI: 10.1016/j.anbehav.2009.02.019) as a consequence of an associated increase in exploration/evaluation capabilities, so it would be expected that in the absence of an anxiolytic outcome after a specific EE protocol (novelty model, use of male mice, specific timing schedule used -see below-, etc...) no such effect may be found. Most of the studies reporting enrichment-induced improvements in Morris water maze, using C57BL/6J, have used animals reared from weaning in EE (revised by Hendershott et al. *Behav Brain Res* 2016, DOI: 10.1016/j.bbr.2016.08.004), not starting with young adults as in the present work, or studies in rats (Harris et al., *op.cit.*).

Whatever the case, we also performed here a Barnes maze (not shown in the first version of the manuscript) to measure spatial navigation in our experimental design. No main differences were found in the latency to exit in training or test phases neither, and a modest, non-significant difference was noted between young vs old animals and between control vs enriched cages (in the number of failures in finding the exit across all trials), so we didn't include these results, but if considered interesting taking into account the reviewer's comments about MWM, it can be included now. Here are the figures related to the Barnes maze results:

Figure Barnes. **a**, Line plots for the Barnes maze tests indicating, across groups, the mean times to exit during the consecutive training sessions across day 1 (3 sessions), day 2 (24 h later, 3 sessions) and final test (72 h later, 1 session). **b**, Boxplots for the Barnes maze tests, representing, across groups, the difference in time to exit between the last training session (day 2, session 3) and the test session (~ 72 h later). **c**, Bar plots for the Barnes maze tests, showing, across groups, the numbers of failed exits considered as > 120 seconds to exit, combined across all sessions. * $p < 0.05$, ** $p < 0.01$, *** $p < 0.001$ for Wilcoxon rank sum tests or Fisher's exact test. P -values are adjusted for multiple testing within sets comparisons. Outliers above 2 % quantiles (or 5 % for line plots) are not shown in graphs.

Finally, the discrepancy in our results assessing cognition and aging with that of others may well be due to differences in the protocols used. As an example, Wimmer et al. Neurobiol Aging 2012 (DOI: 10.1016/j.neurobiolaging.2011.07.007) found no differences between young (4 months old) and aged (24 months old) mice in NOR (as in the present work), while a worst performance in old animals was found in NOL. Probably our NOL protocol was not difficult enough to detect this aging-induced drop in cognitive performance in our 19 months old mice.

To address all of this concerns, we have now rephrased the mentioned sentence as follows: "*Taken together, these observations indicate that our study system manifested the typical features of aging and the expected response to the EE paradigm in a number of structural and morphological parameters, together with an enrichment-induced behavioral outcome consisting of increased anxiety and consequent absence of differences in cognition, frequent in novelty enrichment protocols using male C57BL/6J mice.*"

Nevertheless, we can, if considered adequate, include all the discussion above into the manuscript, or the new Barnes maze results.

(Also, as previously mentioned, we have added text in Discussion to address the limitations of the model used).

2. The authors attempt to connect the different "omic" layers in their study, but unfortunately a clear

combined message is not evident, largely due to the lack of a strong overlap across datasets. Additionally, much of their interpretations are centered on age but not on EE (despite this being shown in many figures early on). It is recommended to show the EE-relevant data only when discussing it in the text (around Fig. 4).

- Regarding the lack of overlap across the omics: we think that this is partly to be expected. It is known that the different molecular layers are partly independent and often show low levels of correlation, e.g. between gene and protein expression (see Wang et al. Proteomics 2014, DOI: 10.1002/pmic.201400184, Takemon et al. Elife 2021, DOI: 10.7554/eLife.62585) or between gene expression and epigenetic marks such as DNA methylation (see Blake et al. Genome Res 2020, DOI: 10.1101/gr.254904.119). In this sense, we believe that the major finding in our study is that we observe an independent aging rejuvenation/reversal in most omic layers (including single cell), particularly within themselves.

Indeed, to explore this idea, throughout the study we performed multiple permutation or enrichment analyses to measure the degree of overlap between omic layers, against what was expected by chance. These analyses have the strength of being agnostic or quite unbiased, and we preferred to give more importance to these kinds of more general results rather than overly focus on cherry-picking sets of genes (of which there will always be candidates) in order to not be too misleading; because it is true that the omic layers are quite independent but also that the rejuvenation occurs within most. To better clarify these ideas we have added the following text to Discussion:

[...]

Thus, our results indicate that EE induces a partial, multi-omic reversal of the aging phenotype in the murine dorsal hippocampus. It must be noted that the nature of this reversal appears to be mostly independent across the different omic layers. Indeed, it is known that different molecular layers often display low levels of correlation^{43,116}, although we did observe significant intersections across several inter-omic changes (see for instance Extended Data Fig. S4h, Fig. 3b-c or Fig. 5d). Nonetheless, these observations point towards the existence of an omic-specific response of rejuvenation in response to environmental stimulation.

[...]

- With respect to the aging- versus EE-relevant data presentation: we acknowledge that many of our interpretations are focused on the aging perspective. We think that the extensive rewriting done to the manuscript (see previous answers) now better reflects the fact that this characterization is a major goal in the design of the study, with EE representing a proxy for environmental/lifestyle stimulation. The reasons for showing EE data from the start are the following: first, because we start by presenting the whole model/resource (we have also updated Fig. 1a in response to the second reviewer to reflect this), but, more importantly, because the whole study design is embedded into many of the statistical models used across the paper. In this sense, the use of all samples allowed us to gain more statistical power and achieve more robust results throughout the study. Of course, these statistical models always correct for/adjust the fact that there are different experimental groups (i.e. control or enriched, young or old; see Methods), and we also validate comparisons using only non-enriched/control samples. Thus, the use of the whole dataset is very important for the robustness of many of the analyses so we have maintained this aspect of the study. To try to state this more clearly, we have added the following

in

Results:

[...]

For subsequent analyses, we made use of all the study samples in order to increase the statistical power (Methods). We performed differential expression analyses [...]

3. In general, this manuscript is overloaded with figure panels. Though I understand the importance of showing everything for the sake of transparency, selected analyses can be presented for clarity.

We understand the concern raised by the reviewer. The quantity and nature of the generated data resources warranted in-depth analyses, but it is true that the presentation of the results may have become too dense. To address this, we have modified or removed the following panels:

1. Removed Fig. 1c (bar plots describing DEG numbers); now only in text.
2. Removed supplementary Figs. 3a and 3b (these PCAs were slightly redundant with the PCA from Fig. 1b and referred to more EE-centric results rather than aging insights).
3. Moved Fig. 1f to supplementary (describing deconvolution results); is now part of Extended Data Fig. S4a).
4. Removed Fig. 1f (bar plots describing DEP numbers); now only in text.
5. Removed Fig. 2b (bar plots with DMR numbers); now only in text.
6. Removed Fig. 2g (bar plots with DAR numbers); now only in text.
7. Collapsed Fig. 2e-f into the same figure (now is Fig. 2d)
8. Removed Extended Data Fig. 6e (describing p-value histograms for the differential enrichment analyses; this information can be accessed from the supplementary table 13)
9. Moved Fig. 3b to supplementary (describing numbers of DERs); is now Fig. S6e.

We hope to have reduced the density of the main figures of the paper, but we are open to removing more panels if the reviewer feel that it is necessary.

4. "Using our segmentations, we confirmed how the previously discovered regions of aging-associated heterochromatin switching transition to Polycomb-associated states in old samples, while losing heterochromatic or non-signal domains (Extended Data Fig. S7f)." – this is not evident in the alluvial plots.

We acknowledge that the message needs to be revised. The alluvial bar plots show the composition in chromatin states of the regions of switching chromatin detected in prior analyses. Initially, we equated "NS" (non-signal) and "Hc-H" (heterochromatin, H3K9me3 associated) because the low-signal chromatin states have been typically defined as quiescent or inactive in the field (see for instance the landmark paper by the Roadmap Consortium: [10.1038/nature14248](https://doi.org/10.1038/nature14248)). Nonetheless, there is no direct demonstration of the fact that this quiescence is directly linked to H3K9me3 heterochromatin, and thus we have reworded that section of the results to simply state that we observe and increase in Polycomb-associated states at these regions. The section now reads as follows:

[...] Using our segmentations, we observed how the previously discovered regions of aging-associated heterochromatin switching present an increase in Polycomb-associated chromatin states in old samples (Extended Data Fig. S7f).

We have also reworded the legend for the Fig. S7f in the hope of making it more clear:

f, Alluvial diagram showing the chromatin state annotations (200 bp genomic bins) at the previously characterized regions of aging-associated heterochromatin switching. The lines connecting the bar plots show the transitions in chromatin state annotations between the experimental groups.

5. The enrichment of glial and myelin-related pathways in EE is interesting. Could the authors speculate how EE could be leading to their remodeling?

We agree with the reviewer that this is an interesting finding. There are studies in the literature linking behavioural stimulation and myelin changes (reviewed in [10.1016/j.conb.2017.09.014](https://doi.org/10.1016/j.conb.2017.09.014)), so that it is thought that there exist, in part, myelination processes which are dependent on, or influenced by,

neuronal activity/experience (reviewed in 10.1111/jnc.14592). For example, myelination has been shown to be required for motor skill learning (10.1126/science.1254960, 10.1523/JNEUROSCI.3048-13.2013) while neuronal activity has also been shown to promote oligodendrogenesis (10.1126/science.1252304).

Thus, we believe our results are in line with the existing literature and would respond on 2 main axes:

- First, that it makes sense EE, being a complex stimulation which doubtless involves both cognitive/neuronal and physical/motor activity changes, has an impact on glial/myelin pathways as has been previously shown
- Second, more in the line of speculation, that: there are studies showing that glial plasticity is maintained through the majority of the murine lifespan (10.1038/s41593-018-0120-6). Thus, we could hypothesize that the glia component may be more plastic and targetable by environmental stimulation, specially at middle-/old-ages than the neural component of the brain.

We have added the following text section to Discussion regarding the mentioned ideas:

[...]

Myelin function is known to be influenced by both neuronal and physical activity^{110,111}, hence it makes sense that the complex stimulation brought on by EE could have an impact on these pathways. Furthermore, there is evidence that glial cell generation is maintained throughout a great fraction of the murine lifespan (up to two years of age)¹¹², so it is possible that, especially for middle- or old-aged subjects, the glial component is more targetable by environmental stimulation than the neural component of the brain.

[...]

6. In the Discussion section, the relevant figures for each finding should be indicated.

We thank the reviewer for the comment. We have now indicated the relevant figures for each finding in the Discussion section significantly improving the clarity and coherence of the text.

7. All code used in this manuscript should be made available on github or similar repository.

We have now uploaded all of the code used in the manuscript to the custom Zenodo repository which accompanies the manuscript. The code consists of a collection of more than 100 scripts for which we have drafted an extensive documentation explaining their uses and outputs.

This is now indicated in the Code Availability section as (also with updated PRIDE and Zenodo accessions in the Data Availability section):

DATA AVAILABILITY

The data underpinning this article are available in the article and in its Supplementary Material. Additionally, the raw sequencing data have been deposited in the European Nucleotide Archive (ENA) under the following accession numbers: PRJEB58981 (RNA-seq), PRJEB59326 (EM-seq), PRJEB59328 (ATAC-seq), PRJEB59330 (ChIP-seq) and PRJEB59404 (single cell RNA-seq and ATAC-seq). The raw proteomics data have been deposited in the Proteomics Identification Database (PRIDE) under the accession number PXD045567. Finally, preprocessed and extended data sets, including chromatin state annotations and preprocessed single cell data (Seurat objects), are available in a Zenodo repository at 10.5281/zenodo.8372432.

CODE AVAILABILITY

All code underlying this study is publicly available at Zenodo: 10.5281/zenodo.8372432.

Minor:

1. In the Extended Abstract – “while uncovering a hitherto undescribed aging-associated heterochromatin-switch phenomenon whereby constitutive heterochromatin loss in the brain is partially mitigated through gains in facultative heterochromatin”. I think this phenomenon has been recently noted in the aging liver and should be cited (PMID: 37116496).

We greatly thank the reviewer for pointing out this paper. It is indeed recent, and we may have missed it during the preparation of our manuscript, but it should be cited and discussed in our work (our findings are also strongly reinforced by it). The main difference in our study system is that, in brain, we do find abundant bidirectional changes of H3K27me3, and not a generalized loss, as observed by Yang et al (10.1016/j.molcel.2023.04.005).

First, we have removed expressions such as “hitherto undescribed” (Abstract) or “has not, to our knowledge, been previously described” (Discussion) from our text.

Second, we have cited and discussed this study in Discussion:

[...]

This phenomenon of “chromatin switching”, which has some parallels with certain experimental models^{77,76,78}, has very recently been described in the aging liver¹⁰⁴. Whereas Yang N. and colleagues reported a generalized H3K27me3-associated heterochromatinization with aging¹⁰⁴, here we describe bidirectional H3K27me3 changes displaying very specific genomic characteristics: loss of H3K27me3 targets bivalent domains whereas gains of this mark occurs at regions of age-associated H3K9me3 loss. We hypothesize that in the brain this latter phenomenon could be caused by the cellular re-repression, using facultative heterochromatin, of the constitutive heterochromatin loss during aging.

[...]

2. Dataset accession numbers in PRIDE and Zenodo are not specified.

We apologize for the oversight. We have now corrected the missing accession numbers, including the updated Zenodo repository with the code in the Data Availability section.

3. The comparisons made for statistical tests (for example in Fig. S1, S2) are difficult to understand and lacks consistency. Please clearly indicate all 4 comparisons with single lines over the groups and if not significant mark with a n.s. for all figures. Representative images for immunofluorescence studies in Fig. S1 must be also shown.

- We apologize for the lack of clarity regarding figures S1 and S2. We have now standardized all of the tests: the figures now show the same 4 comparisons for across all boxplots (age, EE, EE in young, EE in old) using the same non-parametric test (Wilcoxon Rank Sum) and, for each set of comparisons, the p-values are adjusted for multiple testing.
- We now include representative immunofluorescence images for the experiments, which are shown in Fig. S1i.

4. Can the authors explain why they did not use/include the Ximerakis scRNA-seq dataset for deconvolution?

Yes, the reasons for this are the following:

- Initially, we used the Zeisel (~3,000 cells from 2015) and Yao (~75,000 cells from 2021) for bulk RNA deconvolution into major cell subtypes (basically to distinguish neurons and glial types, mainly to check if there were major changes in glia or neurons with aging). We used 2 datasets to cross-validate the findings of both.
- Then, when turning to annotate the single cell data, we first sought to use only the larger dataset (Yao) because it was much more recent and more comprehensive: our own dataset contained ~16,000 cells so it was by itself much deeper than the one from Zeisel. Nonetheless, we still wanted to use 2 datasets for single cell annotation to ensure robustness, so we looked for a newer, sizable dataset and thus used the Ximerakis (~37,000 cells) data. In addition, as discussed in the manuscript, Ximerakis provided annotations which were complementary to the Yao dataset (e.g. oligodendrocyte progenitors).

So that's the reason why, not so much that we didn't use Ximerakis for deconvolution, but rather that we didn't use Zeisel for single cell annotation.

Regardless, we have performed here a new deconvolution analysis using the Ximerakis dataset, to observe the following result: there are again no major changes in non-neuronal or neuronal populations, and the results are similar as to those obtained with Yao or Zeisel (in fact, the recovery of non-neuronal populations at the level of bulk data is slightly worse with the Ximerakis dataset than with Yao or Zeisel). For now, we have not included the third dataset in the final manuscript but will do so if the reviewer feels that it is necessary. Here are the results of the Ximerakis deconvolution:

5. In general, the figure legends need to be expanded so that readers can easily interpret complex analyses. One example is LOLA analysis in multiple figures, where the legend does not expand on the column labels.

The reviewer is completely right, we have expanded the following Figure legends to explain the labels:

- For main Figure 2j, we have added: *The code for the chromatin states shown is: Promoter, Active (Pr-A), Weak (Pr-W), Bivalent (Pr-B) and Flanking (Pr-F); Transcription, Strong (Tr-S), Permissive (Tr-P) and Initiation (Tr-I); Enhancer, Strong TSS-distal (En-Sd), Strong TSS-proximal (En-Sp), Weak (En-W), Poised TSS-distal (En-Pd) and Poised TSS-proximal (En-Pp); Heterochromatin, Polycomb-associated (Hc-P) and H3K9me3-associated (Hc-H); No significant signal (NS).*
- For supplementary Figures S5, S6 and S7d-f, we have added to the figure legends: *The code for the chromatin states learnt in this study is: Promoter, Active (Pr-A), Weak (Pr-W), Bivalent (Pr-*

B) and Flanking (Pr-F); Transcription, Strong (Tr-S), Strong 2 (Tr-S2), Permissive (Tr-P) and Initiation (Tr-I); Enhancer, Strong TSS-distal (En-Sd), Strong TSS-proximal (En-Sp) and Poised TSS-distal (En-Pd); Heterochromatin, Polycomb-associated (Hc-P), Polycomb-associated weak (Hc-Pw) and H3K9me3-associated (Hc-H); No significant signal (NS). The code for the ENCODE chromatin states shown is: Promoter, Active (Pr-A), Weak (Pr-W), Bivalent (Pr-B) and Flanking (Pr-F); Enhancer, Strong TSS-distal (En-Sd), Strong TSS-proximal (En-Sp), Weak (En-W), Poised TSS-distal (En-Pd) and Poised TSS-proximal (En-Pp); Transcription, Strong (Tr-S), Permissive (Tr-P) and Initiation (Tr-I); Heterochromatin, Polycomb-associated (Hc-P) and H3K9me3-associated (Hc-H); No significant signal (NS).

- In main text, we now make clearer reference to using the public ENCODE3 datasets as: [...] we performed Locus Overlap Analysis (LOLA, FDR < 0.05, Methods) on the aging-DARs (Supplementary Table 11) using public ENCODE3 datasets⁷² [...]
- Additionally, we have included in the Zenodo repository a metadata table (5_CHIP_chromatin_states_metadata.txt) explaining the chromatin state annotations, as well as the full dataset with our annotated chromatin states (5_CHIP_chromatin_states.qc).
- For main Figure 6, we attached a supplementary table 25 describing the cell annotations, and for supplementary Figure S9c, we have added to the figure legend the cell type annotations used in the Yao and Ximerakis datasets: *The code used for the Yao cell type annotations is: DG (Neuron, glutamatergic, dentate gyrus), CA1-ProS (Neuron, glutamatergic, CA1, Prosubiculum), CA2-IG-FC (Neuron, glutamatergic CA2 IG or FC regions), CA3 (Neuron, glutamatergic, CA3), SUB-ProS (Neuron, Subiculum, Prosubiculum), NP SUB (Neuron, near-projecting, subiculum), NP PPP (Neuron, near-projecting, subiculum related), L5 PT CTX, L5 PPP and L5/6 NP CTX (Neuron, glutamatergic, Cortical layer 5 or 6, pyramidal tract, cortex (CTX) or subiculum (PPP) related), L6 CT, L6b CTX and L6b/CT ENT (Neuron, glutamatergic, Cortical layer 6 or 6b pyramidal corticothalamic, cortex or ENT related), L2/3 IT RHP (Neuron, glutamatergic, intratelencephalic related, retrohippocampal region), Sst and Pvalb (Neuron, GABAergic, medial ganglionic eminence (MGE) origin, Sst or Pvalb marker), Vip, Sncg and Lamp5 (Neuron, GABAergic, caudal ganglionic eminence (CGE) origin, Vip, Sncg or Lamp marker), CR (Neuron, Cajal-Retzius), Oligo (Oligodendrocyte), Astro (Astrocyte), Endo (Endothelial cell), SMC-Peri (Smooth muscle/pericyte), VLMC (vascular/leptomeningeal cell), PVM (microglia/perivascular macrophage). The code used for the Ximerakis cell type annotations is: mNEUR (Neuron), OLG (Oligodendrocyte), OPC (Oligodendrocyte progenitor), ASC (Astrocyte), TNC (tanocyte), CPC (choroid plexus epithelial cell), VLMC (vascular/leptomeningeal cell), PC (pericyte), ABC (arachnoid barrier cell), DC (dendritic cell), MAC (macrophage), MNC (monocyte), NEUT (neutrophil), EC (endothelial cell), MG (microglia).*

6. In the Discussion section, the authors state – “Interestingly, while we had observed that inflammation was a major player in functional aging alterations, the EE-rejuvenation changes were more closely linked to neuronal and glial pathways, suggesting that cognitive stimulation counteracts specific axes of aging dysregulation which do not involve the age-associated brain inflammatory state”. The glial pathways could be related to inflammation, for example, microglia activation.

We thank the reviewer for the comment. We have made changes to the Discussion section to revise and further clarify these ideas. These results come from pathway enrichment analyses, and we must state that:

1. Pathway enrichment analyses always give indicative but not definitive results.
2. To be able to get the most precise and robust results possible, we interrogated multiple databases, which have different collections of pathways, focused on different processes.
3. Particularly by including the CGP (very specific pathways from other papers) GOBP (general pathways), ImmuneSigDB (specific inflammation pathways) and C8 (specific cell type signatures), we believe that we were in position to detect different types of inflammatory or other signatures (at least to the extent possible with pathway enrichment results).

Having said that, the results that we observed were:

1. When looking at our pathway enrichment results for aging (Fig. 1d, Fig. S3e-I and Table S3), it is very clear that aging leads to a lot of inflammatory enrichments. These terms dominate the enrichment results:
 - a. Unspecific inflammatory pathways:
 - i. seen in Fig. 1d with GO terms linked to "immunity", "leukocyte" etc.
 - ii. seen in the thousands of significant terms for the ImmuneSigDB (Fig. S3e)
 - b. Specific brain inflammation pathways involving microglia: seen in Fig. S3g with WikiPathway terms linked to "microglia", and, indirectly, "macrophage". Also as seen in Fig. S3h, the top cell type signatures (C8 database) are for "microglia". There are also significant "microglia" pathways for GO.
2. On the other hand, the pathway enrichment results for EE-reversal (Fig. S8I, Table S22) are related to "glia" but particularly to "oligodendrocyte" or "gliogenesis". And there are no clear enrichments for inflammatory, or specifically "microglia" pathways. This is especially seen when looking at Table S22:
 - a. The top cell type signatures (C8 database) are for cortex, oligodendrocyte (or oligo progenitors). No significant pathways for microglia.
 - b. The top CGP pathways indicate specifically oligodendrocyte markers, the 2nd top WikiPathway is related to oligodendrocytes, not microglia, top GO pathways are related specifically to myelin and not microglia.
 - c. In general, inflammatory terms or microglia specific do not dominate the enrichment results, and, furthermore, there are no significant results at all for the ImmuneSigDB database (table S22).

Taking all these observations together, we believe that, as a whole, there is no strong evidence of inflammation signatures of any type (unspecific), but also not of specific microglial inflammation, in the EE-reversal, while there is evidence of both in aging.

We have extended that part of Discussion to better convey these ideas:

Interestingly, while we had observed that inflammation was a major player in functional aging alterations, the EE-rejuvenation changes were more closely linked to neuronal and glial pathways (Extended Data Fig. S8I), with no C8 cell type enrichments related to microglia being observed (Supplementary Table 22), contrary to the case of aging (Extended Data Fig. S3h), suggesting that cognitive simulation counteracts specific axes of aging dysregulation which do not involve the age-associated brain inflammatory state¹⁰⁵.

Reviewer #3 (Remarks to the Author):

The manuscript titled "Molecular Rejuvenation of the Hippocampus in Enriched Environments" by Pérez et al. investigates the molecular rejuvenation of the hippocampus in mice exposed to enriched environments, focusing on epigenetic mechanisms underlying aging and their modulation by environmental factors. The authors find that environment enrichment can induce a multi-omic rejuvenation of aging-associated alterations, evidenced by the reversal or mitigation of many aging-related changes in gene and protein expression, as well as epigenomic features, in both young and old mice. The research is methodologically robust, integrating multi-omic data from bulk tissue and single-cell analyses to offer a comprehensive view of aging-related alterations and their reversal through environmental enrichment (EE). The study's strengths are its broad and in-depth analysis of various molecular layers, including gene and protein expression, DNA methylation, chromatin accessibility, and histone modifications, both in bulk tissue and at the single-cell level.

We thank the reviewer for taking the time to examine the manuscript and for their valuable comments. We also appreciate that the reviewer recognizes the range of molecular layers analysed in the study and the robustness of the methodology.

First, we must say that we completely agree with the comments made by the reviewer, particularly those specifically addressing the experimental method used (EE) and the lack of mechanistic investigation. We have made changes to the manuscript which considerably change the perspective and conceptual approach of the paper, and we will first provide a general response to explain them.

Below, specific point-by-point responses highlight the specific changes or additions to the manuscript.

As a first general response, we want to emphasize the following, in relation to the concerns raised regarding the effects of EE involving physical and/or mental stimulation:

Certainly, we cannot rule out a mixed effect of increased physical activity plus cognitive stimulation in our experimental intervention, as it is expected that the animals move along longer distances inside the enriched cage compared to control housing animals. In this case we must say that this was to be expected and intended by our study design: i.e. the use of this particular type of environmental stimulation was deliberate. However, we believe that we have not correctly presented and written the manuscript to reflect these ideas:

- a. The first goal of our study was **to study the epigenomics of brain aging, by generating an epigenomic landscape resource** at a scale which was unprecedented in the field.
- b. The second goal of the study was **to explore if any lifestyle/environmental stimulation (in general) could particularly, or specifically, target aging-affected pathways**, or not.

Thus, we selected EE as a "generic" lifestyle/environmental stimulation because: 1) we know it leads to identifiable molecular changes; 2) the brain is a tissue in which epigenetic marks are known to play relevant roles; 3) as the reviewer acknowledged, it is a more "diffuse" stimulus which is in line with the concept of lifestyle as a complex accrual of a diverse set of stimuli (Queen et al. 2020, DOI: 10.3389/fnins.2020.00605). Thus, **EE, or studying the precise nature of EE, was not the main focus of this study nor was this study specifically designed for this**. In fact, our long-term future plan is to carry out studies looking to try diverse specific stimuli (e.g. specific types of physical activity or mental activity) and then to compare their molecular signatures.

It is evident that we have not clearly conveyed these ideas, so we have rewritten key parts of the text (including extensively changing the study Title, Abstract and Introduction) and hope that now the scope of the study and the questions asked (and hopefully answered) are more apparent.

Below, we detail the point-by-point specific changes to accommodate the comments made.

However, there are a few concerns I hope the authors address in a revision:
1. Based on the methodology, it is difficult to distinguish the effects of an enriched environment from simple increased levels of physical and/or mental activity. Some consideration or controls for the effects of physical activity would enhance this study.

As mentioned before, we completely agree with the comment and feel that this is the case. Our enriched environment leads to increased levels of physical and mental activity. We have made considerable changes in text to better reinforce the aforementioned ideas: 1) that we present a multiomic resource and 2) that the focus is to present the epigenomic landscape of aging and the impact of “environmental stimulation”, and not so much to specifically study the EE paradigm, thus reducing the importance of EE:

- We have changed the Title of the manuscript, which now reads as: *“A multiomic atlas of the aging hippocampus reveals molecular rejuvenation in response to environmental stimulation”* and better encapsulates the goal and design of the study.
- We have rewritten the Abstract (see revised manuscript) and part of the Introduction to accommodate the aforementioned ideas (see revised manuscript), including this last paragraph in the Introduction, in which we also mention physical stimulation and how the EE will produce a general stimulation:

[...]

In this study, we have generated a molecular atlas of the murine dorsal hippocampus encompassing gene and protein expression, DNA methylation, chromatin accessibility, histone modifications, and single cell expression and accessibility. We have characterized young and aged mice which were subjected to lifestyle stimulation in the form of environmental enrichment (EE). We have focused on the dorsal hippocampus because it is an important target of both cognitive and physical stimulus where adult neurogenesis occurs^{18,19} and it is known to suffer aging-associated decline linked to cognitive deterioration²⁰. On the other hand, the EE paradigm is a well-established system of general lifestyle stimulation (both cognitive and physical) linked to hippocampal changes at the cellular and molecular level²¹. Thus, with this in-depth map of aging and EE at both the bulk-tissue and single-cell level we have aimed to explore the molecular alterations associated with aging and with environmental stimulation, and also their putative interactions.

- We have also modified parts of the Discussion: we have added a paragraph discussing the limitations of the model used, in terms of it being an unspecific mixture of physical and cognitive (and also social) stimulation:

[...]

To tackle these issues, here we set up a murine model of hippocampal aging which was stimulated by a medium-term lifestyle intervention based on environmental enrichment. The EE paradigm used involves an unspecific stimulation which entails alterations in physical, cognitive, and social activity^{95,21}. As such, the molecular changes observed in this study comprise a mixture of stimuli which cannot be narrowed down to specific behavioural pathways, and better-controlled interventions should be used to dissect the different molecular signatures attributable, for example, to exercise or cognition⁹⁶. Nonetheless, here we have used EE as a laboratory proxy for general lifestyle or environmental stimulation, which is indeed a very complex phenomenon in humans but for which EE is frequently used as a model⁹⁷. We extensively characterized the molecular dynamics of this model by profiling multiple layers of epigenomic regulation, thus generating a molecular map resource to aid understanding of these processes.

[...]

2. Most of the “rejuvenation effects” observed with EE in old animals, have similar or even stronger effects in young animals. Although these are age-affected genes, it would not make sense to say that EE “rejuvenates” young animals or EE have stronger rejuvenation effects in young animals than in old animals. Instead of emphasizing rejuvenation, the data suggest that EE has stronger beneficial effects in young animals than in old animals.

This is a certainly a relevant point which we should address in the discussion and in analyses of the obtained results. In the manuscript, we did classify the EE changes into “age reversed” or “rejuvenated” (see e.g. Figure 4i-j) depending on if they occur to both young and old animals or are observed only specifically in old. This is relevant because, as the reviewer states, a priori it does not make sense to say that EE rejuvenates animals who are already young. Thus, we used the “age reversed” versus “rejuvenation” nomenclature to distinguish them.

Nonetheless, the fact that there are effects of EE in both age groups warrants some explanation:

We think that this phenomenon is actually in line with the most recent literature in the aging field, in which there have been relevant studies demonstrating how aging, or the ticking of molecular clocks, starts straight after pluripotency loss during embryonic development and continues onwards during life (see for instance the works by Kerepesi et al. 10.1126/sciadv.abg6082 and by Kabacik et al. 10.1038/s43587-022-00220-0). Thus, if the accrual of aging molecular aberrations starts from young age, it would make sense that changes that reverse aging alterations in old subjects also reverse “alterations” or similar patterns in young subjects. Because young subjects are already, to a certain extent, aged.

Second, the fact that, when present in both, EE has stronger effects in young subjects also merits discussion in the paper. This observation would be in line with old subjects having in general a reduced plasticity or sensitivity to external stimulation. To test this, we have now performed a new analysis comparing the actual strength of the alterations of these “reversed” genes, in young and old samples, for the gene and protein expression data:

Interestingly, we observe that for gene expression the EE-induced molecular changes are indeed stronger in young subjects (Wilcoxon $p < 0.001$). However, this is not the case for protein level changes, in which both young and old subjects display alterations of similar strength (Wilcoxon $p > 0.05$).

To incorporate all this discussion and findings, we have done the following changes in the manuscript:

- In Results, we have added the following new supplemental panels for Figure S8j-k:

This figure is accompanied by the following text section:

[...]

At the level of gene expression, the effects of EE reversal were stronger in young subjects when compared with their aged counterparts (Wilcoxon both $p < 0.001$, Extended data Fig. S8j), suggesting that these individuals could be more plastic or sensitive to EE stimulation, a finding in agreement with the well-known loss of plasticity occurring during brain aging⁸⁴. Intriguingly, this was not the case for protein changes (Wilcoxon both $p > 0.05$, Extended data Fig. S8k), indicating that, in terms of protein regulation, old subjects could retain responses more comparable to young individuals.
[.]

- In Discussion, we have added the following paragraph:

[...]
A great number of gene and protein expression EE-reversal changes were found in both young and old individuals. This a priori counterintuitive finding could be explained in the light of recent evidence showing that molecular aging starts very early in life^{113,114}, so that young subjects could already have accumulated molecular damage amenable to EE reversal. The strength of the gene expression reversal was stronger in young individuals, as expected due to their increased plasticity⁸⁴, but this was not the case for the reversal of protein levels, with older subjects displaying similar levels of change.
[...]

Due to lack of mechanistic investigation, this manuscript is better suited for publishing as a resource paper.

Indeed, this is certainly an article with a focus on providing an extensive multiomic resource, and as such we have publicly shared all the raw data in ENA and PRIDE repositories. In addition, with the aim of facilitating data access and analysis (and also partly in response to Reviewer #2) we have created a custom Zenodo repository (available at 10.5281/zenodo.8372432 and cited in the Data Availability section) which includes pre-processed datasets (ready for analysis) and also the code used (more than 100 scripts for which we have drafted an extensive documentation detailing their use and output). Additionally, some of the changes in the Title, Abstract and Introduction mentioned in the previous responses also give emphasis to the data generation part.

To emphasize the resource aspect of the manuscript, we have modified Figure 1A, which now shows the specific omic layers profiled in the study, and has the following appearance:

On the other hand, we do believe that any generation of data warrants its in-depth exploration, which is what we have attempted to do in the study and which has led to certain observations of interest. Nonetheless, the mechanistic follow-up of these observations we consider to be out of the scope of this

study (which is, by itself, already probably verging on being too sizable, as has been suggested by Reviewer #2, and we have had to remove some panels from the figures).

REVIEWER COMMENTS

Reviewer #1 (Remarks to the Author):

In this study, Perez et al present a multi-omic atlas of the aging mouse dorsal hippocampus and highlight 3 unique signatures; inflammation, dysregulation of mRNA metabolism and a heterochromatin switch. They used young (9 weeks) and middle-aged (17 months) male mice and performed bulk RNA-seq, proteomics, EM-seq, ATAC-seq, histone modification (H3K4me1, H3K4me3, H3K27ac, H3K36me3, H3K27me3 and H3K9me3) ChIP-seq and single-cell multiome studies to provide an overview of age-related molecular changes. Finally, the authors observed multi-omic rejuvenation of a subset of age-associated alterations when old male mice were exposed to enriched environments (EE). While this study presents an immense body of work, the lack of a central message and novelty slightly dampen enthusiasm. There are additional major concerns I note below:

Major:

1. The following conclusion needs to be clarified – “Taken together, these observations indicate that our study system manifested the typical features of aging and the expected response to the EE paradigm”. From the learning/memory tests done, NOR/NOL did not show any improvement, neither did CFC in the STM phase. EPM is a test of anxiety and EE is known to decrease emotional/anxiety related behavior (reviewed in PMID: 27012955). However, the authors clearly show the opposite. Although the authors try to explain this later, I am unsure whether their EE mice are behaving as expected. Did the authors do a Morris Water Maze in the mice subjected to EE? Relatedly, their 17 month-old mice also had no defects in the NOR/NOL or CFC suggesting their learning and memory are also not impaired. This is a significant concern in this study.

This concern still remains. It is up to the editor to seek more expert advice.

2. The authors attempt to connect the different “omic” layers in their study, but unfortunately a clear combined message is not evident, largely due to the lack of a strong overlap across datasets. Additionally, much of their interpretations are centered on age but not on EE (despite this being shown in many figures early on). It is recommended to show the EE-relevant data only when discussing it in the text (around Fig. 4).

The authors have partially addressed this point by reorganizing the text. But I think this is indeed a resource paper and not an article due to lack of a clear mechanistic link across the omic layers.

3. In general, this manuscript is overloaded with figure panels. Though I understand the importance of showing everything for the sake of transparency, selected analyses can be presented for clarity.

Thank you for simplifying.

4. “Using our segmentations, we confirmed how the previously discovered regions of aging-associated heterochromatin switching transition to Polycomb-associated states in old samples, while losing heterochromatic or non-signal domains (Extended Data Fig. S7f).” – this is not evident in the alluvial plots.

This has been addressed.

5. The enrichment of glial and myelin-related pathways in EE is interesting. Could the authors speculate how EE could be leading to their remodeling?

This has been addressed.

6. In the Discussion section, the relevant figures for each finding should be indicated.

This has been addressed.

7. All code used in this manuscript should be made available on github or similar repository.

This has been addressed.

Minor:

1. In the Extended Abstract – “while uncovering a hitherto undescribed aging-associated heterochromatin-switch phenomenon whereby constitutive heterochromatin loss in the brain is partially mitigated through gains in facultative heterochromatin”. I think this phenomenon has been recently noted in the aging liver and should be cited (PMID: 37116496).

Thank you for citing the paper. However, the interpretation of the data is not correct. The authors in that paper note a chromatin switch (K9me3 to K27me3) in lamin-associated domains but a decrease in H3K27me3 in PRC2-targeted developmental gene promoters which are known to be bivalent.

2. Dataset accession numbers in PRIDE and Zenodo are not specified.

This has been addressed.

3. The comparisons made for statistical tests (for example in Fig. S1, S2) are difficult to understand and lacks consistency. Please clearly indicate all 4 comparisons with single lines over the groups and if not significant mark with a n.s. for all figures. Representative images for immunofluorescence studies in Fig. S1 must be also shown.

This has been addressed.

4. Can the authors explain why they did not use/include the Ximerakis scRNA-seq dataset for deconvolution?

This has been addressed. Please include this data and relevant text in the manuscript.

5. In general, the figure legends need to be expanded so that readers can easily interpret complex analyses. One example is LOLA analysis in multiple figures, where the legend does not expand on the column labels.

This has been addressed.

6. In the Discussion section, the authors state – “Interestingly, while we had observed that inflammation was a major player in functional aging alterations, the EE-rejuvenation changes were more closely linked to neuronal and glial pathways, suggesting that cognitive stimulation counteracts specific axes of aging dysregulation which do not involve the age-associated brain inflammatory state”. The glial pathways could be related to inflammation, for example, microglia activation.

This has been addressed. This Discussion and any relevant figures/tables should be included in revised manuscript.

Reviewer #2 (Remarks to the Author):

I think the authors have substantially addressed the reviewers' concerns. I would recommend accepting this manuscript for publication in Nature Communications.

RESPONSES TO THE REVIEWERS - II

General comments:

In the following response letter we provide the second-iteration responses to the reviewers' comments. The original reviewer comments are indicated in black, their second-iteration responses are indicated in green. Our responses are indicated in blue text, while we use red text to refer to specific changes in the manuscript. In the manuscript document itself, text changes are also indicated with red text.

REVIEWER COMMENTS

Reviewer #1 (Remarks to the Author):

Reviewer #1 (Remarks to the Author):

In this study, Perez et al present a multi-omic atlas of the aging mouse dorsal hippocampus and highlight 3 unique signatures; inflammation, dysregulation of mRNA metabolism and a heterochromatin switch. They used young (9 weeks) and middle-aged (17 months) male mice and performed bulk RNA-seq, proteomics, EM-seq, ATAC-seq, histone modification (H3K4me1, H3K4me3, H3K27ac, H3K36me3, H3K27me3 and H3K9me3) ChIP-seq and single-cell multiome studies to provide an overview of age-related molecular changes. Finally, the authors observed multi-omic rejuvenation of a subset of age-associated alterations when old male mice were exposed to enriched environments (EE). While this study presents an immense body of work, the lack of a central message and novelty slightly dampen enthusiasm. There are additional major concerns I note below:

Major:

1. The following conclusion needs to be clarified – “Taken together, these observations indicate that our study system manifested the typical features of aging and the expected response to the EE paradigm”. From the learning/memory tests done, NOR/NOL did not show any improvement, neither did CFC in the STM phase. EPM is a test of anxiety and EE is known to decrease emotional/anxiety related behavior (reviewed in PMID: 27012955). However, the authors clearly show the opposite. Although the authors try to explain this later, I am unsure whether their EE mice are behaving as expected. Did the authors do a Morris Water Maze in the mice subjected to EE? Relatedly, their 17 month-old mice also had no defects in the NOR/NOL or CFC suggesting their learning and memory are also not impaired. This is a significant concern in this study.

This concern still remains. It is up to the editor to seek more expert advice.

We believe we have responded to the concerns raised to the best of our capacities.

We would like to emphasize that this part of the work has been developed with the participation of several authors who are experts in the use of animal models with paradigms of environmental stimulation and behavioural testing, such as the laboratory of Dr. José Luis Trejo. Their extensive decade-long experience and knowledge in the field supports the robustness of the results and their interpretation (e.g. [10.1073/pnas.1816781116](https://doi.org/10.1073/pnas.1816781116), [10.1096/fj.201801600R](https://doi.org/10.1096/fj.201801600R), [10.1111/adb.13244](https://doi.org/10.1111/adb.13244), [10.1523/JNEUROSCI.2619-20.2021](https://doi.org/10.1523/JNEUROSCI.2619-20.2021), [10.1016/j.neuropharm.2016.12.019](https://doi.org/10.1016/j.neuropharm.2016.12.019), [10.1002/hipo.22568](https://doi.org/10.1002/hipo.22568), etc.).

2. The authors attempt to connect the different “omic” layers in their study, but unfortunately a clear combined message is not evident, largely due to the lack of a strong overlap across datasets. Additionally, much of their interpretations are centered on age but not on EE (despite this being shown in many figures early on). It is recommended to show the EE-relevant data only when discussing it in the text (around Fig. 4).

The authors have partially addressed this point by reorganizing the text. But I think this is indeed a resource paper and not an article due to lack of a clear mechanistic link across the omic layers.

We agree that there is a resource aspect to the work, to emphasize this, we have updated the wording to speak of “multiomic atlas”, “map”, “dataset” and “resource” multiple times across the text, in the Introduction and Discussion, including sentences such as: “[...] Thus, with this in-depth dataset of aging and EE at both the bulk-tissue and single-cell level”, “[...] thus generating a molecular map resource to aid understanding of these processes.”, etc.

Nonetheless, we must say that: 1) we feel that the distinction of considering the article as more resource-based is less of a scientific issue and more about the editorial presentation alternatives of the manuscript; 2) we do believe that our article presents a considerable number of results and analyses

even when compared with other more resourced-based articles. Of course, it is not a molecular, mechanistic research project, but an epigenomic profiling involving data generation and integration analyses.

3. In general, this manuscript is overloaded with figure panels. Though I understand the importance of showing everything for the sake of transparency, selected analyses can be presented for clarity.

Thank you for simplifying.

4. "Using our segmentations, we confirmed how the previously discovered regions of aging-associated heterochromatin switching transition to Polycomb-associated states in old samples, while losing heterochromatic or non-signal domains (Extended Data Fig. S7f)." – this is not evident in the alluvial plots.

This has been addressed.

5. The enrichment of glial and myelin-related pathways in EE is interesting. Could the authors speculate how EE could be leading to their remodeling?

This has been addressed.

6. In the Discussion section, the relevant figures for each finding should be indicated.

This has been addressed.

7. All code used in this manuscript should be made available on github or similar repository.

This has been addressed.

Minor:

1. In the Extended Abstract – "while uncovering a hitherto undescribed aging-associated heterochromatin-switch phenomenon whereby constitutive heterochromatin loss in the brain is partially mitigated through gains in facultative heterochromatin". I think this phenomenon has been recently noted in the aging liver and should be cited (PMID: 37116496).

Thank you for citing the paper. However, the interpretation of the data is not correct. The authors in that paper note a chromatin switch (K9me3 to K27me3) in lamin-associated domains but a decrease in H3K27me3 in PRC2-targeted developmental gene promoters which are known to be bivalent.

We apologize for the oversight. We have modified this section of the Discussion, which now reads as follows:

[...]

This phenomenon of "chromatin switching", which has some parallels with certain experimental models^{77,76,78}, has very recently been described in the aging liver¹⁰⁴. Whereas Yang N. and colleagues reported a generalized H3K27me3-associated heterochromatinization with aging¹⁰⁴, here we describe more balanced, bidirectional H3K27me3 changes. Nonetheless, our results are in line with their observation that localized loss of H3K27me3 is associated with developmental gene promoters while gains of H3K27me3 are linked to H3K9me3 loss at lamin-associated domains¹⁰⁴. We hypothesize that in the brain this latter phenomenon could be caused by the cellular re-repression, using facultative heterochromatin, of the constitutive heterochromatin loss during aging.

[...]

2. Dataset accession numbers in PRIDE and Zenodo are not specified.

This has been addressed.

3. The comparisons made for statistical tests (for example in Fig. S1, S2) are difficult to understand and lacks consistency. Please clearly indicate all 4 comparisons with single lines over the groups and if not significant mark with a n.s. for all figures. Representative images for immunofluorescence studies in Fig. S1 must be also shown.

This has been addressed.

4. Can the authors explain why they did not use/include the Ximerakis scRNA-seq dataset for deconvolution?

This has been addressed. Please include this data and relevant text in the manuscript.

We have finally not included this third deconvolution because the Ximerakis dataset is available only as normalized counts, rather than raw data counts. Although there are papers that use deconvolution algorithms with different data normalizations (10.1038/s41467-020-19015-1), the recommendation for most algorithms is the use of raw counts. Thus, even though the Ximerakis results are a validation of our deconvolution using the other two datasets (Yao and Zeisel) and serve well as exploratory results, we feel safer not including them as it is not the best praxis to use the normalized counts.

5. In general, the figure legends need to be expanded so that readers can easily interpret complex analyses. One example is LOLA analysis in multiple figures, where the legend does not expand on the column labels.

This has been addressed.

6. In the Discussion section, the authors state – “Interestingly, while we had observed that inflammation was a major player in functional aging alterations, the EE-rejuvenation changes were more closely linked to neuronal and glial pathways, suggesting that cognitive simulation counteracts specific axes of aging dysregulation which do not involve the age-associated brain inflammatory state”. The glial pathways could be related to inflammation, for example, microglia activation.

This has been addressed. This Discussion and any relevant figures/tables should be included in revised manuscript.

We have updated the Discussion section while more comprehensively citing figures and tables to address these observations:

[...]

Interestingly, we had initially observed that inflammation was a major player in functional aging alterations, with multiple generic and microglia-associated pathways being detected in the gene set enrichment analyses (see Fig. 1d, Fig. S3e-i, Supplementary Table 3). However, the EE-rejuvenation changes were more closely linked to neuronal and glial pathways (Extended Data Fig. S8I), with no C8 cell type enrichments related to microglia being observed (Supplementary Table 22), contrary to the case of aging (Extended Data Fig. S3h), suggesting that cognitive simulation counteracts specific axes of aging dysregulation which do not involve the age-associated brain inflammatory state¹⁰⁵.

[...]

Reviewer #2 (Remarks to the Author):

I think the authors have substantially addressed the reviewers' concerns. I would recommend accepting this manuscript for publication in Nature Communications.